# Efficient Simple Regret Algorithms for Stochastic Contextual Bandits

## Abstract

We study stochastic contextual logistic bandits under the simple regret objective. While simple regret guarantees have been established for the linear case, no such results were previously known for the logistic setting. Building on ideas from contextual linear bandits and self-concordant analysis, we propose the first algorithm that achieves simple regret $\tilde{\mathcal{O}}(d/\sqrt{T})$. Notably, the leading term of our regret bound is free of the constant $\kappa = \mathcal{O}(\exp(S))$, where $S$ is a bound on the magnitude of the unknown parameter vector. The algorithm is shown to be fully tractable when the action set is finite. We also introduce a new variant of Thompson Sampling tailored to the simple-regret setting. This yields the first simple regret guarantee for randomized algorithms in stochastic contextual linear bandits, with regret $\tilde{\mathcal{O}}(d^{3/2}/\sqrt{T})$. Extending this method to the logistic case, we obtain a similarly structured Thompson Sampling algorithm that achieves the same regret bound — $\tilde{\mathcal{O}}(d^{3/2}/\sqrt{T})$ — again with no dependence on $\kappa$ in the leading term. The randomized algorithms, as expected, are cheaper to run than their deterministic counterparts. Finally, we conducted a series of experiments to empirically validate these theoretical guarantees.

## 1 Introduction

We study stochastic contextual bandits with simple regret (Bubeck et al., 2009), focusing on both linear and logistic models. In each round, the learner observes a set of actions and a context drawn independently from the same distribution. The learner selects one of the actions and observes a corresponding reward, whose expectation — conditioned on the chosen action and observed context — is a function of the inner product of the known feature vector of the action and the context, and an unknown parameter vector of dimension $d$. The goal is to identify a policy that maps contexts to actions, minimizing the expected regret over future draws from the context distribution.

We start with the linear case, which is simpler and included primarily for completeness. We propose a deterministic algorithm that selects the action with the most uncertain predicted reward, an idea also used in Zanette et al. (2021). We explain the difference between our algorithm and that of Zanette et al. (2021) in detail later in this section. We also include a randomized version based on Thompson sampling (TS) (Thompson, 1935), which samples a parameter from the posterior distribution constructed using observed contexts and *zero-valued* rewards, and chooses the action whose feature vector, up to a potential change of sign, is best aligned with the chosen parameter vector. We show that this randomized algorithm enjoys a simple regret of $\tilde{\mathcal{O}}(d^{3/2}/\sqrt{T})$ which matches the dimension dependency in stochastic linear bandits with cumulative regret (Agrawal & Goyal, 2013; Abeille & Lazaric, 2017).

For the logistic case (Filippi et al., 2010), we extend the uncertainty-based action selection approach by incorporating the nonlinearity of the problem. The deterministic algorithm estimates a lower bound on the Hessian of the logistic loss to compute meaningful uncertainty estimates. It then jointly selects a parameter (from a carefully constructed confidence set) and an action that maximizes the resulting uncertainty. This is the first algorithm in the logistic setting to achieve simple regret of $\tilde{\mathcal{O}}(d/\sqrt{T})$ where the leading term is free of $\kappa$, a potentially large constant that appeared in early work on cumulative regret minimization, matching the seminal result of Faury et al. (2020). In particular, $\kappa = \exp(S)$ where $S$ is a bound on the $\ell_2$-norm of the optimal, unknown parameter vector, which

needs to be given to all the algorithms published to date. The algorithm is also tractable when the action set is finite. We also analyze a randomized logistic algorithm based on TS. Here, similarly to the linear case, a parameter is sampled from the posterior induced by pseudo-observations (rewards set to zero). Then, the action with the highest predicted uncertainty given this parameter is selected. As expected, the randomized method saves computation over the deterministic method. We prove a regret bound of $\tilde{\mathcal{O}}(d^{3/2}/\sqrt{T})$, again with no $\kappa$-dependency in the leading term.

As is mentioned before, the linear contextual bandit problem under the simple regret objective was also studied by Zanette et al. (2021). Their approach requires a pre-collected dataset of contexts on which they compute a data-collection policy then collect online data using this policy. However, the two-phase structure makes the method less suitable for online use, as it requires deciding in advance how to split the time horizon between the two phases. As such, more modifications to the algorithm are needed when the total number of rounds is unknown. Our work complements this by providing a simpler, fully online approach with matching regret guarantees. Although we do not explore this direction here, our method can also be easily adapted to settings where some contexts are pre-collected. For these reasons, we view the fully online algorithm as more versatile: it avoids phase-length tuning without requiring any additional parameters. Lastly, the linear contextual bandit problem can be viewed as a special case of the linear Markov Decision Process (MDP) problem with a horizon of 1. We leave more detailed discussions of the work on linear MDPs to Appendix A.

For the logistic case, the simple regret setting has, to our knowledge, not been previously studied. The logistic bandit model was introduced as a special case of generalized linear bandits by Filippi et al. (2010). In that work, the regret bounds for cumulative regret included the earlier mentioned, potentially very large, constant $\kappa$ in the leading term. Subsequent work (Jun et al., 2017; Li et al., 2017) have been suffering from $\kappa$ until Faury et al. (2020) used the self-concordance of the logistic loss, which was first proposed by Bach (2010), to push $\kappa$ into lower-order terms, and Abeille et al. (2021) proposed an instance-wise minimax-optimal optimistic algorithm with a matching lower bound. TS and its variants in logistic bandits under cumulative regret are studied in Abeille & Lazaric (2017); Kveton et al. (2020); Ding et al. (2021); Jun et al. (2017); Faury et al. (2022). In contrast, our work initiates a study of simple regret for logistic bandits, and provides the first deterministic and randomized algorithms with leading-order guarantees that are independent of $\kappa$.

The main technical novelty of our work is three fold. Firstly, We provide a completely new way to analyze randomized algorithms in stochastic contextual linear (logistic) bandits under simple regret criteria and the idea is completely different from the cumulative case. We believe that our analysis could also serve as a starting point for other related problems (e.g. stochastic contextual generalized linear bandits under simple regret). Secondly, on the deterministic logistic bandit algorithm side, we did substantive algorithmic design and more involved analysis to handle the complexities of the logistic bandit model, where we borrowed tools from Faury et al. (2020); Abeille et al. (2021). To complement our theoretical results, we also provide empirical comparisons of the proposed algorithms with natural baselines on synthetic data. Thirdly, we correct a technical mistake in Zanette et al. (2021) and provide a corrected version of their martingale concentration result (Theorem 3 in their paper) in Appendix F.

## 2 Preliminaries

In this section we introduce the notation, followed by explaining the problem formulation. We finish the section with the review of some tools that will be useful for discussing our algorithms and results.

### 2.1 Notation

For a real-valued single-variable differentiable function $f : \mathbb{R} \to \mathbb{R}$, we use $\dot{f}, \ddot{f}$ to denote the first and second order derivatives of $f$, respectively. The $\ell_2$-norm on $\mathbb{R}^d$ is denoted by $\| \cdot \|$. We use $\mathbb{S}^{d-1}$ and $\mathbb{B}_d(r)$ to denote the unit sphere and the $d$-dimensional ball with radius $r$ in $(\mathbb{R}^d, \| \cdot \|)$, respectively. For a positive definite matrix $A$, we define the norm induced by $A$ to be $\|x\|_A = \sqrt{x^\top A x}$. For a set $\mathcal{K}$, let $\mathcal{M}_1(\mathcal{K})$ denote the set of all distributions over $\mathcal{K}$; we assume the associated measurability structure is clear from context. For a distribution $P \in \mathcal{M}_1(\mathcal{K})$, we write $\mathrm{supp}(P) \subseteq \mathcal{K}$ to denote its support. The abbreviation "a.s." refers to statements that hold almost surely. For positive semidefinite

matrices $P, S$, we use $P \preceq S$ to denote that $S - P$ is positive semidefinite. We use $\mathbb{R}^+$ to denote the positive reals.

## 2.2 Problem Setup

We study stochastic logistic and linear contextual bandits with stochastic contexts. A bandit instance is described by a tuple $(\mathcal{S}, \mathcal{A}, \nu, \mathcal{P}, \mu, \phi, \theta_*, )$. Here $\mathcal{S}$ is the context space that potentially could be infinite. The set of available actions can depend on the context; the structure $\mathcal{A}$ specifies these sets. In particular, when the context is $s \in \mathcal{S}$, the actions available are the elements of the set $\mathcal{A}(s)$, which we assume to be finite.[1] We let $\mathcal{Z} = \cup_{s \in \mathcal{S}}(\{s\} \times \mathcal{A}(s))$ denote the context-action space. Next, $\nu \in \mathcal{M}_1(\mathcal{S})$ is a distribution over the contexts and without loss of generality (w.l.o.g.) we assume $\mathrm{supp}(\nu) = \mathcal{S}$. $\mathcal{P}$ is a probability kernel from $\mathcal{Z}$ to the reals: For $(s, a) \in \mathcal{Z}$, $P(\cdot|s, a) \in \mathcal{M}_1(\mathbb{R})$ is a distribution of the reward that will be received when in context action $a$ is chosen. Finally, $\mu : \mathbb{R} \to \mathbb{R}$, $\phi : \mathcal{Z} \to \mathbb{R}^d$, $\theta_* \in \mathbb{R}^d$ and these are such that $\mu(\phi(s, a)^\top \theta_*) = \int r P(dr|s, a)$ gives the expected reward associated with the pair $(s, a) \in \mathcal{Z}$. It is assumed that for each $(s, a) \in \mathcal{Z}$, after centering, the reward distribution $P(dr|s, a)$ is 1-subgaussian.

For $T \geq 1$, the learner interacts with the environment in rounds $t = 1, 2, \ldots, T$. The learner does have access to $\mathcal{S}, \mathcal{A}$ and $\phi$, but does not know $\nu, \mathcal{P}$ or $\theta_*$. At the beginning of each round $t$, the learner observes a context $S_t \in \mathcal{S}$ sampled from $\nu$. Next, given its past information, the learner chooses an action $A_t \in \mathcal{A}(S_t)$, after which they receive the reward

$$X_t \sim \mathcal{P}(\cdot|S_t, A_t).$$

Here, the meaning of the above identity is that given the past $(S_1, A_1, X_1, \ldots, S_t, A_t)$, the distribution of $X_t$ is $\mathcal{P}(\cdot|S_t, A_t)$. We will find it useful to introduce the "noise" associated with the reward $X_t$, $\epsilon_t = X_t - \mu(\phi(S_t, A_t)^\top \theta_*)$. For *linear contextual bandits*, $\mu$ is the identity function (i.e., $\mu(x) = x$ for all $x \in \mathbb{R}$). For *logistic contextual bandits*, $\mu(\cdot)$ is the logistic function, i.e., $\mu(x) = \frac{1}{1+\exp(-x)}$ and $\mathcal{P}(\cdot|s, a)$ is the Bernoulli distribution with parameter $\mu(\phi(s, a)^\top \theta_*)$.

The goal of the learner is to learn enough about the instance to be able to produce a policy $\pi$ that gives as much reward as possible when deployed. Here, a policy is a map from $\mathcal{S}$ to $\cup_{s \in \mathcal{S}} \mathcal{A}(s)$ such that for any $s \in \mathcal{S}$ context, $\pi(s) \in \mathcal{A}(s)$. We define the *value* of a policy $\pi$ to be

$$v(\pi) = \int \nu(ds) \int r\mathcal{P}(dr|s, \pi(s)) = \int \mu(\phi(s, \pi(s))^\top \theta_*)\nu(ds),$$

which gives the expected reward when action $\pi(s)$ is used the second equality is by our assumptions. Let $\Pi$ denote the set of all possible policies. The simple regret of a policy is defined to be

$$\mathfrak{R}(\pi) = \sup_{\pi' \in \Pi} v(\pi') - v(\pi) \tag{1}$$

which is the loss compared to using an optimal policy. Since we assumed that all action sets are finite, an optimal policy $\pi^*$ exists and is determined by $\pi^*(s) = \arg\max_{a \in \mathcal{A}(s)} \mu(\phi(s, a)^\top \theta_*)$.[2]

We assume that the features are normalized in the following sense:

**Assumption 1.** *For all $(s, a) \in \mathcal{Z}$, $\|\phi(s, a)\| \leq 1$.*

The algorithms also need to know an upper bound on the norm of the unknown parameter:

**Assumption 2.** *There is a* known *constant $S > 0$ such that $\|\theta_*\| \leq S$.*

These assumption are standard when studying generalized linear bandits (Faury et al., 2020; Abeille et al., 2021; Janz et al., 2024; Lee et al., 2024; Liu et al., 2024).

In the logistic case, the nonlinearity of the mean function $\mu$ with respect to the parameter $\theta$ is measured by the following quantity (Filippi et al., 2010; Faury et al., 2020)

$$\kappa = \sup_{u \in \phi(\mathcal{Z}), \theta \in \mathbb{B}_d(S)} \frac{1}{\dot{\mu}(u^\top \theta)}. \tag{2}$$

---

[1]Or an infinite set such that $\{\phi(s, a)\}_{a \in \mathcal{A}(s)}$ is a compact subset of $\mathbb{R}^d$.

[2]For the extension to infinite action sets, $\pi^*$ also exists because a continuous image of a compact set is compact and hence the maximum is attained.

Here, $\phi(\mathcal{Z}) = \{\phi(z) : z \in \mathcal{Z}\}$, as usual. As noted in Faury et al. (2020), from the definition of $\kappa$ and the sigmoid function $\mu$, it holds that $\kappa = 1 + \max_{u \in \phi(\mathcal{Z}), \theta \in \mathbb{B}_2(d)} \exp(u^\top \theta)$, which could scale exponentially with the size of the admissible parameter set $S$, for example, when $\phi(\mathcal{Z}) = \mathbb{B}_d(1)$.

## 2.3 LINEAR AND LOGISTIC REGRESSION

Given the data $\mathcal{D} = \{(\phi(S_i, A_i), X_i)\}_{i=1}^{t-1}$ collected from an environment, linear (respectively, logistic) regression estimates the unknown parameter $\theta_*$ by minimizing some loss $\mathcal{L}_t^\lambda : \mathbb{R}^d \to \mathbb{R}$: $\hat{\theta}_t = \arg\min_{\theta \in \mathbb{R}^d} \mathcal{L}_t^\lambda(\theta)$. The estimators differ in terms of the loss used. In this work, we will consider regularized versions of the respective losses.

**Linear regression**   For the linear case, we consider the regularized squared loss defined via

$$\mathcal{L}_t^\lambda(\theta) := \lambda\|\theta\|^2 + \sum_{i=1}^{t-1}(X_i - \phi(S_i, A_i)^\top\theta)^2. \tag{3}$$

The regularized least-squares (RLS) estimator returns the minimizer of this loss. This minimizer in this case is availably in closed form: $\hat{\theta}_t = V_t^{-1}\sum_{i=1}^{t-1}X_i\phi(S_i, A_i)$, where

$$V_t = \lambda I + \sum_{i=1}^{t-1}\phi(S_i, A_i)\phi(S_i, A_i)^\top. \tag{4}$$

**Logistic regression**   For the logistic case, we consider the regularized negative log-likelihood function associated with our probabilistic model:

$$\mathcal{L}_t^\lambda(\theta) = \lambda\|\theta\|^2 - \sum_{i=1}^{t-1}\ell(\mu(\phi(S_i, A_i)^\top\theta), X_i), \tag{5}$$

where $\ell(x, y) = y\log(x) + (1 - y)\log(1 - x)$ is the binary cross-entropy function. (To reduce clutter, we recycle $\mathcal{L}_t^\lambda$ to denote the loss both for the linear and logistic case: the meaning of $\mathcal{L}_t^\lambda$ should always be clear from the context.) The minimizer of this loss will be called the regularized maximum likelihood estimate (MLE). While in this case, the minimizer is not available in closed form, since the loss $\mathcal{L}_t^\lambda$ is strongly convex, $\hat{\theta}_t$ is the unique stationary point of $\mathcal{L}_t^\lambda$. Defining $g_t$ by $g_t(\theta) = \sum_{i=1}^{t-1}\mu(\phi(S_i, A_i)^\top\theta)\phi(S_i, A_i) + \lambda\theta$, it follows that

$$\nabla_\theta\mathcal{L}_t^\lambda(\hat{\theta}_t) = g_t(\hat{\theta}_t) - \sum_{i=1}^{t-1}\phi(S_i, A_i)X_i = 0, \tag{6}$$

and the above equation has no other solution. We will also need the Hessian of the loss:

$$H_t(\theta) := \nabla_\theta^2\mathcal{L}_t(\theta) = \lambda I + \sum_{i=1}^{t-1}\dot{\mu}(\phi(S_i, A_i)^\top\theta)\phi(S_i, A_i)\phi(S_i, A_i)^\top. \tag{7}$$

## 3 ALGORITHMS AND RESULTS

In this section, we will present our algorithms and the associated main results, both for the linear and the logistic cases. We start with presenting the deterministic algorithms for both settings as the intuition developed from the design and the analysis of these algorithms will be helpful for the design and the analysis of the randomized algorithms.

## 3.1 DIRECT UNCERTAINTY MAXIMIZATION: DETERMINISTIC ALGORITHMS

The main idea of the deterministic algorithms is to choose an action with maximal associated uncertainty. That is, given the context $S_t$ for each round, the learner computes an uncertainty score associated with each of the actions and then chooses the action that maximizes it. The way the uncertainty score is calculated depends on the setting, but the general idea is that given the data

**Algorithm 1** Max-Uncertainty-Lin (MULIN)

**Require:** $\lambda > 0$
1: $V_1 = \lambda I, \mathcal{D} = \{\}$    // $\mathcal{D}$ is a multiset
2: **for** $t : 1 \to T$ **do**
3:    Observe context $S_t \sim \nu$
4:    Select $A_t = \arg\max_{a \in \mathcal{A}(S_t)} \|\phi(S_t, a)\|_{V_t^{-1}}$
5:    Receive reward $X_t$
6:    $\mathcal{D} = \mathcal{D} \cup \{(S_t, A_t, X_t)\}$
7: **end for**
8: $\hat{\theta}_T = V_{T+1}^{-1} \sum_{t=1}^{T} X_t \phi(S_t, A_t)$
9: Return $\hat{\pi} : s \mapsto \arg\max_{a \in \mathcal{A}(s)} \phi(s, a)^\top \hat{\theta}_T$

**Algorithm 2** Max-Uncertainty-Log (MULOG)

**Require:** $\lambda > 0, S > 0$
1: $L_1 = \lambda I, \mathcal{D} = \{\}$    // $\mathcal{D}$ is a multiset
2: **for** $t : 1 \to T$ **do**
3:    Observe context $S_t \sim \nu$
4:    Solve for $\hat{\theta}_t$ and build $\mathcal{W}_t$ (Eq. (12))
5:    Select $(A_t, \theta_t)$ by Eq. (10) and receive $X_t$
6:    $\mathcal{D} = \mathcal{D} \cup \{(S_t, A_t, X_t)\}$
7:    Solve for $\theta'_t$ and update $L_{t+1}$ as in Eq. (8)
8: **end for**
9: Solve for $\hat{\theta}_{T+1}$ and build $\mathcal{W}_{T+1}$ (Eq. (12))
10: Pick any $\theta_{T+1}^{\mathrm{Log}} \in \mathcal{W}_{T+1}$.
11: Return $\hat{\pi} : s \mapsto \arg\max_{a \in \mathcal{A}(s)} \phi(s, a)^\top \theta_{T+1}^{\mathrm{Log}}$

available to the learner, it should be the width of the tightest confidence interval that the learner can use in predicting the reward associated to the context and the action. By greedily choosing the action that maximizes uncertainty, the learner aims to shrink the width of the associated confidence intervals as fast as possible. Thus, for each of the cases, two things remain: *(i)* constructing the uncertainty score and *(ii)* establishing a guarantee that shows the uncertainty will be sufficiently reduced. Note that the challenge lies in the fact that, with finite data, the learner cannot reduce uncertainty for all possible future contexts—as some low probability contexts may never be observed. Nevertheless, as we will show, this simple strategy still ensures uniform progress over time. When the exploration phase is terminated, the algorithm produces a parameter estimate by minimizing the associated loss and returns the policy that is optimal under the given parameter estimate.

**MULIN: Maximizing Uncertainty in the Linear Case** In round $t$, the uncertainty score of $(S_t, a) \in \mathcal{Z}$ is $\|\phi(S_t, a)\|_{V_t^{-1}}$ where $V_t$ is the regularized design matrix (see Eq. (4)). The pseudocode of the full method is given in Algorithm 1. Note that the algorithm does not use the rewards for action selection. The rewards are nevertheless stored and used at the end in choosing the policy to be returned.

The key idea of our analysis, borrowed from Zanette et al. (2021), is that the worst-case uncertainty is guaranteed to decrease during the execution of the algorithm. Indeed, since $V_t \preceq V_{t+1}$ and $V_{t+1}^{-1} \preceq V_t^{-1}$, we have for any $(s, a) \in \mathcal{Z}$ that $\|\phi(s, a)\|_{V_{t+1}^{-1}} \leq \|\phi(s, a)\|_{V_t^{-1}}$. Taking the maximum over the actions and integrating over the context using $\nu$ gives the following simple lemma:

**Lemma 1.** *[Decreasing Uncertainty Lemma, Lemma 6 of Zanette et al. (2021)] For every $t \geq 1$, it holds that*

$$\int \max_{a \in \mathcal{A}(s)} \|\phi(s, a)\|_{V_{t+1}^{-1}} \nu(ds) \leq \int \max_{a \in \mathcal{A}(s)} \|\phi(s, a)\|_{V_t^{-1}} \nu(ds).$$

Thus, the elliptical potential lemma (Abbasi-Yadkori et al., 2011), a standard tool in linear sequential analysis, can be used to upper bound the expected predictive uncertainty. Altogether, this leads to the following result:

**Theorem 1** (MULIN Simple Regret Bound). *Under Assumptions 1 and 2, there exists some universal constant $\mathfrak{c} > 0$ such that the following holds: Let $\delta \in [0, 1)$, $T \geq 1$ be arbitrary. Then, with probability at least $1 - \delta$, it holds that the simple regret of the policy $\hat{\pi}$ computed by MULIN (Algorithm 1) with an appropriate choice of $\lambda$, after $T$ rounds is upper bounded by*

$$\mathfrak{R}(\hat{\pi}) \leq \mathfrak{c} \, d \sqrt{\log(T/\delta)/T}.$$

The explicit choice of $\lambda$, which only depends on $\delta, \log(T)$ and $S$, is given in Appendix B.3. For this and all subsequent results, their proofs are given in the appendix.

**MULOG: Maximizing Uncertainty in the Logistic Case** Let us now turn to the case of logistic bandits. Define the uncertainty associated with a state-action pair $(s, a) \in \mathcal{Z}$, a parameter vector $\theta \in \mathbb{R}^d$ and a positive definite matrix $L$ using

$$U(s, a, \theta, L) := \dot{\mu}(\phi(s, a)^\top \theta) \|\phi(s, a)\|_{L^{-1}}.$$

Here, the term $\|\phi(s,a)\|_{L^{-1}}$ is similar to the one we have seen in the linear case. This term can be thought of governing the uncertainty of the parameter of the Bernoulli distribution associated with $(s,a) \in \mathcal{Z}$. As opposed to the linear case, to make this work, it is not sufficient to use $L = V_t$. Indeed, from Eq. (7), we see that if the loss is approximated using a second-order Taylor expansion at $\theta = \theta_*$, the terms $\phi(S_i, A_i)\phi(S_i, A_i)^\top$ need to be multiplied by $\dot\mu(\phi(S_i, A_i)^\top \theta_*)$ (for any $x \in \mathbb{R}$, the variance of the Bernoulli distribution with mean $\mu(x)$ happens to be $\dot\mu(x)$, hence the origin of $\dot\mu$ in these expressions). Since $\theta_*$ is not available, we need to be conservative in computing the uncertainty. Therefore our algorithm will use carefully constructed confidence sets to get lower bounds on $H_t(\theta_*)$. In round $t$, we propose to use the following matrix:

$$L_t = \lambda I + \sum_{i=1}^{t-1} \dot\mu(\phi(S_i, A_i)^\top \theta_i')\phi(S_i, A_i)\phi(S_i, A_i)^\top \text{ where } \theta_i' = \arg\min_{\theta \in \mathcal{W}_i} \dot\mu(\phi(S_i, A_i)^\top \theta), \quad (8)$$

where $\mathcal{W}_i$ is a confidence set available at the beginning of round $i$ that will be defined momentarily. Since decreases as $|z|$ increases, the minimization in Eq. (8) is equivalent to

$$\theta_i' = \arg\max_{\theta \in \mathcal{W}_i} |\phi(S_i, A_i)^\top \theta| \quad (9)$$

Note that when $\mathcal{W}_i$ is convex (which will be our case), this problem can be solved by finding both the maximizer and the minimizer of $\phi(S_i, A_i)^\top \theta$ over $\mathcal{W}_i$ and then choosing the one that gives the highest absolute value. Since both the objective in these problems is linear, the two subproblems are convex minimization problems and as such can be efficiently solved.

The other term $\dot\mu(\phi(s,a)^\top \theta)$ in $U$ comes from the first-order Taylor series expansion of $\theta \mapsto \mu(\phi(s,a)^\top \theta)$. Hence, the value of $\theta$ to be used here should be close to the true value. Since, again, we want a conservative estimate, we use our confidence set $\mathcal{W}_t$ to find the value that gives the largest uncertainty estimate for each action, leading to

$$(A_t, \theta_t) = \arg\max_{a \in \mathcal{A}(S_t), \theta \in \mathcal{W}_t} U(S_t, a, \theta, L_t). \quad (10)$$

By the choice of $\mu$, using the same argument as before, for any fixed action $a \in \mathcal{A}(S_t)$, the optimization problem can be replaced by the minimization of the convex function $\theta \mapsto |\phi(S_t, a)\theta|$ over $\mathcal{W}_t$ and as such is tractable.

It remains to choose the confidence sets $(\mathcal{W}_t)_{t \geq 1}$ and the policy returned at the end. For the confidence set construction, the algorithm first solves for the unconstrained MLE $\hat\theta_t$. Letting

$$\mathcal{C}_t(\delta, \theta_\circ) = \left\{ \theta : \mathcal{L}_t^\lambda(\theta) - \mathcal{L}_t^\lambda(\theta_\circ) \leq \beta_t(\delta) \right\}, \quad (11)$$

where for $\delta \in (0,1]$, $\beta_t(\delta) : \mathbb{R}^+ \to \mathbb{R}^+$ is an increasing function in $t$ whose value is introduced in Appendix D.2 in detail, we choose

$$\mathcal{W}_t = \cap_{i=1}^{t-1} \mathcal{C}_i(\delta, \hat\theta_i) \cap \mathbb{B}_d(S), \quad (12)$$

so that $\{\mathcal{W}_t\}_{t \geq 1}$ decreases, a step necessary to ensure the uncertainty measures decrease – a key requirement in the analysis. The following result is borrowed from Abeille et al. (2021):

**Lemma 2.** *(Lemma 1 of Abeille et al. (2021)) Let $\delta \in [0,1)$. It follows that $\mathbb{P}(\forall t \geq 1, \theta_* \in \mathcal{C}_t(\delta, \hat\theta_t)) \geq 1 - \delta$.*

Lemma 2 implies that $\theta_*$ lies in $\mathcal{W}_t$ for all $t \geq 1$ with probability at least $1 - \delta$. In the evaluation phase, we repeat the procedure described above to construct the confidence set $\mathcal{W}_{T+1}$ in which we pick any vector $\theta_{T+1}^{\text{Log}}$. When implementing our algorithm, one can employ different ways to do so, e.g., projecting $\hat\theta_{T+1}$ to $\mathcal{W}_{T+1}$. The output policy $\hat\pi$ acts greedily w.r.t. $\theta_{T+1}^{\text{Log}}$. Algorithm 2 is computationally efficient for finite action sets. Specifically, lines 4,7 are convex optimization problems. For line 5, one can first iterate over $\mathcal{A}(S_t)$ and the fact that the maximizer of $\dot\mu(\phi(s,a)^\top \theta)$ is the minimizer of $|\phi(s,a)^\top \theta|$ makes the problem convex.

The following lemma is written in a slightly more general form for reusing it in analyzing THATS, the Thompson sampling algorithm in this setting, which will be introduced in Section 3.2. The proof is deferred to Appendix D.3. When applying it to MULog, we let $K = L_t$, $K' = L_{t+1}$, $\mathcal{Y} = \mathcal{W}_t$ and $\mathcal{Y}' = \mathcal{W}_{t+1}$. As seen from the proof that $\mathcal{W}_t$ is shrinking and $L_t$ is increasing is critical. In fact, one motivation to use the specific sequence $L_t$ was to ensure this increasing property.

**Algorithm 3** Simple Regret Linear Thompson Sampling (SIMPLELINTS)

**Require:** $\lambda > 0$
1: $V_1 = \lambda I, \mathcal{D} = \{\}$  $\quad$ // $\mathcal{D}$ is a multiset
2: **for** $t : 1 \to T$ **do**
3: $\quad$ Observe context $S_t \sim \nu$
4: $\quad$ Sample $\tilde{\theta}_t \sim \mathcal{N}(0, V_t^{-1})$
5: $\quad$ Select $A_t$ by Eq. (13) and receive $X_t$
6: $\quad$ $\mathcal{D} = \mathcal{D} \cup \{(S_t, A_t, X_t)\}$
7: **end for**
8: $\hat{\theta}_{T+1} = V_{T+1}^{-1} \sum_{t=1}^{T} X_t \phi(S_t, A_t)$
9: Return $\hat{\pi} : s \mapsto \arg\max_{a \in \mathcal{A}(s)} \phi(s, a)^\top \hat{\theta}_{T+1}$

**Algorithm 4** Try Hard Thompson Sampling (THATS)

**Require:** $\lambda > 0$
1: $L_1 = \lambda I, \mathcal{D} = \{\}$  $\quad$ // $\mathcal{D}$ is a multiset
2: **for** $t : 1 \to T$ **do**
3: $\quad$ Observe context $S_t \sim \nu$
4: $\quad$ Sample $\tilde{\theta}_t \sim \mathcal{N}(0, L_t^{-1})$
5: $\quad$ Solve $\bar{\theta}_t$ and construct $\mathcal{V}_t$ by Eq. (15)
6: $\quad$ Select $A_t$ by Eq. (16) and receive $X_t$
7: $\quad$ $\mathcal{D} = \mathcal{D} \cup \{(S_t, A_t, X_t)\}$
8: $\quad$ Solve $\theta'_t$ and build $L_{t+1}$ as in Eq. (14)
9: **end for**
10: Solve $\bar{\theta}_{T+1}$ (15) and construct $\mathcal{V}_{T+1}$ by (17)
11: Pick any $\theta_{T+1}^{\mathrm{Log}} \in \mathcal{V}_{T+1}$
12: Return $\hat{\pi} : s \mapsto \arg\max_{a \in \mathcal{A}(s)} \phi(s, a)^\top \theta_{T+1}^{\mathrm{Log}}$

**Lemma 3** (Decreasing Uncertainty Lemma – Logistic Bandits). *Let $K' \succeq K$ be $d \times d$ positive definite matrices and $\mathcal{Y}' \subseteq \mathcal{Y} \subseteq \mathbb{R}^d$ bounded closed sets. Then,*

$$\int \max_{a \in \mathcal{A}(s), \theta \in \mathcal{Y}'} U(s, a, \theta, K') \nu(ds) \leq \int \max_{a \in \mathcal{A}(s), \theta \in \mathcal{Y}} U(s, a, \theta, K) \nu(ds).$$

The proof can be finished based on this lemma just like before, though due to the presence of the nonlinear function $\mu$, the proof becomes significantly more technical.

**Theorem 2** (MULOG Simple Regret Bound). *Under Assumptions 1 and 2, there exists some universal constant $\mathfrak{c} > 0$ such that the following holds: Let $\delta \in [0, 1)$, $T \geq 1$ be arbitrary. Then, with probability at least $1 - \delta$, it holds that the simple regret of the policy $\hat{\pi}$ computed by MULOG (Algorithm 2) with an appropriate choice of $\lambda$, after $T$ rounds is upper bounded by*

$$\mathfrak{R}(\hat{\pi}) \leq \mathfrak{c} \, d \sqrt{\log(T/\delta)/T} + \mathrm{poly}\left(d, \kappa, \log(T), \log(1/\delta)\right)/T.$$

The explicit choice of $\lambda$, which only depends on $\delta, T, S$, and proof, is given in Appendix D.3.

**Connection and difference to Faury et al. (2020)** $\quad$ Regarding the use of $L_t$, naively replacing $V_t$ with $H_t(\theta)$, which seems to be a natural choice inherited from the cumulative regret side, would not achieve decreasing uncertainty without introducing a factor of $\sqrt{\kappa}$, no matter what $\theta$ is. One way to deal with this is to employ a matrix quantifying uncertainty that can be built in an online fashion. Faury et al. (2020) also considers such a matrix on which we built our idea of using $L_t$ to quantify uncertainty. Nevertheless, our algorithm is completely different from Log-UCB-2 of Faury et al. (2020) in spirit. We leave detailed comparisons to Appendix D.4.

### 3.2 RANDOMIZED ALGORITHMS

In this section, we first present a simple regret Thompson Sampling (SIMPLELINTS) algorithm for linear contextual bandits. Then we extend the idea to the logistic setting, giving a new algorithm called Try Hard Thompson Sampling (THATS).

**SIMPLELINTS: Linear Thompson Sampling** $\quad$ The key difference between SIMPLELINTS and MULIN is that instead of maximizing the uncertainty in each step, we sample a random vector $\tilde{\theta}_t$ from $\mathcal{N}(0, V_t^{-1})$ first. The action is then chosen to be the one that, up to a sign flip, aligns with the direction of $\tilde{\theta}_t$ most:

$$A_t = \arg\max_{a \in \mathcal{A}(S_t)} |\phi(S_t, a)^\top \tilde{\theta}_t|. \tag{13}$$

This choice can be justified as follows: We have seen that a reasonable choice in round $t$ is to choose the action whose feature vector $u \in \mathbb{R}^d$ maximizes $||u||_{V_t^{-1}}$. Now, when $\tilde{\theta} \sim \mathcal{N}(0, V^{-1})$ for

some positive definite matrix $V$, $\mathbb{E}(u^\top \tilde{\theta})^2 = ||u||^2_{V^{-1}}$. Thus, $(u^\top \tilde{\theta})^2$ can be seen as a one-sample Monte-Carlo approximation to $||u||^2_{V^{-1}}$ and maximizing this score can be seen as an approximate approach to maximizing $||u||^2_{V^{-1}}$.

Similarly to the cumulative regret setting, the exploration done by SIMPLELINTS is not as effective as the one done by MULIN because the direction of $\tilde{\theta}_t$ does not necessarily suggest the maximum uncertainty direction $\phi(S_t, A_t^{\mathrm{MU}})$. As it turns out, one can lower bound the expected uncertainty achieved by $A_t$ relative to the maximum uncertainty by, roughly, the expected normalized correlation between $M_t = V_t^{1/2}\tilde{\theta}_t$ and $V_t^{-1/2}\phi(S_t, A_t^{\mathrm{MU}})$ given the past. This is the subject of the next lemma which is the key to our contribution on the new analysis in randomized algorithms for linear (logistic) contextual bandit promised in Section 1. Since the distribution of $M_t$, given the past and $S_t$, is standard normal, the distribution of $M_t/||M_t||$ is uniform on the sphere:

**Lemma 4.** *Fix $t \geq 1$. Then, almost surely,*

$$\|\phi(S_t, A_t^{\mathrm{MU}})\|_{V_t^{-1}} \cdot I(S_t, A_t^{\mathrm{MU}}, V_t) \leq \mathbb{E}\left[\|\phi(S_t, A_t)\|_{V_t^{-1}}\Big|\mathcal{F}_{t-1}, S_t\right],$$

*where for $(s, a) \in \mathcal{Z}$ and $V \succeq 0$, we let $I(s, a, V) = \int_{\mathbb{S}^{d-1}} \left|\left\langle x, \frac{V^{-1/2}\phi(s,a)}{\|V^{-1/2}\phi(s,a)\|}\right\rangle\right| dx$.*

With standard tools in probability theory, $I(s, a, V) = \Omega(1/\sqrt{d})$ for all $(s, a) \in \mathcal{Z}$ and $V \succ 0$. Lemma 4 combined with the previous arguments developed for MULIN gives the following result:

**Theorem 3.** *Under Assumptions 1 and 2, there exists some universal constant $\mathfrak{c} > 0$ such that the following holds: Let $\delta \in [0, 1)$, $T \geq 1$ be arbitrary. Then, with probability at least $1 - \delta$, it holds that the simple regret of the policy $\hat{\pi}$ computed by SIMPLELINTS (Algorithm 3) with an appropriate choice of $\lambda$ after $T$ rounds is upper bounded by*

$$\mathfrak{R}(\hat{\pi}) \leq \mathfrak{c}\, d^{3/2}\sqrt{\log(T/\delta)/T}\,.$$

The simple regret bound of SIMPLELINTS exhibits a dependency on $d$ that matches that of TS in the cumulative regret setting. Note that unlike the other algorithms, one can also set $\lambda$ without the knowledge of $S$ with incurring a minimal extra cost in the simple regret that depends on $\|\theta_*\|$.

**THATS: Try Hard Thompson Sampling – Logistic Case** The pseudocode of THATS is shown in Algorithm 4. Similar to SIMPLELINTS, THATS samples a parameter vector from a Gaussian centered at zero. However, in this case, motivated by the construction of MULOG, the covariance of the Gaussian is $L_t^{-1}$, where $L_t$ is the increasing sequence which is constructed in an analogous way to what was seen in MULOG:

$$L_t = \lambda I + \sum_{i=1}^{t-1} \dot{\mu}(\phi(S_i, A_i)^\top \theta_i')\phi(S_i, A_i)\phi(S_i, A_i)^\top \text{ where } \theta_i' = \underset{\theta \in \mathcal{E}_i(\delta, \bar{\theta}_i)}{\arg\min} \dot{\mu}(\phi(S_i, A_i)^\top \theta). \quad (14)$$

A slight difference to MULOG is that the confidence sets used here will be centered at $\bar{\theta}_t$, the MLE over the $S$-ball. As a result, we will slightly increase the radii of the confidence sets:

$$\mathcal{E}_t(\delta, \theta_\circ) = \{\theta \in \mathbb{B}_d(S) : \mathcal{L}_t^\lambda(\theta) - \mathcal{L}_t^\lambda(\theta_\circ) \leq 2\beta_t(\delta)^2\} \text{ and } \bar{\theta}_t = \underset{\theta \in \mathbb{B}_d(S)}{\arg\min} \mathcal{L}_t^\lambda(\theta)\,. \quad (15)$$

These confidence sets are also convex. The reason for using these confidence sets is so that in the action selection, we can avoid the step of searching for a parameter vector that is in the intersection of our confidence set *and the $S$-ball*. In particular, for action selection we will simply use

$$A_t = \underset{a \in \mathcal{A}(S_t)}{\arg\max} \dot{\mu}(\phi(S_t, a)^\top \bar{\theta}_t)\left|\phi(S_t, a)^\top \tilde{\theta}_t\right|\,, \quad (16)$$

which combines ideas from SIMPLELINTS and MULOG. Note that this is a step where we save on computation. We also save on computation in Eq. (14) by doing the optimization constrained on $\mathcal{E}_i$, instead of using intersections of these sets. When the algorithm returns a policy, we pick any vector in the intersection of all of the confidence sets,

$$\mathcal{V}_{T+1} = \cap_{i=1}^T \mathcal{E}_i(\delta, \bar{\theta}_i)\,, \quad (17)$$

and return a policy that is greedy with respect to the estimated mean rewards. The purpose of intersecting the confidence sets is to guarantee that the decreasing uncertainty argument goes through.

We will find it useful to relate the new confidence sets with the ones used previously (see Eq. (11)):

**Lemma 5.** *Let $\delta \in [0, 1)$. With probability at least $1 - \delta$, $\bar{\theta}_t \in \mathcal{C}_t(\delta, \hat{\theta}_t) \cap \mathbb{B}_d(S)$. Furthermore, with probability at least $1 - \delta$, $\mathcal{C}_t(\delta, \hat{\theta}_t) \cap \mathbb{B}_d(S) \subseteq \mathcal{E}_t(\delta, \bar{\theta}_t)$.*

We also have a results similar to Lemma 4 in the logistic case to quantify the discrepancy between the uncertainty of the action selected by THATS and the max-uncertainty action $A_t^{\mathrm{MU}}$:

**Lemma 6.** *Let $t \geq 1$, $A_t^{\mathrm{MU}} = \arg\max_{a \in \mathcal{A}(S_t)} \max_{\theta \in \mathcal{V}_t} \dot{\mu}(\phi(S_t, a)^\top \theta) \|\phi(S_t, a)\|_{L_t^{-1}}$. Then, it holds almost surely that for $I(\cdot)$ defined in Lemma 4,*

$$\min_{\theta \in \mathcal{E}_t} U(S_t, A_t^{\mathrm{MU}}, \theta, L_t) \cdot I(S_t, A_t^{\mathrm{MU}}, L_t) \leq \mathbb{E}\left[U(S_t, A_t, \bar{\theta}_t, L_t) \Big| \mathcal{F}_{t-1}, S_t\right].$$

Using this in the analysis gives the following result:

**Theorem 4.** *Under Assumptions 1 and 2, there exists some universal constant $\mathfrak{c} > 0$ such that the following holds: Let $\delta \in [0, 1)$, $T \geq 1$ be arbitrary. Then, with probability at least $1 - \delta$, it holds that the simple regret of the policy $\hat{\pi}$ computed by THATS (Algorithm 4) with an appropriate choice of $\lambda$ after $T$ rounds is upper bounded by*

$$\mathfrak{R}(\hat{\pi}) \leq \mathfrak{C}d^{3/2}\sqrt{\log(T/\delta)/T} + \mathrm{poly}\left(\kappa, d, \log(T), \log(1/\delta)\right)/T.$$

## 4 NUMERICAL EXPERIMENT

This section presents numerical results for our proposed randomized algorithms, which are computationally tractable, across both linear and logistic models. We outline the experimental design here; full setup details and results are available in Appendix G.

**Linear Case** To demonstrate the benefits of strategic exploration, we use an environment adversarial to uniform exploration (UE), where all suboptimal arms are orthogonal to the optimal one. In this setting, UE wastes samples on suboptimal arms, whereas SIMPLELINTS quickly eliminates them. We include TS for cumulative regret (CumuLinTS) as a baseline to show that its need to balance exploration and exploitation results in slower convergence than SIMPLELINTS. This comparison illustrates the importance of tailoring algorithms specifically for the simple regret objective.

**Logistic Case** This experiment shows that an MLE extension of SIMPLELINTS (replacing least squares with logistic regression) performs poorly compared to our more sophisticated method, THATS, in the logistic setting. We designed an environment specifically to highlight this weakness. The arm set is $\{-e_i\}_{i=1}^{d-1} \cup \{0.3 \cdot e_d, -0.3 \cdot e_d\}$ and we set $\theta_* = [M, M, \ldots, 1]$. The optimal and second optimal arm are $\pm 0.3 \cdot e_d$ respectively. This construction makes the rewards from $\pm 0.3 \cdot e_d$ very **noisy** (mean reward is close to 0.5), while rewards from the other arms are **certain** (and equally bad). A good algorithm should pull the noisy arms more often to get better estimates. THATS successfully does this while the SIMPLELINTS extension fails because it incentivizes simply growing the magnitude of its design matrix $V_t$, causing it to neglect the crucial noisy arms.

## 5 CONCLUSIONS

In this paper we considered stochastic contextual linear and logistic bandits where the objective is to keep simple regret small. We proposed and analyzed a deterministic and a randomized algorithm for both settings. All algorithms are constructing data by choosing actions in each round that give the largest estimated uncertainty given the past information; an idea that has been explored in some related setting in previous works. The main novelty of our approach is in the new analysis paradigm of randomized algorithms and logistic case where uncertainty estimates need to use model parameters due to the nonlinearity of the reward function model. With our novel constructions, all algorithms are efficient and the bounds are essentially tight. One interesting question that is left open is whether the computational cost of our randomized algorithms can be further reduced. In particular, the algorithms still require the construction of confidence sets and solving a few linear optimization problems over these sets in each round. This step can still be quite expensive. Another interesting question is to reduce compute cost for large, but structured action sets.

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

## A   MORE RELATED WORK

Linear contextual bandits can be viewed as a special case of linear MDPs (Yang & Wang, 2019; Jin et al., 2020), linear mixture MDPs (Modi et al., 2020; Ayoub et al., 2020) and linear contextual MDPs (Deng et al., 2024) with horizon $H = 1$, and for linear mixture MDPs the number of base models $K = 1$. We report their results by converting their sample complexity bounds into simple regret bounds. Since the hardness in reinforcement learning stems from the unknown transition kernel, most works assume that the rewards are known and deterministic (Antos et al., 2007; Chen & Jiang, 2019; Ayoub et al., 2024). However, these results can be extended to the case of unknown stochastic rewards without changing the their qualitative behavior. In linear MDPs, Jin et al. (2020) obtained a simple regret of $\tilde{\mathcal{O}}(d^{3/2}H^2/\sqrt{T})$, while later Wagenmaker & Jamieson (2022) improve this to $\tilde{\mathcal{O}}(dH^3/\sqrt{T})$. In linear mixture MDPs, Modi et al. (2020) state a simple regret bound of $\tilde{\mathcal{O}}(d^{3/2}KH/\sqrt{T})$; Chen et al. (2022) obtain an improved simple regret bound of $\tilde{\mathcal{O}}(d/\sqrt{T})$ (in their setting $d$ is the total number of parameters, while in the setting of Modi et al. (2020), $dK$ is the total number of parameters). In linear contextual MDPs, Deng et al. (2024) report a simple regret bound of $\tilde{\mathcal{O}}(d^{3/2}H^{5/2}/\sqrt{T})$.

## B   REGRET ANALYSIS OF MULIN (THEOREM 1)

In this section, we will analyze the simple regret of MULIN (Algorithm 1).

## B.1 CONFIDENCE SET

In the analysis, we will use the confidence set from Abbasi-Yadkori et al. (2011)

**Lemma 7** (Theorem 2 of Abbasi-Yadkori et al. (2011)). *Let $\delta \in (0, 1)$. Then with probability at least $1 - \delta$, it holds that for all $t \geq 1$,*

$$\|\hat{\theta}_t - \theta_*\|_{V_t} \leq \sqrt{\lambda}\|\theta_*\| + \sqrt{2\log(1/\delta) + d\log\left(1 + \frac{t}{d\lambda}\right)} := \tau_t(\delta). \tag{18}$$

## B.2 DECREASING UNCERTAINTY

In this section we restate our result on the decreasing uncertainty mentioned in Section 3.1, specifically, Lemma 1. This lemma allows us to relate the analysis techniques in the cumulative regret setting Abbasi-Yadkori et al. (2011) to the analysis in the simple regret setting.

**Lemma 1.** *[Decreasing Uncertainty Lemma, Lemma 6 of Zanette et al. (2021)] For every $t \geq 1$, it holds that*

$$\int \max_{a \in \mathcal{A}(s)} \|\phi(s, a)\|_{V_{t+1}^{-1}} \nu(ds) \leq \int \max_{a \in \mathcal{A}(s)} \|\phi(s, a)\|_{V_t^{-1}} \nu(ds).$$

## B.3 PROOF OF THE REGRET BOUND THEOREM 1

In this section, we first state the formal version of Theorem 1 where all the constants and dependencies are detailed then proofs are provided.

**Theorem 5** (MULIN Simple Regret Bound). *Under Assumptions 1 and 2, there exists some universal constant $\mathfrak{c} > 0$ such that the following holds: Let $\delta \in [0, 1)$, $T \geq 1$ be arbitrary. Then, with probability at least $1 - \delta$, it holds that the simple regret of the policy $\hat{\pi}$ computed by MULIN (Algorithm 1) with an appropriate choice of $\lambda$, after $T$ rounds is upper bounded by*

$$\mathfrak{R}(\hat{\pi}) \leq \frac{4\tau_{T+1}(\delta)\sqrt{d\log((d\lambda_T + T)/(d\lambda_T))}}{\sqrt{T}} + \frac{4\tau_{T+1}(\delta)16\log(\log(2T/\delta))}{T},$$

*where $\tau_{T+1}(\delta) = \tilde{\mathcal{O}}(\sqrt{d + \log(1/\delta)} + \|\theta_*\|)$ whose full expression can be found in Eq. (18).*

A related objective to $\mathfrak{R}$, which is explained in the lemma that follows, when $\hat{\theta}_{T+1}$ is used, is the expected maximum prediction error

$$D_{\text{Lin}}(\hat{\theta}_{T+1}) = \int \max_{a \in \mathcal{A}(s)} |\phi(s, a)^\top (\theta^* - \hat{\theta}_{T+1})| \nu(ds).$$

**Lemma 8.** *For a vector $\bar{\theta} \in \mathbb{R}^d$, let $\bar{\pi}$ be greedy w.r.t $\bar{\theta}$, i.e.,*

$$\bar{\pi}(s) = \arg\max_{a \in \mathcal{A}(s)} \phi(s, a)^\top \bar{\theta},$$

*it then follows that*

$$\mathfrak{R}(\bar{\pi}) \leq 2D_{\text{Lin}}(\bar{\theta}).$$

*Proof.* The proof is a simple application of triangle inequality. For $s \in \mathcal{S}$, let $\Delta(s) := \max_{a \in \mathcal{A}} |\phi(s, a)^\top (\theta^* - \hat{\theta}_T)|$, we then add and subtract the same term below

$$\begin{aligned}
&\phi(s, \pi^*(s))^\top \theta^* - \phi(s, \bar{\pi}(s))^\top \theta^* \\
&= \phi(s, \pi^*(s))^\top \theta^* - \phi(s, \pi^*(s))^\top \bar{\theta} + \phi(s, \pi^*(s))^\top \bar{\theta} - \phi(s, \bar{\pi}(s))^\top \theta^* \\
&\leq \phi(s, \pi^*(s))^\top \theta^* - \phi(s, \pi^*(s))^\top \bar{\theta} + \phi(s, \bar{\pi}(s))^\top \bar{\theta} - \phi(s, \bar{\pi}(s))^\top \theta^* &\text{(Defn. of } \bar{\pi}_T) \\
&\leq \left|\phi(s, \pi^*(s))^\top \theta^* - \phi(s, \pi^*(s))^\top \bar{\theta}\right| + \left|\phi(s, \bar{\pi}(s))^\top \bar{\theta} - \phi(s, \bar{\pi}(s))^\top \theta^*\right| \\
&\leq 2\Delta(s).
\end{aligned}$$

Finally integrating on both sides on $s$ give the desired result. $\square$

Hence we focus on bounding $D_{\text{Lin}}(\hat{\theta}_{T+1})$. By Cauchy-Schwarz inequality, we have

$$D_{\text{Lin}}(\hat{\theta}_{T+1}) = \int \max_{a \in \mathcal{A}(s)} |\phi(s,a)^\top (\theta^* - \hat{\theta}_{T+1})| \nu(ds)$$

$$\leq \int \max_{a \in \mathcal{A}(s)} \|\phi(s,a)\|_{V_{T+1}^{-1}} \|\theta_* - \hat{\theta}_{T+1}\|_{V_{T+1}} \nu(ds)$$

$$\leq \tau_{T+1}(\delta) \int \max_{a \in \mathcal{A}(s)} \|\phi(s,a)\|_{V_{T+1}^{-1}} \nu(ds),$$

where in the last line we used Lemma 7. Then by Lemma 1 we have that for all $1 \leq t \leq T$,

$$\int \max_{a \in \mathcal{A}(s)} \|\phi(s,a)\|_{V_{T+1}^{-1}} \nu(ds) \leq \int \max_{a \in \mathcal{A}(s)} \|\phi(s,a)\|_{V_t^{-1}} \nu(ds).$$

Hence,

$$\int \max_{a \in \mathcal{A}(s)} \|\phi(s,a)\|_{V_{T+1}^{-1}} \nu(ds) \leq \frac{1}{T} \sum_{t=1}^{T} \int \max_{a \in \mathcal{A}(s)} \|\phi(s,a)\|_{V_t^{-1}} \nu(ds).$$

We can then plug this into the bound for $D_{\text{Lin}}(\hat{\theta}_{T+1})$ to get

$$\tau_{T+1}(\delta) \int \max_{a \in \mathcal{A}(s)} \|\phi(s,a)\|_{V_{T+1}^{-1}} \nu(ds) \leq \tau_{T+1}(\delta) \frac{1}{T} \sum_{t=1}^{T} \int \max_{a \in \mathcal{A}(s)} \|\phi(s,a)\|_{V_t^{-1}} \nu(ds)$$

$$= \tau_{T+1}(\delta) \frac{1}{T} \sum_{t=1}^{T} \mathbb{E}\left[ \max_{a \in \mathcal{A}(s)} \|\phi(S_t,a)\|_{V_t^{-1}} \Big| \mathcal{F}_{t-1} \right]. \quad (19)$$

Now before we can call Elliptical Potential Lemma (Lemma 19), there is still one more step to be done, that is, bounding conditional expectations with realizations. To be more specific, note that for a stochastic process $\{Y_t\}_{t \geq 1}$ adapted to a filtration $\{\mathbf{F}_t\}_{t \geq 1}$, even if $\sum_{t=1}^{T} Y_t \leq c_T$ for some constants $\{c_t\}_{t \geq 1}$ with probability 1, it does not necessarily follow that $\mathbb{E}[\sum_{t=1}^{T} Y_t | \mathbf{F}_t] \leq c_T$ with probability 1 [3]. The good news is similar inequality holds with some blow up of $\{c_t\}_{t \geq 1}$, which is shown in Corollary 4. From Corollary 4,

$$\frac{1}{T} \sum_{t=1}^{T} \mathbb{E}\left[ \max_{a \in \mathcal{A}(s)} \|\phi(s,a)\|_{V_t^{-1}} \Big| \mathcal{F}_{t-1} \right] \leq \frac{1}{T} \left( 4\sqrt{\log(\log(2T/\delta))} + \sqrt{\sum_{t=1}^{T} \|\phi(S_t,A_t)\|_{V_t^{-1}}} \right)^2$$

$$\leq \frac{2}{T} \left( 16\log(\log(2T/\delta)) + \sum_{t=1}^{T} \|\phi(S_t,A_t)\|_{V_t^{-1}} \right),$$

where the last line uses $(a+b)^2 \leq 2a^2 + 2b^2$. Plug the above result back in Eq. (19),

$$D_{\text{Lin}}(\hat{\theta}_{T+1}) \leq \tau_{T+1}(\delta) \frac{1}{T} \sum_{t=1}^{T} \mathbb{E}\left[ \max_{a \in \mathcal{A}(S_t)} \|\phi(S_t,a)\|_{V_t^{-1}} \Big| \mathcal{F}_{t-1} \right] \quad (20)$$

$$\leq \frac{2\tau_{T+1}(\delta)}{T} \left( 16\log(\log(2T/\delta)) + \sum_{t=1}^{T} \|\phi(S_t,A_t)\|_{V_t^{-1}} \right) \quad (21)$$

$$\leq \frac{2\tau_{T+1}(\delta)}{T} \left( 16\log(\log(2T/\delta)) + \sqrt{T} \sqrt{\sum_{t=1}^{T} \|\phi(S_t,A_t)\|_{V_t^{-1}}^2} \right) \quad (22)$$

$$\leq \frac{2\tau_{T+1}(\delta)}{T} \left( 16\log(\log(2T/\delta)) + \sqrt{T} \sqrt{d\log((d\lambda_T + T)/(d\lambda_T))} \right), \quad (23)$$

where in the third line we used Cauchy-Schwarz inequality and the last line follows from elliptical potential lemma (Lemma 19). Finally, chaining the above bound with Lemma 8 gives the desired result.

---

[3]See this nice example.

## C  ANALYSIS OF SIMPLELINTS (ALGORITHM 3)

In this section we will analyze the simple regret of SIMPLELINTS (Algorithm 3).

### C.1  ANALYSIS ON THE EXPLORATION DONE BY SIMPLELINTS AND MULIN

The analysis of SIMPLELINTS is highly related to that of MULIN algorithm. Hence in order to identify the actions that gives the maximum uncertainty, we define it to be

$$A_t^{\mathrm{MU}} = \underset{a \in \mathcal{A}(S_t)}{\arg\max} \|\phi(S_t, a)\|_{V_t^{-1}}.$$

We now present the lemma that allows us to relate the analysis of MULIN to SIMPLELINTS.

**Lemma 4.** *Fix $t \geq 1$. Then, almost surely,*

$$\|\phi(S_t, A_t^{\mathrm{MU}})\|_{V_t^{-1}} \cdot I(S_t, A_t^{\mathrm{MU}}, V_t) \leq \mathbb{E}\left[\|\phi(S_t, A_t)\|_{V_t^{-1}} \Big| \mathcal{F}_{t-1}, S_t\right],$$

*where for $(s, a) \in \mathcal{Z}$ and $V \succeq 0$, we let $I(s, a, V) = \int_{\mathbb{S}^{d-1}} \left|\left\langle x, \frac{V^{-1/2}\phi(s,a)}{\|V^{-1/2}\phi(s,a)\|}\right\rangle\right| dx.$*

*Proof.* We start by rewriting the right hand side of the inequality

$$\mathbb{E}\left[\|\phi(S_t, A_t)\|_{V_t^{-1}} \Big| \mathcal{F}_{t-1}, S_t\right]$$

$$= \mathbb{E}\left[\max_{x \in \mathbb{S}^{d-1}} \left\langle V_t^{-1/2}x, \phi(S_t, A_t)\right\rangle \Big| \mathcal{F}_{t-1}, S_t\right]$$

$$\geq \mathbb{E}\left[\left|\left\langle \frac{V_t^{-1/2} \cdot V_t^{1/2}\tilde{\theta}_t}{\|V_t^{1/2}\tilde{\theta}_t\|_2}, \phi(S_t, A_t)\right\rangle\right| \Big| \mathcal{F}_{t-1}, S_t\right]$$

$$\geq \mathbb{E}\left[\left|\left\langle \frac{\tilde{\theta}_t}{\|V_t^{1/2}\tilde{\theta}_t\|_2}, \phi(S_t, A_t^{\mathrm{MU}})\right\rangle\right| \Big| \mathcal{F}_{t-1}, S_t\right],$$

where in the last line we used the definition of $A_t$. Since $\tilde{\theta}_t \sim \mathcal{N}(0, V_t^{-1})$, we can rewrite it as $\tilde{\theta}_t = V_t^{-1/2}M_t$ for $M_t \sim \mathcal{N}(0, I)$ given the past and the current context. Then plug it back in the above expression,

$$\mathbb{E}\left[\left|\left\langle \frac{\tilde{\theta}_t}{\|V_t^{1/2}\tilde{\theta}_t\|_2}, \phi(S_t, A_t^{\mathrm{MU}})\right\rangle\right| \Big| \mathcal{F}_{t-1}, S_t\right]$$

$$= \mathbb{E}\left[\left|\left\langle \frac{V_t^{-1/2}M_t}{\|M_t\|_2}, \phi(S_t, A_t^{\mathrm{MU}})\right\rangle\right| \Big| \mathcal{F}_{t-1}, S_t\right]$$

$$= \mathbb{E}\left[\left|\left\langle \frac{M_t}{\|M_t\|_2}, V_t^{-1/2}\phi(S_t, A_t^{\mathrm{MU}})\right\rangle\right| \Big| \mathcal{F}_{t-1}, S_t\right]$$

$$= \|\phi(S_t, A_t^{\mathrm{MU}})\|_{V_t^{-1}} \mathbb{E}\left[\left|\left\langle \frac{M_t}{\|M_t\|_2}, \frac{V_t^{-1/2}\phi(S_t, A_t^{\mathrm{MU}})}{\|\phi(S_t, A_t^{\mathrm{MU}})\|_{V_t^{-1}}}\right\rangle\right| \Big| \mathcal{F}_{t-1}, S_t\right]$$

$$= \|\phi(S_t, A_t^{\mathrm{MU}})\|_{V_t^{-1}} I(S_t, A_t^{\mathrm{MU}}, L_t),$$

where the third line used $V_t^{-1/2}$ is symmetric (as $V_t$ is positive definite); the fourth line follows by $A_t^{\mathrm{MU}}$ is $\mathcal{F}_{t-1}$-measurable and Proposition 1 and the last line follows by the definition of $I(S_t, A_t^{\mathrm{MU}}, L_t)$. □

**Corollary 1.** *For all $t \geq 1$, it holds almost surely that*

$$\mathbb{E}\left[\|\phi(S_t, A_t^{\mathrm{MU}})\|_{V_t^{-1}} \Big| \mathcal{F}_{t-1}\right] \leq \sqrt{\frac{\pi d}{2}} \mathbb{E}\left[\|\phi(S_t, A_t)\|_{V_t^{-1}} \Big| \mathcal{F}_{t-1}\right].$$

*Proof.* The proof follows by dividing both sides of the inequality showed in Lemma 4 by $I(S_t, A_t^{\mathrm{MU}}, V_t)$ and then showing that $I(S_t, A_t^{\mathrm{MU}}, V_t)$ is lower bounded by $\sqrt{\frac{2}{\pi d}}$. The latter follows using Proposition 2 and Proposition 3. Together we get,

$$\|\phi(S_t, A_t^{\mathrm{MU}})\|_{V_t^{-1}} \leq \sqrt{\frac{\pi d}{2}} \mathbb{E}\left[\|\phi(S_t, A_t)\|_{V_t^{-1}}\Big|\mathcal{F}_{t-1}, S_t\right],$$

from which the tower rule gives the desired result. $\qquad\square$

## C.2 ANALYSIS OF SIMPLELINTS

In this section, we first state formally our regret bound on SIMPLELINTS where detailed constant and polynomial dependency is presented. After that we give proof on it.

**Theorem 6** (SIMPLELINTS Simple Regret Bound). *Under Assumptions 1 and 2, there exists some universal constant $\mathfrak{c} > 0$ such that the following holds: Let $\delta \in [0, 1)$, $T \geq 1$ be arbitrary. Then, with probability at least $1 - \delta$, it holds that the simple regret of the policy $\hat{\pi}$ computed by SIMPLELINTS (Algorithm 3) with an appropriate choice of $\lambda$, after $T$ rounds is upper bounded by*

$$\mathfrak{R}(\hat{\pi}) \leq \frac{4\tau_{T+1}(\delta)d\sqrt{\log((d\lambda_T + T)/(d\lambda_T))}}{\sqrt{T}} + \frac{4\tau_{T+1}(\delta)16\log(\log(2T/\delta))}{T},$$

*where $\tau_{T+1}(\delta) = \tilde{\mathcal{O}}(\sqrt{d + \log(1/\delta)} + \|\theta_*\|)$ whose full expression can be found in Eq. (18).*

All we need to do is to plug Lemma 4 into the analysis of MULIN. Lemma 8 still holds so we start by doing the same analysis as in Appendix B.3 and then plug in the results of Lemma 4 to get the final bound. For clarity we copy the analysis in Appendix B.3; for readers who are familiar with the analysis of MULIN, they can skip to the end of this section where we highlight the step that is unique to SIMPLELINTS to be red. By Cauchy-Schwarz inequality, we have

$$D_{\mathrm{Lin}}(\hat{\theta}_{T+1}) = \int \max_{a \in \mathcal{A}(s)} |\phi(s, a)^\top (\theta^* - \hat{\theta}_{T+1})|\nu(ds)$$

$$\leq \int \max_{a \in \mathcal{A}(s)} \|\phi(s, a)\|_{V_{T+1}^{-1}} \|\theta_* - \hat{\theta}_{T+1}\|_{V_{T+1}} \nu(ds)$$

$$\leq \tau_{T+1}(\delta) \int \max_{a \in \mathcal{A}(s)} \|\phi(s, a)\|_{V_{T+1}^{-1}} \nu(ds),$$

where in the last line we used Lemma 7. Then by Lemma 1 we have that for all $1 \leq t \leq T$,

$$\int \max_{a \in \mathcal{A}(s)} \|\phi(s, a)\|_{V_{T+1}^{-1}} \nu(ds) \leq \int \max_{a \in \mathcal{A}(s)} \|\phi(s, a)\|_{V_t^{-1}} \nu(ds).$$

Hence,

$$\int \max_{a \in \mathcal{A}(s)} \|\phi(s, a)\|_{V_{T+1}^{-1}} \nu(ds) \leq \frac{1}{T} \sum_{t=1}^{T} \int \max_{a \in \mathcal{A}(s)} \|\phi(s, a)\|_{V_t^{-1}} \nu(ds).$$

We can then plug this into the bound for $D_{\mathrm{Lin}}(\hat{\theta}_{T+1})$ to get

$$\tau_{T+1}(\delta) \int \max_{a \in \mathcal{A}(s)} \|\phi(s, a)\|_{V_{T+1}^{-1}} \nu(ds) \leq \tau_{T+1}(\delta)\frac{1}{T} \sum_{t=1}^{T} \int \max_{a \in \mathcal{A}(s)} \|\phi(s, a)\|_{V_t^{-1}} \nu(ds)$$

$$= \tau_{T+1}(\delta)\frac{1}{T} \sum_{t=1}^{T} \mathbb{E}\left[\max_{a \in \mathcal{A}(S_t)} \|\phi(S_t, a)\|_{V_t^{-1}}\Big|\mathcal{F}_{t-1}\right].$$

From Corollary 4,

$$\frac{1}{T} \sum_{t=1}^{T} \mathbb{E}\left[\max_{a \in \mathcal{A}(s)} \|\phi(s, a)\|_{V_t^{-1}}\Big|\mathcal{F}_{t-1}\right] \leq \frac{1}{T}\left(4\sqrt{\log(\log(2T/\delta))} + \sqrt{\sum_{t=1}^{T} \|\phi(S_t, A_t)\|_{V_t^{-1}}}\right)^2$$

$$\leq \frac{2}{T}\left(16\log(\log(2T/\delta)) + \sum_{t=1}^{T} \|\phi(S_t, A_t)\|_{V_t^{-1}}\right),$$

where the last line uses $(a+b)^2 \leq 2a^2 + 2b^2$. Plug the above result back in Eq. (19),

$$D_{\mathrm{Lin}}(\hat{\theta}_{T+1}) \leq \tau_{T+1}(\delta) \frac{1}{T} \sum_{t=1}^{T} \mathbb{E}\left[ \max_{a \in \mathcal{A}(s)} \|\phi(s,a)\|_{V_t^{-1}} \Big| \mathcal{F}_{t-1} \right]$$

$$= \tau_{T+1}(\delta) \frac{1}{T} \sum_{t=1}^{T} \mathbb{E}\left[ \|\phi(S_t, A_t^{\mathrm{MU}})\|_{V_t^{-1}} \Big| \mathcal{F}_{t-1} \right]$$

$$\leq \tau_{T+1}(\delta) \frac{1}{T} \sqrt{d} \sum_{t=1}^{T} \mathbb{E}\left[ \|\phi(S_t, A_t)\|_{V_t^{-1}} \Big| \mathcal{F}_{t-1} \right]$$

$$\leq \frac{2\tau_{T+1}(\delta)\sqrt{d}}{T} \left( 16 \log(\log(2T/\delta)) + \sum_{t=1}^{T} \|\phi(S_t, A_t)\|_{V_t^{-1}} \right)$$

$$\leq \frac{2\tau_{T+1}(\delta)\sqrt{d}}{T} \left( 16 \log(\log(2T/\delta)) + \sqrt{T} \sqrt{\sum_{t=1}^{T} \|\phi(S_t, A_t)\|_{V_t^{-1}}^2} \right)$$

$$\leq \frac{2\tau_{T+1}(\delta)\sqrt{d}}{T} \left( 16 \log(\log(2T/\delta)) + \sqrt{T} \sqrt{d \log((d\lambda_T + T)/(d\lambda_T))} \right).$$

Finally, chaining the above bound with Lemma 8 gives the desired result.

## D   REGRET ANALYSIS OF MULog (THEOREM 2)

The sigmoid function is known to be (generalized) self-concordant (Bach, 2010; Faury et al., 2020; Liu et al., 2024), to be more specific,

$$|\ddot{\mu}(z)| \leq \dot{\mu}(z) \quad \text{for all } z \in \mathbb{R}. \tag{24}$$

The logistic function is also $1/4$-Lipschitz, i.e.,

$$\dot{\mu} \leq 1/4 \tag{25}$$

which can be seen from the decomposiion of $\dot{\mu} = \mu(1-\mu)$ and $\mu \in [0,1]^{\mathbb{R}}$.

We also consider the Hessian of the loss, which takes the form of

$$\nabla^2 \mathcal{L}_t^{\lambda}(\theta) =: H_t(\theta) = \sum_{s=1}^{t-1} \dot{\mu}(\phi(S_s, A_s)^{\top}\theta)\phi(S_s, A_s)\phi(S_s, A_s)^{\top} + \lambda I$$

In the analysis, we also consider the follwing matrices that are closely related to $H_t(\theta; \{A_s\}_{s=1}^{t-1})$.

$$G_t(\theta_1, \theta_2) = \lambda I + \sum_{s=1}^{t-1} \alpha(\phi(S_s, A_s), \theta_1, \theta_2)\phi(S_s, A_s)\phi(S_s, A_s)^{\top}$$

$$\tilde{G}_t(\theta_1, \theta_2) = \lambda I + \sum_{s=1}^{t-1} \tilde{\alpha}(\phi(S_s, A_s), \theta_1, \theta_2)\phi(S_s, A_s)\phi(S_s, a_s)^{\top},$$

where for $x, \theta_1, \theta_2 \in \mathbb{R}^d$,

$$\alpha(x, \theta_1, \theta_2) = \int_{v=0}^{1} \dot{\mu}(x^{\top}\theta_1 + vx^{\top}(\theta_2 - \theta_1))dv$$

$$\tilde{\alpha}(x, \theta_1, \theta_2) = \int_{v=0}^{1} (1-v)\dot{\mu}(x^{\top}\theta_1 + vx^{\top}(\theta_2 - \theta_1))dv.$$

### D.1   SELF-CONCORDANCE CONTROL

Self-concordance is a property of the logistic function $\mu(\cdot)$ that allows us to utilize the curvature information. Here are the lemmas borrowed from Abeille et al. (2021); Faury et al. (2020) that we will use in the analysis.

**Lemma 9** (Lemma 9 of Abeille et al. (2021))**.** *For all $z_1, z_2 \in \mathbb{R}$, it follows that*

$$\dot{\mu}(z_2) \exp(-|z_2 - z_1|) \leq \dot{\mu}(z_1) \leq \dot{\mu}(z_2) \exp(|z_2 - z_1|).$$

**Lemma 10** (First order self-concordance control, Lemma 9 of Faury et al. (2020))**.** *For all $z_1, z_2 \in \mathbb{R}$, it follows that*

$$\dot{\mu}(z_1)\frac{1}{1 + |z_1 - z_2|} \leq \int_{v=0}^{1} \dot{\mu}(z_1 + v(z_2 - z_1))dv \leq \dot{\mu}(z_1)\frac{\exp(|z_1 - z_2| - 1)}{|z_1 - z_2|}.$$

**Lemma 11** (Second order self-concordance control, Lemma 8 of Abeille et al. (2021))**.** *For all $z_1, z_2 \in \mathbb{R}$,*

$$\int_{v=0}^{1} (1 - v)\ddot{\mu}(z_1 + v(z_2 - z_1))dv \geq \frac{\dot{\mu}(z_1)}{2 + |z_1 - z_2|}.$$

**Lemma 12** (Eqs.(7,8) of Abeille et al. (2021))**.** *Let $\theta_1, \theta_2 \in \mathbb{R}^d$. For $t \geq 1$*

$$u_t = \begin{cases} 0, & \text{if } t = 1 \\ \max_{x \in \{\phi(S_s, A_s)\}_{s=1}^{t-1}} |x^\top (\theta_1 - \theta_2)|, & \text{if } t \geq 2 \end{cases}$$

*Then it follows that*

$$G_t(\theta_1, \theta_2) \succeq (1 + 2u)^{-1} H_t(\theta) \text{ for } \theta \in \{\theta_1, \theta_2\}$$
$$\tilde{G}_t(\theta_1, \theta_2) \succeq (2 + 2u)^{-1} H_t(\theta_1)$$

### D.2 RESULTS ON CONFIDENCE SET

In this section, we state the lemmas on the confidence set that we will use in the analysis.

The following confidence set from Faury et al. (2020) is also used in our analysis:

**Lemma 13** (Lemma 1 of Faury et al. (2020))**.** *Let $\delta \in (0, 1]$. Under assumptions 1, 2, with probability at least $1 - \delta$,*

$$\forall t \geq 1, \quad \|g_t(\hat{\theta}_t) - g_t(\theta_*)\|_{H_t^{-1}(\theta_*)} \leq \rho_t(\delta),$$

*where*

$$\lambda_T = 1 \vee \frac{2d}{S} \log \left( e\sqrt{1 + \frac{T}{4d}} \vee 1/\delta \right), \tag{26}$$

$$\rho_t(\delta) = \left(\frac{1}{2} + S\right) \sqrt{\lambda_T} + \frac{4d}{\sqrt{\lambda_T}} \log \left( e\sqrt{1 + \frac{t}{4d}} \vee 1/\delta \right). \tag{27}$$

**Lemma 2.** *(Lemma 1 of Abeille et al. (2021)) Let $\delta \in [0, 1)$. It follows that $\mathbb{P}\big(\forall t \geq 1, \theta_* \in \mathcal{C}_t(\delta, \hat{\theta}_t)\big) \geq 1 - \delta$.*

Recall that

$$\mathcal{C}_t(\delta, \theta_\circ) = \left\{ \theta : \mathcal{L}_t^\lambda(\theta) - \mathcal{L}_t^\lambda(\theta_\circ) \leq \beta_t(\delta) \right\},$$

where according to Abeille et al. (2021), $\beta_t(\delta)$ is set to be

$$\beta_t(\delta) = \rho_t(\delta) + \frac{\rho_t(\delta)^2}{\sqrt{\lambda_T}}. \tag{28}$$

Combining the above two lemmas (Lemmas 2 and 13) together, we have the following lemma.

**Lemma 14.** *Under Assumption 1,2, for all $\delta \in [0, 1)$, with probability at least $1 - 2\delta$, for all $t \geq 1$ and $\theta \in \mathcal{C}_t(\hat{\theta}_t, \delta) \cap \mathbb{B}_d(S)$, we have that*

$$\|\theta - \theta_*\|_{H_t(\theta_*)} \leq (4 + 4S)\rho_t(\delta) + \sqrt{(8S + 8)}\beta_t(\delta) =: \gamma_t(\delta),$$

*where $\beta_t(\delta)$ is defined in Eq. (28).*

*Proof.* We start from Taylor expansion. For all $\theta' \in \mathbb{R}^d$, we have that

$$\mathcal{L}_t^\lambda(\theta') = \mathcal{L}_t^\lambda(\theta_*) + \nabla\mathcal{L}_t^\lambda(\theta_*)^\top(\theta' - \theta_*) + \frac{1}{2}\|\theta' - \theta_*\|_{\bar{G}_t(\theta_*,\theta')}^2.$$

Let $\theta \in \mathcal{C}_t(\hat{\theta}_t, \delta)(\delta) \cap \mathbb{B}_d(S)$. Rearrange the terms, apply absolute value and plug in $\theta$,

$$|\mathcal{L}_t^\lambda(\theta) - \mathcal{L}_t^\lambda(\theta_*) - \nabla\mathcal{L}_t^\lambda(\theta_*)(\theta - \theta_*)| = \frac{1}{2}\|\theta - \theta_*\|_{\tilde{G}_t(\theta_*,\theta)}^2 \geq \frac{1}{2(2+2S)}\|\theta - \theta_*\|_{H_t(\theta_*)}^2,$$

where the last inequality follows from Lemma 12 and $\theta, \theta_* \in \mathbb{B}_d(S)$. It remains to upper bound the left most side of the above equation. By triangle inequality we can split it into two terms and we bound them separately.

$$|\mathcal{L}_t^\lambda(\theta) - \mathcal{L}_t^\lambda(\theta_*) - \nabla\mathcal{L}_t^\lambda(\theta_*)(\theta - \theta_*)| \leq \underbrace{|\mathcal{L}_t^\lambda(\theta) - \mathcal{L}_t^\lambda(\theta_*)|}_{(a)} + \underbrace{|\nabla\mathcal{L}_t^\lambda(\theta_*)(\theta - \theta_*)|}_{(b)}.$$

For $(a)$, with probability at least $1 - \delta$ we have that $\theta_* \in \mathcal{C}_t(\delta) \cap \mathbb{B}_d(S)$, then

$$\begin{aligned}
(a) &= |\mathcal{L}_t^\lambda(\theta) - \mathcal{L}_t^\lambda(\hat{\theta}_t) + \mathcal{L}_t^\lambda(\hat{\theta}_t) - \mathcal{L}_t^\lambda(\theta_*)| \\
&\leq |\mathcal{L}_t^\lambda(\theta) - \mathcal{L}_t^\lambda(\hat{\theta}_t)| + |\mathcal{L}_t^\lambda(\hat{\theta}_t) - \mathcal{L}_t^\lambda(\theta_*)| \\
&= \mathcal{L}_t^\lambda(\theta) - \mathcal{L}_t^\lambda(\hat{\theta}_t) + \mathcal{L}_t^\lambda(\theta_*) - \mathcal{L}_t^\lambda(\hat{\theta}_t) \\
&\leq 2\beta_t(\delta)^2,
\end{aligned}$$

where in the third line we used the fact that $\mathcal{L}_t^\lambda(\theta) \geq \mathcal{L}_t^\lambda(\hat{\theta}_t)$ for all $\theta \in \mathbb{R}^d$; in the last line we used **??** and that $\theta \in \mathcal{C}_t(\delta)$. For $(b)$, note that by definition of $\hat{\theta}_t$, $\nabla\mathcal{L}_t^\lambda(\theta_*) = g_t(\theta_*) - g_t(\hat{\theta}_t)$. To be more specific, for all $\theta \in \mathbb{R}^d$,

$$\nabla_\theta\mathcal{L}(\theta) = g_t(\theta) - \underbrace{\sum_{i=1}^{t-1}\phi(S_i, A_i)X_i}_{=g_t(\hat{\theta}_t) \text{ by Eq. (6)}}. \tag{29}$$

Then by Cauchy-Schwarz, by Lemma 13, with probability $1 - \delta$, we have that $\|g_t(\hat{\theta}_t) - g_t(\hat{\theta}_*)\|_{H_t^{-1}(\theta_*)} \leq \rho_t(\delta)$, then

$$\begin{aligned}
(b) &\leq \|g_t(\hat{\theta}_t) - g_t(\theta_*)\|_{H_t^{-1}(\theta_*)}\|\theta - \theta_*\|_{H_t(\theta_*)} \\
&\leq \rho_t(\delta)\|\theta - \theta_*\|_{H_t(\theta_*)},
\end{aligned}$$

where in the last inequality we used Lemma 13. Chaining all the inequality together and use the fact that $\mathbb{P}(A \cap B) \geq 1 - \mathbb{P}(A^c) - \mathbb{P}(B^c)$, we have that with probability at least $1 - 2\delta$, we have that

$$\frac{1}{2(2+2S)}\|\theta - \theta_*\|_{H_t(\theta_*)}^2 \leq \rho_t(\delta)\|\theta - \theta_*\|_{H_t(\theta_*)} + 2\beta_t(\delta)^2.$$

Solving the above inequality gives us

$$\|\theta - \theta_*\|_{H_t(\theta_*)} \leq (4 + 4S)\rho_t(\delta) + \sqrt{(8S + 8)}\beta_t(\delta).$$

$\square$

### D.3 PROOF OF THE REGRET BOUND OF MULOG (THEOREM 2)

In this section, we first state the formal version of Theorem 2 where all the constants and dependencies are detailed then proofs are provided.

**Theorem 7** (Formal statement of Theorem 2). *Let $\delta \in [0, 1)$. Under Assumptions 1 and 2, there exists some universal constant $\mathfrak{c} > 0$ such that the following holds: Let $\delta \in [0, 1)$, $T \geq 1$ be arbitrary. Then, with probability at least $1 - \delta$, it holds that the simple regret of the policy $\hat{\pi}$ computed by MULOG (Algorithm 2) with $\lambda$ chosen to be $\lambda_T$ in Eq. (26), after $T$ rounds is upper bounded by*

$$\begin{aligned}
\mathfrak{R}^{\text{Log}}(\hat{\pi}) \leq{}& \frac{\gamma_{T+1}(\delta/2)\sqrt{d\log((d\lambda_T + T)/(d\lambda_T))}}{\sqrt{T}} \\
&+ \frac{\kappa\gamma_{T+1}(\delta/2)^2 d\log((d\lambda_T + T)/(d\lambda_T))}{T} + \frac{16\log(\log(4T/\delta))}{T} \\
&+ \frac{1}{16T}\kappa^3\gamma_{T+1}(\delta/2)^2\left(16\log(\log(4T/\delta)) + d\log((d\lambda_T + T)/(d\lambda_T))\right)
\end{aligned}$$

A related objective to $\mathfrak{R}$, which is explained in the lemma that follows, when $\theta_{T+1}^{\mathrm{Log}}$ is used, is the expected maximum prediction error

$$D_{\mathrm{Log}}(\theta_{T+1}^{\mathrm{Log}}) := \int \max_{a \in \mathcal{A}} \left| \mu\left(\phi(s,a)^\top \theta^*\right) - \mu\left(\phi(s,a)^\top \theta_{T+1}^{\mathrm{Log}}\right) \right| \nu(ds) \qquad (30)$$

**Lemma 15.** *For a vector $\hat{\theta} \in \mathbb{R}^d$, let $\hat{\pi}$ be greedy w.r.t $\hat{\theta}$, it then follows that*

$$\mathfrak{R}_{\mathrm{Log}}(\hat{\pi}) \leq 2 D_{\mathrm{Log}}(\hat{\theta}).$$

*Proof.* The proof is a simple application of triangle inequality. We add and subtract the same term below. For all $s \in \mathcal{S}$,

$$\mu\left(\phi(s,\pi^*(s))^\top \theta^*\right) - \mu\left(\phi(s,\hat{\pi}(s))^\top \theta^*\right)$$

$$= \mu\left(\phi(s,\pi^*(s))^\top \theta^*\right) - \mu\left(\phi(s,\pi^*(s))^\top \hat{\theta}\right) + \mu\left(\phi(s,\pi^*(s))^\top \hat{\theta}\right) - \mu\left(\phi(s,\hat{\pi}(s))^\top \theta^*\right)$$

$$\leq \mu\left(\phi(s,\pi^*(s))^\top \theta^*\right) - \mu\left(\phi(s,\pi^*(s))^\top \hat{\theta}\right) + \mu\left(\phi(s,\hat{\pi}(s))^\top \hat{\theta}\right) - \mu\left(\phi(s,\hat{\pi}(s))^\top \theta^*\right)$$

$$\leq \left| \mu\left(\phi(s,\pi^*(s))^\top \theta^*\right) - \mu\left(\phi(s,\pi^*(s))^\top \hat{\theta}\right) \right| + \left| \mu\left(\phi(s,\hat{\pi}(s))^\top \hat{\theta}\right) - \mu\left(\phi(s,\hat{\pi}(s))^\top \theta^*\right) \right|$$

$$\leq 2 \max_{a \in \mathcal{A}} \left| \mu\left(\phi(s,a)^\top \theta^*\right) - \mu\left(\phi(s,a)^\top \hat{\theta}\right) \right|,$$

where in the third line we used the fact that $\mu$ is an increasing function and the definition of $\hat{\pi}(s)$. Finally taking integral on both sides finishes the proof. $\qquad\square$

The expected simple regret is upper bounded by $2 D_{\mathrm{Log}}(\theta_{T+1}^{\mathrm{Log}})$ hence we focus on bounding $D_{\mathrm{Log}}(\theta_{T+1}^{\mathrm{Log}})$.

We are going to use the following lemma to establish decreasing uncertainty. It is written in a compact form that is reusable in other contexts. When applying it to our setting, we set $\mathcal{Y} = \mathcal{W}_t$, $\mathcal{Y}' = \mathcal{W}_{T+1}$ and $K' = L_{T+1}$, $K = L_t$ for $t \leq T$. The bounded closed set is for the purpose of ensuring the maximum is attained.

**Lemma 3** (Decreasing Uncertainty Lemma – Logistic Bandits). *Let $K' \succeq K$ be $d \times d$ positive definite matrices and $\mathcal{Y}' \subseteq \mathcal{Y} \subseteq \mathbb{R}^d$ bounded closed sets. Then,*

$$\int \max_{a \in \mathcal{A}(s),\, \theta \in \mathcal{Y}'} U(s,a,\theta,K') \nu(ds) \leq \int \max_{a \in \mathcal{A}(s),\, \theta \in \mathcal{Y}} U(s,a,\theta,K) \nu(ds).$$

*Proof.* Fix $(s,a) \in \mathcal{Z}$, $\theta \in \mathbb{R}^d$. Since $K' \succeq K$ and $\dot{\mu}$ is positive valued,

$$U(s,a,\theta,K') = \dot{\mu}(\phi(s,a)^\top \theta) \|\phi(s,a)\|_{(K')^{-1}} \leq \dot{\mu}(\phi(s,a)^\top \theta) \|\phi(s,a)\|_{K^{-1}} = U(s,a,\theta,K') .$$

Since $\mathcal{Y}' \subseteq \mathcal{Y}$, by the definition of $U(s,a,\theta,K)$, we have

$$\max_{a \in \mathcal{A}(s), \theta \in \mathcal{Y}'} U(s,a,\theta,K') \leq \max_{a \in \mathcal{A}(s), \theta \in \mathcal{Y}} U(s,a,\theta,K) .$$

Integrate over $s$ on both sides using $\nu$ gives the result. $\qquad\square$

$$D_{\mathrm{Log}}(\theta_{T+1}^{\mathrm{Log}}) = \int \max_a |\mu(\phi(s,a)^\top \theta_{T+1}^{\mathrm{Log}}) - \mu(\phi(s,a)^\top \theta_{T+1}^{\mathrm{Log}})| \nu(ds)$$

$$\leq \int \max_a \dot{\mu}(\phi(s,a)^\top \theta_{T+1}^{\mathrm{Log}}) |\phi(s,a)^\top (\theta_* - \theta_{T+1}^{\mathrm{Log}})| + \ddot{\mu}(\xi) |\phi(s,a)^\top (\theta_* - \theta_{T+1}^{\mathrm{Log}})|^2 \nu(ds)$$

$$\leq \int \max_a \dot{\mu}(\phi(s,a)^\top \theta_{T+1}^{\mathrm{Log}}) |\phi(S_1,a)^\top (\theta_* - \theta_{T+1}^{\mathrm{Log}})| \nu(ds) + \int_s \max_a \ddot{\mu}(\xi_a) |\phi(S_1,a)^\top (\theta_* - \theta_{T+1}^{\mathrm{Log}})|^2 \nu(ds),$$

$$= \underbrace{\mathbb{E}\left[ \max_a \dot{\mu}(\phi(S_{T+1},a)^\top \theta_{T+1}^{\mathrm{Log}}) |\phi(S_{T+1},a)^\top (\theta_* - \theta_{T+1}^{\mathrm{Log}})| \Big| \mathcal{F}_T \right]}_{R_1}$$

$$+ \underbrace{\mathbb{E}\left[ \max_a \ddot{\mu}(\xi_a) |\phi(S_{T+1},a)^\top (\theta_* - \theta_{T+1}^{\mathrm{Log}})|^2 \Big| \mathcal{F}_T \right]}_{R_2}, \qquad (31)$$

where $\xi_a$ in the last line is some point in between $\phi(S, a)^\top \theta_*$ and $\phi(S, a)^\top \theta_{T+1}^{\text{Log}}$ for all $a$; in the third line we upper bound $\ddot\mu$ by 1 (Eqs. (24) and (25)). We start by bounding $R_1$; the last line follows by definition of the conditional expectation. By Cauchy-Schwarz, using Lemma 14 to obtain a bound on $\|\theta_* - \theta_{T+1}^{\text{Log}}\|_{H_{T+1}(\theta_*)}$, with probability at least $1 - 2\delta$,

$$R_1 \leq \mathbb{E}\left[\max_a \dot\mu\left(\phi(S_{T+1}, a)^\top \theta_{T+1}^{\text{Log}}\right)\|\phi(S_{T+1}, a)\|_{H_{T+1}^{-1}(\theta_*)}\|\theta_* - \theta_{T+1}^{\text{Log}}\|_{H_{T+1}(\theta_*)}\Big|\mathcal{F}_T\right]$$

$$\leq \gamma_{T+1}(\delta)\mathbb{E}\left[\max_a \dot\mu\left(\phi(S_{T+1}, a)^\top \theta_{T+1}^{\text{Log}}\right)\|\phi(S_{T+1}, a)\|_{H_{T+1}^{-1}(\theta_*)}\Big|\mathcal{F}_T\right]$$

$$\leq \gamma_{T+1}(\delta)\mathbb{E}\left[\max_a \dot\mu\left(\phi(S_{T+1}, a)^\top \theta_{T+1}^{\text{Log}}\right)\|\phi(S_{T+1}, a)\|_{L_{T+1}^{-1}}\Big|\mathcal{F}_T\right]$$

$$\leq \gamma_{T+1}(\delta)\mathbb{E}\left[\max_{a,\theta\in\mathcal{W}_{T+1}} \dot\mu\left(\phi(S_{T+1}, a)^\top \theta\right)\|\phi(S_{T+1}, a)\|_{L_{T+1}^{-1}}\Big|\mathcal{F}_T\right], \tag{32}$$

where in the third line we use the fact that $H_{T+1}(\theta_*) \succeq L_{T+1}$ and the last line follows from the definition of $\mathcal{W}_{T+1}$. We have that $\mathcal{W}_{t+1} \subseteq \mathcal{W}_t$ by definition, and $L_{t+1} \subseteq L_t$ by definition. By letting $\mathcal{Y} = \mathcal{W}_t$ and $\mathcal{Y}' = \mathcal{W}_{t+1}$; $K = L_t$ and $K' = L_{t+1}$ in Lemma 3, it then follows that for all $1 \leq t \leq T$,

$$\mathbb{E}\left[\max_{a,\theta\in\mathcal{W}_{T+1}} U(S_{T+1}, a, \theta, L_{T+1})\Big|\mathcal{F}_T\right] \leq \mathbb{E}\left[\max_{a,\theta\in\mathcal{W}_t} U(S_t, a, \theta, L_t)\Big|\mathcal{F}_{t-1}\right].$$

Hence,

$$\mathbb{E}\left[\max_{a,\theta\in\mathcal{W}_{T+1}} U(S_{T+1}, a, \theta, L_{T+1})\Big|\mathcal{F}_T\right] \leq \frac{1}{T}\sum_{t=1}^T \mathbb{E}\left[\max_{a,\theta\in\mathcal{W}_t} U(S_t, a, \theta, L_t)\Big|\mathcal{F}_{t-1}\right] \tag{33}$$

$$= \frac{1}{T}\sum_{t=1}^T \mathbb{E}\left[U(S_t, A_t, \theta_t, L_t)\Big|\mathcal{F}_{t-1}\right]. \tag{34}$$

Plug the above result back in $R_1$, and use Corollary 4,

$$R_1 \leq \gamma_{T+1}(\delta)\frac{1}{T}\left(4\sqrt{\log(\log(2T/\delta))} + \sqrt{\sum_{t=1}^T \dot\mu\left(\phi(S_t, A_t)^\top \theta_t\right)\|\phi(S_t, A_t)\|_{L_t^{-1}}}\right)^2$$

$$\leq 2\gamma_{T+1}(\delta)\frac{1}{T}\left(16\log(\log(2T/\delta)) + \sum_{t=1}^T \dot\mu\left(\phi(S_t, A_t)^\top \theta_t\right)\|\phi(S_t, A_t)\|_{L_t^{-1}}\right).$$

For reasons why we need Corollary 4 instead of directly using elliptical potential lemma (Lemma 19), we refer the reader to the argument following Eq. (19). In order to reduce clutter, we let $\phi_t := \phi(S_t, A_t)$. It remains to bound $\sum_{t=1}^T \dot\mu(\phi_t^\top \theta_t)\|\phi_t\|_{L_t^{-1}}$. Applying Taylor expansion at $\theta_t'$, for $t \geq 1$ and $\zeta_t$ between $\theta_t$ and $\theta_*$, we have that

$$\sum_{t=1}^T \dot\mu(\phi_t^\top \theta_t)\|\phi_t\|_{L_t^{-1}} = \sum_{t=1}^T \dot\mu(\phi_t^\top \theta_t')\|\phi_t\|_{L_t^{-1}} + \sum_{t=1}^T \ddot\mu(\zeta_t)\|\phi_t\|_{L_t^{-1}}|\phi_t^\top(\theta_t - \theta_t')|$$

$$\leq \frac{1}{2}\sum_{t=1}^T \sqrt{\dot\mu(\phi_t^\top \theta_t')}\|\phi_t\|_{L_t^{-1}} + \sum_{t=1}^T \ddot\mu(\zeta_t)\|\phi_t\|_{L_t^{-1}}\|\phi_t\|_{H_t^{-1}(\theta_*)} \cdot \underbrace{\|\theta_t' - \theta_t\|_{H_t(\theta_*)}}_{\leq \|\theta_* - \theta_t\|_{H_t(\theta_*)} + \|\theta_* - \theta_t'\|_{H_t(\theta_*)}}$$

$$\leq \frac{1}{2}\sum_{t=1}^T \|\tilde\phi_t\|_{\tilde V_t^{-1}} + \frac{1}{2}\gamma_{T+1}(\delta)\kappa\sum_{t=1}^T \|\phi_t\|_{V_t^{-1}}^2$$

$$\leq \frac{1}{2}\sqrt{T}\sqrt{\sum_{t=1}^T \|\tilde\phi_t\|_{\tilde V_t^{-1}}^2} + \frac{1}{2}\gamma_{T+1}(\delta)\kappa\sum_{t=1}^T \|\phi_t\|_{V_t^{-1}}^2$$

$$\leq \frac{1}{2}\sqrt{T}\sqrt{d\log((d\lambda_T + T)/(d\lambda_T))} + \frac{1}{2}\gamma_{T+1}(\delta)\kappa\left(d\log((d\lambda_T + T)/(d\lambda_T))\right), \tag{35}$$

where in the second line we bounded $\dot{\mu}(\cdot)$ by $1/4$ (Eq. (25)); in the third line we

1. used self-concordance $|\ddot{\mu}(\cdot)| \leq \dot{\mu}(\cdot)$ Eq. (24) and upper bounded $\dot{\mu}(\cdot)$ by $1/4$ (Eq. (25));

2. we applied Lemma 14 twice to bound $\|\theta'_t - \theta_t\|_{H_t(\theta_*)}$ as demonstrated in the inline explanation;

3. we defined $\tilde{\phi}_t := \sqrt{\dot{\mu}(\phi_t^\top \theta_t)} \phi_t$ and $\tilde{V}_t := \sum_{s=1}^{t-1} \tilde{\phi}_s \tilde{\phi}_s^\top = L_t$;

in the last line we applied elliptical potential lemma (Lemma 19) twice. Putting everything together, we have that

$$R_1 \leq \frac{\gamma_{T+1}(\delta)\sqrt{d\log((d\lambda_T + T)/(d\lambda_T))}}{\sqrt{T}} + \frac{\kappa\gamma_{T+1}(\delta)^2 d\log((d\lambda_T + T)/(d\lambda_T))}{T} + \frac{16\log(\log(2T/\delta))}{T}$$

Now we move to $R_2$. We need a lemma that's similar to Lemma 3 that we will use to bound $R_2$.

**Lemma 16** (Second order decreasing uncertainty – Logistic Bandits). *Let $K' \succeq K$ be $d \times d$ positive definite matrices and $\mathcal{Y}' \subseteq \mathcal{Y} \subseteq \mathbb{R}^d$ be bounded closed sets. Then,*

$$\int \max_{a \in \mathcal{A}(s), \theta \in \mathcal{Y}'} U(s, a, \theta, K')^2 \nu(ds) \leq \int \max_{a \in \mathcal{A}(s), \theta \in \mathcal{Y}} U(s, a, \theta, K)^2 \nu(ds).$$

*Proof.* Note that for all $s \in \mathcal{S}, \mathcal{Y} \subseteq \mathbb{R}^d$ and positive definite $K \in \mathbb{R}^{d \times d}$, we have that

$$\arg\max_{a \in \mathcal{A}(s), \theta \in \mathcal{Y}} U(s, a, \theta, K)^2 = \arg\max_{a \in \mathcal{A}(s), \theta \in \mathcal{Y}} U(s, a, \theta, K).$$

Everything then follows from the proof of Lemma 3. □

We can now bound $R_2$. By Cauchy-Schwarz, we have that

$$R_2 \leq \frac{1}{4}\mathbb{E}\left[\max_a \left|\phi(S_{T+1}, a)^\top (\theta_* - \theta_{T+1}^{\text{Log}})\right|^2 \bigg| \mathcal{F}_T\right]$$

$$\leq \frac{1}{4}\mathbb{E}\left[\max_a \|\phi(S_{T+1}, a)\|^2_{H_{T+1}^{-1}(\theta_*)} \|\theta_* - \theta_{T+1}^{\text{Log}}\|^2_{H_{T+1}(\theta_*)} \bigg| \mathcal{F}_T\right]$$

$$\leq \frac{1}{4}\kappa^2 \gamma_{T+1}(\delta)^2 \mathbb{E}\left[\max_{a, \theta \in \mathcal{W}_{T+1}} \dot{\mu}(\phi(S_{T+1}, a)^\top \theta)^2 \|\phi(S_{T+1}, a)\|^2_{H_{T+1}^{-1}(\theta_*)} \bigg| \mathcal{F}_T\right]$$

$$\leq \frac{1}{4}\kappa^2 \gamma_{T+1}(\delta)^2 \mathbb{E}\left[\max_{a, \theta \in \mathcal{W}_{T+1}} \dot{\mu}(\phi(S_{T+1}, a)^\top \theta)^2 \|\phi(S_{T+1}, a)\|^2_{L_{T+1}^{-1}} \bigg| \mathcal{F}_T\right]$$

$$\leq \frac{1}{4T}\kappa^2 \gamma_{T+1}(\delta)^2 \sum_{t=1}^T \mathbb{E}\left[\dot{\mu}(\phi(S_t, A_t)^\top \theta_t)^2 \|\phi(S_t, A_t)\|^2_{L_t^{-1}} \big| \mathcal{F}_{t-1}\right]$$

$$\leq \frac{1}{64T}\kappa^3 \gamma_{T+1}(\delta)^2 \sum_{t=1}^T \mathbb{E}\left[\|\phi(S_t, A_t)\|^2_{V_t^{-1}} \big| \mathcal{F}_{t-1}\right]$$

$$\leq \frac{1}{64T}\kappa^3 \gamma_{T+1}(\delta)^2 \left(4\sqrt{\log(\log(2T/\delta))} + \sqrt{\sum_{t=1}^T \|\phi_t\|^2_{V_t^{-1}}}\right)^2$$

$$\leq \frac{1}{32T}\kappa^3 \gamma_{T+1}(\delta)^2 \left(16\log(\log(2T/\delta)) + \sum_{t=1}^T \|\phi_t\|^2_{V_t^{-1}}\right)$$

$$\leq \frac{1}{32T}\kappa^3 \gamma_{T+1}(\delta)^2 \left(16\log(\log(2T/\delta)) + d\log((d\lambda_T + T)/(d\lambda_T))\right),$$

where in the second line we used Cauchy-Schwarz; in the third line we applied definition of $\mathcal{W}_{T+1}$ and the fact that $\theta_{T+1} \in \mathcal{W}_{T+1}$ by construction; in the fourth line $H_{T+1}(\theta_*) \succeq L_{T+1}$; in the fifth line we used Lemma 16; in the sixth line we used $L_t \succeq \frac{1}{\kappa}V_t$; in the seventh line we used Corollary 4; in the eighth line we used $(a + b)^2 \leq 2(a^2 + b^2)$; in the ninth line we used Lemma 19.

Chaining the result for $R_1$ and $R_2$ together, the regret can be upper bounded by

$$\mathfrak{R}(\hat{\pi}) \leq 2D(\theta_{T+1}^{\mathrm{Log}})$$

$$\leq 2(R_1 + R_2)$$

$$\leq \frac{2\gamma_{T+1}(\delta)\sqrt{d\log((d\lambda_T + T)/(d\lambda_T))}}{\sqrt{T}} + \frac{2\kappa\gamma_{T+1}(\delta)^2 d\log((d\lambda_T + T)/(d\lambda_T))}{T}$$

$$+ \frac{32\log(\log(2T/\delta))}{T}$$

$$+ \frac{1}{16T}\kappa^3\gamma_{T+1}(\delta)^2 \left(16\log(\log(2T/\delta)) + d\log((d\lambda_T + T)/(d\lambda_T))\right).$$

Finally replace $\delta$ with $\delta/2$ finishes the proof.

### D.4 DETAILED EXPLANATION ON OUR DIFFERENCE WITH FAURY ET AL. (2020)

As is mentioned before, our algorithm is different from Faury et al. (2020) in several aspects where we did novel algorithmic enhancements explained in the following to make the algorithm provably $\kappa$-free:

1. We avoid their non-convex optimization problem by adopting tools introduced in Abeille et al. (2021), making our algorithm computationally tractable. Note that naively replacing the admissible parameter set in Faury et al. (2020) with the one from Abeille et al. (2021) would not work. The reason is that the objective in Faury et al. (2020) (Eq. 9) is a non-convex function of $\theta$.

2. We completely changed the purpose of $\theta_t$. In Faury et al. (2020), it was used to shape an admissible parameter set. In our case, not only did we drop the admissible set shaped by $\theta_t$ to avoid intractability as mentioned before, but we also incorporated it into the quantification of uncertainty, the key to max-uncertainty type algorithm, serving as a non-trivial extension from the linear case to the logistic case. As a result, Lemmas 3 and 16 are novel.

3. The matrix is directly used in the algorithm, allowing us to construct an estimation of Hessian in an online-fashion. Sherman-Morrison can then be used to avoid matrix inversion in each step. We also directly use as the uncertainty quantification (exploration bonus in cumulative regret setting) while Faury et al. (2020) still uses $H_t$ and $V_t$ as the uncertainty quantification.

## E REGRET ANALYSIS OF THATS (THEOREM 4)

In this section, we first state formally our regret bound on THATS where detailed constant and polynomial dependency is presented. After that we give proof on it.

**Theorem 8** (Formal statement of Theorem 4). *Under Assumptions 1 and 2, there exists some universal constant $\mathfrak{c} > 0$ such that the following holds: Let $\delta \in [0,1)$, $T \geq 1$ be arbitrary. Then, with probability at least $1 - \delta$, it holds that the simple regret of the policy $\hat{\pi}$ computed by THATS (Algorithm 4) with an appropriate choice of $\lambda$ after $T$ rounds is upper bounded by*

$$\mathfrak{R}(\hat{\pi}) \leq \frac{d\gamma_{T+1}(\delta)\sqrt{\log((d\lambda_T + T)/(d\lambda_T))}}{\sqrt{T}} + \frac{\kappa\gamma_{T+1}(\delta)^2 d^{3/2}\lambda_T \log((d\lambda_T + T)/(d\lambda_T))}{T}$$

$$+ \frac{16\sqrt{d}\log(\log(2T/\delta))}{T}$$

$$+ \frac{2\left(d\kappa^3\gamma_{T+1}(\delta) + d\kappa^3\gamma_{T+1}^2(\delta) + \frac{d\kappa^5}{\lambda_1^2}\gamma_{T+1}^4(\delta)\right) \cdot \left(16\log(\log(2T/\delta)) + d\log\left(\frac{d\lambda_T + T}{d\lambda_T}\right)\right)}{T}$$

### E.1 NEW CONFIDENCE SET

Recall the definition of $\mathcal{E}_t(\delta, \bar{\theta}_t)$:

$$\mathcal{E}_t(\delta, \bar{\theta}_t) = \{\theta \in \mathbb{B}_d(S) : \mathcal{L}_t^\lambda(\theta) - \mathcal{L}_t^\lambda(\bar{\theta}_t) \leq 2\bar{\beta}_t^2(\delta)\}$$

As is promised in Section 3.2, we relate $\mathcal{E}_t(\delta)$ to the confidence set $\mathcal{C}_t(\delta)$.

**Lemma 5.** *Let $\delta \in [0, 1)$. With probability at least $1 - \delta$, $\bar{\theta}_t \in \mathcal{C}_t(\delta, \hat{\theta}_t) \cap \mathbb{B}_d(S)$. Furthermore, with probability at least $1 - \delta$, $\mathcal{C}_t(\delta, \hat{\theta}_t) \cap \mathbb{B}_d(S) \subseteq \mathcal{E}_t(\delta, \bar{\theta}_t)$.*

*Proof.* By definition of $\bar{\theta}_t$, we have $\bar{\theta}_t \in \mathbb{B}_d(S)$. Since $\theta_* \in \mathbb{B}_d(S)$, it then follows that with probability at least $1 - \delta$,

$$\mathcal{L}_t^\lambda(\bar{\theta}_t) - \mathcal{L}_t^\lambda(\hat{\theta}_t) \leq \mathcal{L}_t^\lambda(\theta_*) - \mathcal{L}_t^\lambda(\hat{\theta}_t) \leq \beta_t(\delta)^2,$$

where in the first inequality we used the fact that $\hat{\theta}_t$ is the global minimizer of $\mathcal{L}_t^\lambda$, with probability at least $1 - \delta$, $\theta_* \in \mathcal{C}_t(\delta, \hat{\theta}_t) \cap \mathbb{B}_d(S)$ and $\mathcal{L}_t^\lambda(\bar{\theta}_t) \leq \mathcal{L}_t^\lambda(\theta_*)$ by definition of $\bar{\theta}_t$; in the second inequality we used **??**. For the "furthermore" part, let $\theta \in \mathcal{C}_t(\delta, \hat{\theta}_t) \cap \mathbb{B}_d(S)$, by construction $\mathcal{L}_t^\lambda(\bar{\theta}_t) \leq \mathcal{L}_t^\lambda(\theta)$ hence $\mathcal{L}_t^\lambda(\theta) - \mathcal{L}_t^\lambda(\bar{\theta}_t) = |\mathcal{L}_t^\lambda(\theta) - \mathcal{L}_t^\lambda(\bar{\theta}_t)|$ and by triangle inequality,

$$\begin{aligned}
\mathcal{L}_t^\lambda(\theta) - \mathcal{L}_t^\lambda(\bar{\theta}_t) &\leq |\mathcal{L}_t^\lambda(\theta) - \mathcal{L}_t^\lambda(\hat{\theta}_t)| + |\mathcal{L}_t^\lambda(\bar{\theta}_t) - \mathcal{L}_t^\lambda(\hat{\theta}_t)| \\
&= \mathcal{L}_t^\lambda(\theta) - \mathcal{L}_t^\lambda(\hat{\theta}_t) + \mathcal{L}_t^\lambda(\bar{\theta}_t) - \mathcal{L}_t^\lambda(\hat{\theta}_t) \\
&\leq 2\beta_t(\delta)^2.
\end{aligned}$$

$\square$

Given the new confidence set, we can show a similar lemma to Lemma 14.

**Lemma 17.** *Let $\delta \in [0, 1)$. With probability at least $1 - 2\delta$, for all $t \geq 1$ and $\theta \in \mathcal{E}_t(\delta, \bar{\theta}_t)$,*

$$\|\theta - \theta_*\|_{H_t(\theta_*)} \leq 2\gamma_t(\delta).$$

*Proof.* The proof is almost exactly the same as that of Lemma 14. To be more specific, $\hat{\theta}_t$ is replaced by $\bar{\theta}_t$ a few times when needed. We start from Taylor expansion. For all $\theta' \in \mathbb{R}^d$, we have that

$$\mathcal{L}_t^\lambda(\theta') = \mathcal{L}_t^\lambda(\theta_*) + \nabla \mathcal{L}_t^\lambda(\theta_*)^\top (\theta' - \theta_*) + \frac{1}{2} \|\theta' - \theta_*\|_{\tilde{G}_t(\theta_*, \theta')}^2.$$

Let $\theta \in \mathcal{E}_t(\delta, \bar{\theta}_t)$. Rearrange the terms, apply absolute value and plug in $\theta$,

$$|\mathcal{L}_t^\lambda(\theta) - \mathcal{L}_t^\lambda(\theta_*) - \nabla \mathcal{L}_t^\lambda(\theta_*)(\theta - \theta_*)| = \frac{1}{2} \|\theta - \theta_*\|_{\tilde{G}_t(\theta_*, \theta)}^2 \geq \frac{1}{2(2 + 2S)} \|\theta - \theta_*\|_{H_t(\theta_*)}^2,$$

where the last inequality follows from Lemma 12 and $\theta, \theta_* \in \mathbb{B}_d(S)$. It remains to upper bound the left most side of the above equation. By triangle inequality we can split it into two terms and we bound them separately.

$$|\mathcal{L}_t^\lambda(\theta) - \mathcal{L}_t^\lambda(\theta_*) - \nabla \mathcal{L}_t^\lambda(\theta_*)(\theta - \theta_*)| \leq \underbrace{|\mathcal{L}_t^\lambda(\theta) - \mathcal{L}_t^\lambda(\theta_*)|}_{(a)} + \underbrace{|\nabla \mathcal{L}_t^\lambda(\theta_*)(\theta - \theta_*)|}_{(b)}.$$

For $(a)$, with probability at least $1 - \delta$ we have that $\theta_* \in \mathcal{C}_t(\delta) \cap \mathbb{B}_d(S)$, then

$$\begin{aligned}
(a) &= |\mathcal{L}_t^\lambda(\theta) - \mathcal{L}_t^\lambda(\bar{\theta}_t) + \mathcal{L}_t^\lambda(\bar{\theta}_t) - \mathcal{L}_t^\lambda(\theta_*)| \\
&\leq |\mathcal{L}_t^\lambda(\theta) - \mathcal{L}_t^\lambda(\bar{\theta}_t)| + |\mathcal{L}_t^\lambda(\bar{\theta}_t) - \mathcal{L}_t^\lambda(\theta_*)| \\
&= \mathcal{L}_t^\lambda(\theta) - \mathcal{L}_t^\lambda(\bar{\theta}_t) + \mathcal{L}_t^\lambda(\bar{\theta}_t) - \mathcal{L}_t^\lambda(\theta_*) \\
&\leq 4\beta_t(\delta)^2,
\end{aligned}$$

where in the last line we used Lemma 5 and that $\theta_* \in \mathcal{C}_t(\delta) \cap \mathbb{B}_d(S)$. For $(b)$, note that by definition of $\hat{\theta}_t$, $\nabla \mathcal{L}_t^\lambda(\theta_*) = g_t(\theta_*) - g_t(\hat{\theta}_t)$. To be more specific, for all $\theta \in \mathbb{R}^d$,

$$\nabla_\theta \mathcal{L}(\theta) = g_t(\theta) - \underbrace{\sum_{i=1}^{t-1} \phi(S_i, A_i) X_i}_{= g_t(\hat{\theta}_t) \text{ by Eq. (6)}}. \tag{36}$$

Then by Cauchy-Schwarz, by Lemma 13, with probability $1 - \delta$, we have that $\|g_t(\hat{\theta}_t) - g_t(\hat{\theta}_*)\|_{H_t^{-1}(\theta_*)} \leq \rho_t(\delta)$, then

$$(b) \leq \|g_t(\hat{\theta}_t) - g_t(\theta_*)\|_{H_t^{-1}(\theta_*)} \|\theta - \theta_*\|_{H_t(\theta_*)}$$
$$\leq \rho_t(\delta) \|\theta - \theta_*\|_{H_t(\theta_*)},$$

where in the last inequality we used Lemma 13. Chaining all the inequality and use the fact that $\mathbb{P}(A \cap B) \geq 1 - \mathbb{P}(A^c) - \mathbb{P}(B^c)$, we have that with probability at least $1 - 2\delta$,

$$\frac{1}{2(2 + 2S)} \|\theta - \theta_*\|_{H_t(\theta_*)}^2 \leq \rho_t(\delta) \|\theta - \theta_*\|_{H_t(\theta_*)} + 4\beta_t(\delta)^2.$$

Solving the above inequality gives us

$$\|\theta - \theta_*\|_{H_t(\theta_*)} \leq (4 + 4S)\rho_t(\delta) + \sqrt{(16S + 16)}\beta_t(\delta) \leq 2\gamma_t(\delta).$$

$\square$

### E.2 ANALYSIS ON THE EXPLORATION DONE BY THATS COMPARED TO MULOG

The analysis of THATS is highly related to that of MULOG. Hence in order to identify the actions and parameter that gives the maximum uncertainty, we define them to be

$$\theta_t^{\mathrm{MU}}, A_t^{\mathrm{MU}} = \underset{a \in \mathcal{A}(S_t), \theta \in \mathcal{V}_t}{\arg\max} U(S_t, a, \theta, L_t) \tag{37}$$

$$\omega_t^{\mathrm{MU}} = \underset{\theta \in \mathcal{E}_t(\delta, \bar{\theta}_t)}{\arg\min} \dot{\mu}(\phi(S_t, A_t^{\mathrm{MU}})^\top \theta). \tag{38}$$

We would like to emphasize that $\theta_t^{\mathrm{MU}}, A_t^{\mathrm{MU}}$ are **not** actions and parameters pulled by the max-uncertainty algorithm. They are simply the ones that give the maximum uncertainty at time $t$. The actions $A_t$ are pulled by THATS instead of max uncertainty in this section.

**Lemma 6.** *Let $t \geq 1$, $A_t^{\mathrm{MU}} = \arg\max_{a \in \mathcal{A}(S_t)} \max_{\theta \in \mathcal{V}_t} \dot{\mu}(\phi(S_t, a)^\top \theta)\|\phi(S_t, a)\|_{L_t^{-1}}$. Then, it holds almost surely that for $I(\cdot)$ defined in Lemma 4,*

$$\min_{\theta \in \mathcal{E}_t} U(S_t, A_t^{\mathrm{MU}}, \theta, L_t) \cdot I(S_t, A_t^{\mathrm{MU}}, L_t) \leq \mathbb{E}\left[ U(S_t, A_t, \bar{\theta}_t, L_t) \Big| \mathcal{F}_{t-1}, S_t \right].$$

*Proof.* We start by rewriting the right hand side of the inequality

$$\mathbb{E}\left[ \dot{\mu}(\phi(S_t, A_t)^\top \bar{\theta}_t) \|\phi(S_t, A_t)\|_{L_t^{-1}} \Big| \mathcal{F}_{t-1}, S_t \right]$$

$$= \mathbb{E}\left[ \max_{x \in \mathbb{S}^{d-1}} \left\langle L_t^{-1/2} x, \dot{\mu}(\phi(S_t, A_t)^\top \bar{\theta}_t) \phi(S_t, A_t) \right\rangle \Big| \mathcal{F}_{t-1}, S_t \right]$$

$$\geq \mathbb{E}\left[ \left| \left\langle \frac{L_t^{-1/2} \cdot L_t^{1/2} \tilde{\theta}_t}{\|L_t^{1/2} \tilde{\theta}_t\|_2}, \dot{\mu}(\phi(S_t, A_t)^\top \bar{\theta}_t) \phi(S_t, A_t) \right\rangle \right| \Big| \mathcal{F}_{t-1}, S_t \right]$$

$$\geq \mathbb{E}\left[ \left| \left\langle \frac{\tilde{\theta}_t}{\|L_t^{1/2} \tilde{\theta}_t\|_2}, \dot{\mu}(\phi(S_t, A_t^{\mathrm{MU}})^\top \bar{\theta}_t) \phi(S_t, A_t^{\mathrm{MU}}) \right\rangle \right| \Big| \mathcal{F}_{t-1}, S_t \right],$$

where in the last line we used the definition of $A_t$. Note that $\bar{\theta}_t$ and $\omega_t^{\mathrm{MU}}$ are both in the confidence set $\mathcal{E}_t(\delta, \bar{\theta}_t) \cap \mathbb{B}_d(S)$,

$$\mathbb{E}\left[ \left| \left\langle \frac{\tilde{\theta}_t}{\|L_t^{1/2} \tilde{\theta}_t\|_2}, \dot{\mu}(\phi(S_t, A_t^{\mathrm{MU}})^\top \bar{\theta}_t) \phi(S_t, A_t^{\mathrm{MU}}) \right\rangle \right| \Big| \mathcal{F}_{t-1}, S_t \right]$$

$$\geq \mathbb{E}\left[ \left| \left\langle \frac{\tilde{\theta}_t}{\|L_t^{1/2} \tilde{\theta}_t\|_2}, \dot{\mu}(\phi(S_t, A_t^{\mathrm{MU}})^\top \omega_t^{\mathrm{MU}}) \phi(S_t, A_t^{\mathrm{MU}}) \right\rangle \right| \Big| \mathcal{F}_{t-1}, S_t \right]$$

Since $\tilde{\theta}_t \sim \mathcal{N}(0, L_t^{-1})$, we can rewrite it as $\tilde{\theta}_t = L_t^{-1/2} M_t$ for $M_t \sim \mathcal{N}(0, I)$ given the past and the current context. Then plug it back in the above expression,

$$\mathbb{E}\left[\left|\left\langle \frac{\tilde{\theta}_t}{\|L_t^{1/2}\tilde{\theta}_t\|_2}, \dot{\mu}(\phi(S_t, A_t^{\mathrm{MU}})^\top \omega_t^{\mathrm{MU}})\phi(S_t, A_t^{\mathrm{MU}}) \right\rangle\right| \Big| \mathcal{F}_{t-1}, S_t\right]$$

$$= \mathbb{E}\left[\left|\left\langle \frac{L_t^{-1/2} M_t}{\|M_t\|_2}, \dot{\mu}(\phi(S_t, A_t^{\mathrm{MU}})^\top \omega_t^{\mathrm{MU}})\phi(S_t, A_t^{\mathrm{MU}}) \right\rangle\right| \Big| \mathcal{F}_{t-1}, S_t\right]$$

$$= \mathbb{E}\left[\left|\left\langle \frac{M_t}{\|M_t\|_2}, \dot{\mu}(\phi(S_t, A_t^{\mathrm{MU}})^\top \omega_t^{\mathrm{MU}})L_t^{-1/2}\phi(S_t, A_t^{\mathrm{MU}}) \right\rangle\right| \Big| \mathcal{F}_{t-1}, S_t\right]$$

$$= \dot{\mu}(\phi(S_t, A_t^{\mathrm{MU}})^\top \omega_t^{\mathrm{MU}})\|\phi(S_t, A_t^{\mathrm{MU}})\|_{L_t^{-1}} \mathbb{E}\left[\left|\left\langle \frac{M_t}{\|M_t\|_2}, \frac{L_t^{-1/2}\phi(S_t, A_t^{\mathrm{MU}})}{\|\phi(S_t, A_t^{\mathrm{MU}})\|_{L_t^{-1}}} \right\rangle\right| \Big| \mathcal{F}_{t-1}, S_t\right]$$

$$= \dot{\mu}(\phi(S_t, A_t^{\mathrm{MU}})^\top \omega_t^{\mathrm{MU}})\|\phi(S_t, A_t^{\mathrm{MU}})\|_{L_t^{-1}} I(S_t, A_t^{\mathrm{MU}}, L_t),$$

where the third line used $L_t^{-1/2}$ is symmetric (as $L_t$ is positive definite); the fourth line follows by $A_t^{\mathrm{MU}}, \omega_t^{\mathrm{MU}}$ are $\mathcal{F}_{t-1}$-measurable and Proposition 1 and the last line follows by the definition of $I(S_t, A_t^{\mathrm{MU}}, L_t)$. $\qquad\square$

**Corollary 2.** *For all $t \geq 1$, it holds almost surely that*

$$\mathbb{E}\left[U(S_t, A_t^{\mathrm{MU}}, \omega_t^{\mathrm{MU}}, L_t)\Big|\mathcal{F}_{t-1}\right] \leq \sqrt{\frac{\pi d}{2}}\mathbb{E}\left[U(S_t, A_t, \bar{\theta}_t, L_t)\Big|\mathcal{F}_{t-1}\right].$$

*Proof.* The proof follows by dividing both sides of the inequality showed in Lemma 6 by $I(S_t, A_t^{\mathrm{MU}}, L_t)$ and then showing that $I(S_t, A_t^{\mathrm{MU}}, L_t)$ is lower bounded by $\sqrt{\frac{2}{\pi d}}$. The latter follows using Proposition 2 and Proposition 3. Together we get,

$$U(S_t, A_t^{\mathrm{MU}}, \omega_t^{\mathrm{MU}}, L_t) \leq \sqrt{\frac{\pi d}{2}}\mathbb{E}\left[U(S_t, A_t, \bar{\theta}_t, L_t)\Big|\mathcal{F}_{t-1}, S_t\right],$$

from which the tower rule gives the desired result. $\qquad\square$

**Lemma 18.** *It holds almost surely that*

$$\min_{\theta \in \mathcal{E}_t} U(S_t, A_t^{\mathrm{MU}}, \theta, L_t)^2 \cdot \bar{I}(S_t, A_t^{\mathrm{MU}}, L_t) \leq \mathbb{E}\left[U(S_t, A_t, \bar{\theta}_t, L_t)\Big|\mathcal{F}_{t-1}, S_t\right],$$

*where for $(s, a) \in \mathcal{Z}$ and $L \succeq 0$*

$$\bar{I}(s, a, L) = \int_{\mathbb{S}^{d-1}} \left(\left\langle x, \frac{L^{-1/2}\phi(s, a)}{\|L^{-1/2}\phi(s, a)\|} \right\rangle\right)^2 dx.$$

*Proof.* The proof is similar to that of Lemma 6. We replace square terms a few times when needed compared to the proof of Lemma 6. We start by rewriting the right hand side of the inequality

$$\mathbb{E}\left[\dot{\mu}(\phi(S_t, A_t)^\top \bar{\theta}_t)^2 \|\phi(S_t, A_t)^\top \tilde{\theta}_t\|_{L_t^{-1}}^2 \Big|\mathcal{F}_{t-1}, S_t\right]$$

$$= \mathbb{E}\left[\left(\max_{x \in \mathbb{S}^{d-1}} \left\langle L_t^{-1/2} x, \dot{\mu}(\phi(S_t, A_t)^\top \bar{\theta}_t)\phi(S_t, A_t) \right\rangle\right)^2 \Big|\mathcal{F}_{t-1}, S_t\right]$$

$$\geq \mathbb{E}\left[\left|\left\langle \frac{\tilde{\theta}_t}{\|L_t^{1/2}\tilde{\theta}_t\|_2}, \dot{\mu}(\phi(S_t, A_t)^\top \bar{\theta}_t)\phi(S_t, A_t) \right\rangle\right|^2 \Big|\mathcal{F}_{t-1}, S_t\right]$$

$$\geq \mathbb{E}\left[\left|\left\langle \frac{\tilde{\theta}_t}{\|L_t^{1/2}\tilde{\theta}_t\|_2}, \dot{\mu}(\phi(S_t, A_t^{\mathrm{MU}})^\top \bar{\theta}_t)\phi(S_t, A_t^{\mathrm{MU}}) \right\rangle\right|^2 \Big|\mathcal{F}_{t-1}, S_t\right],$$

where in the last line we used the definition of $A_t$ and that

$$\arg\max_{a \in \mathcal{A}(S_t)} \dot{\mu}(\phi(S_t, a)^\top \bar{\theta}_t) \left| \left\langle \phi(S_t, a), \tilde{\theta}_t \right\rangle \right| = \arg\max_{a \in \mathcal{A}(S_t)} \dot{\mu}(\phi(S_t, a)^\top \bar{\theta}_t)^2 \left( \left\langle \phi(S_t, a), \tilde{\theta}_t \right\rangle \right)^2 \quad (39)$$

By definition of $\omega_t$, the same reasoning as Eq. (39),

$$\mathbb{E}\left[ \left| \left\langle \frac{\tilde{\theta}_t}{\|L_t^{1/2}\tilde{\theta}_t\|_2}, \dot{\mu}(\phi(S_t, A_t^{\text{MU}})^\top \bar{\theta}_t)\phi(S_t, A_t^{\text{MU}}) \right\rangle \right|^2 \Big| \mathcal{F}_{t-1}, S_t \right]$$

$$\geq \mathbb{E}\left[ \left| \left\langle \frac{\tilde{\theta}_t}{\|L_t^{1/2}\tilde{\theta}_t\|_2}, \dot{\mu}(\phi(S_t, A_t^{\text{MU}})^\top \omega_t^{\text{MU}})\phi(S_t, A_t^{\text{MU}}) \right\rangle \right|^2 \Big| \mathcal{F}_{t-1}, S_t \right]$$

Since $\tilde{\theta}_t \sim \mathcal{N}(0, L_t^{-1})$, we can rewrite it as $\tilde{\theta}_t = L_t^{-1/2}M_t$ for $M_t \sim \mathcal{N}(0, I)$. Then plug it back in the above expression,

$$\mathbb{E}\left[ \left| \left\langle \frac{\tilde{\theta}_t}{\|L_t^{1/2}\tilde{\theta}_t\|_2}, \dot{\mu}(\phi(S_t, A_t^{\text{MU}})^\top \omega_t^{\text{MU}})\phi(S_t, A_t^{\text{MU}}) \right\rangle \right|^2 \Big| \mathcal{F}_{t-1}, S_t \right]$$

$$= \mathbb{E}\left[ \left| \left\langle \frac{M_t}{\|M_t\|_2}, L_t^{-1/2}\dot{\mu}(\phi(S_t, A_t^{\text{MU}})^\top \omega_t^{\text{MU}})\phi(S_t, A_t^{\text{MU}}) \right\rangle \right|^2 \mathcal{F}_{t-1}, S_t \right]$$

$$= \dot{\mu}(\phi(S_t, A_t^{\text{MU}})^\top \omega_t^{\text{MU}})^2 \|\phi(S_t, A_t^{\text{MU}})\|_{L_t^{-1}}^2 \mathbb{E}\left[ \left| \left\langle \frac{M_t}{\|M_t\|_2}, \frac{L_t^{-1/2}\phi(S_t, A_t^{\text{MU}})}{\|\phi(S_t, A_t^{\text{MU}})\|_{L_t^{-1}}} \right\rangle \right|^2 \mathcal{F}_{t-1}, S_t \right]$$

$$= U(S_t, A_t^{\text{MU}}, \omega_t^{\text{MU}}, L_t)^2 \cdot \bar{I}(S_t, A_t^{\text{MU}}, L_t),$$

where the second line follows because $L_t^{-1/2}$ is symmetric (as $L_t$ is positive definite); the third line follows by $A_t^{\text{MU}}, \omega_t^{\text{MU}}$ are $\mathcal{F}_{t-1}$-measurable and Proposition 1; the fourth line follows by definition of $\bar{I}(S_t, A_t^{\text{MU}}, L_t)$. $\qquad\square$

**Corollary 3.** *For all $t \geq 1$, it holds almost surely that*

$$\mathbb{E}\left[ U(S_t, A_t^{\text{MU}}, \omega_t^{\text{MU}}, L_t)^2 \Big| \mathcal{F}_{t-1} \right] \leq d\mathbb{E}\left[ U(S_t, A_t, \tilde{\theta}_t, L_t)^2 \Big| \mathcal{F}_{t-1} \right].$$

*Proof.* The proof follows by dividing both sides of the inequality showed in Lemma 6 by $\bar{I}(S_t, A_t^{\text{MU}}, L_t)$ and then showing that $\bar{I}_t(S_t)$ is exactly $1/d$. The latter follows using Proposition 2 and Proposition 4. Together we get,

$$U(S_t, A_t^{\text{MU}}, \omega_t^{\text{MU}}, L_t)^2 \leq dU(S_t, A_t, \tilde{\theta}_t, L_t)^2.$$

Finally tower rule gives the desired result. $\qquad\square$

### E.3   REGRET ANALYSIS THATS

Now we start the regret analysis. The beginning of the analysis is similar to that of Algorithm 2. We do the same Taylor expansion on $D_{\text{Log}}(\hat{\pi})$ where we substitute $\theta_{T+1}^{\text{Log}}$ with $\tilde{\theta}_{T+1}^{\text{Log}}$ as what we did in Eq. (31):

$$D_{\text{Log}}(\hat{\pi}) \leq \underbrace{\mathbb{E}\left[ \max_a \dot{\mu}(\phi(S_{T+1}, a)^\top \tilde{\theta}_{T+1}^{\text{Log}})|\phi(S_{T+1}, a)^\top(\theta_* - \tilde{\theta}_{T+1}^{\text{Log}})| \Big| \mathcal{F}_T \right]}_{R_1}$$

$$+ \underbrace{\mathbb{E}\left[ \max_a \ddot{\mu}(\xi_a)|\phi(S_{T+1}, a)^\top(\theta_* - \tilde{\theta}_{T+1}^{\text{Log}})|^2 \Big| \mathcal{F}_T \right]}_{R_2}.$$

For $R_1$, we copy the analysis from that of Algorithm 2, as what we did in Eq. (32). Recall $\tilde{\theta}_{T+1}^{\text{Log}} \in \mathcal{V}_{T+1}$.

$$
\begin{aligned}
R_1 &\leq \mathbb{E}\left[\max_a \dot{\mu}\left(\phi(S_{T+1}, a)^\top \tilde{\theta}_{T+1}^{\text{Log}}\right) \|\phi(S_{T+1}, a)\|_{H_{T+1}^{-1}(\theta_*)} \|\theta_* - \tilde{\theta}_{T+1}^{\text{Log}}\|_{H_{T+1}(\theta_*)} \Big| \mathcal{F}_T\right] \\
&\leq 2\gamma_{T+1}(\delta)\mathbb{E}\left[\max_a \dot{\mu}\left(\phi(S_{T+1}, a)^\top \tilde{\theta}_{T+1}^{\text{Log}}\right) \|\phi(S_{T+1}, a)\|_{H_{T+1}^{-1}(\theta_*)} \Big| \mathcal{F}_T\right] \\
&\leq 2\gamma_{T+1}(\delta)\mathbb{E}\left[\max_a \dot{\mu}\left(\phi(S_{T+1}, a)^\top \tilde{\theta}_{T+1}^{\text{Log}}\right) \|\phi(S_{T+1}, a)\|_{L_{T+1}^{-1}} \Big| \mathcal{F}_T\right] \\
&\leq 2\gamma_{T+1}(\delta)\mathbb{E}\left[\max_{a,\theta \in \mathcal{V}_{T+1}} \dot{\mu}\left(\phi(S_{T+1}, a)^\top \theta\right) \|\phi(S_{T+1}, a)\|_{L_{T+1}^{-1}} \Big| \mathcal{F}_T\right] \\
&= 2\gamma_{T+1}(\delta)\mathbb{E}\left[\max_{a,\theta \in \mathcal{V}_{T+1}} U(S_{T+1}, a, \theta, L_{T+1}) \Big| \mathcal{F}_T\right],
\end{aligned}
$$

where in the second line we use Lemma 17; in the third line we use $L_{T+1} \preceq H_{T+1}(\theta_*)$; in the fourth line we use $\tilde{\theta}_{T+1}^{\text{Log}} \in \mathcal{V}_{T+1}$. Now we use Lemma 3, it follows that for all $1 \leq t \leq T$, note that $\mathcal{V}_t \subseteq \mathcal{V}_{t+1}$ and $L_t \preceq L_{t+1}$ hence satisfying the conditions of Lemma 3. Using the same argument as Eqs. (33) and (34),

$$
\mathbb{E}\left[\max_{a,\theta \in \mathcal{V}_{T+1}} U(S_{T+1}, a, \theta, L_{T+1}) \Big| \mathcal{F}_T\right] \leq \frac{1}{T}\sum_{t=1}^{T} \mathbb{E}\left[\max_{a,\theta \in \mathcal{V}_t} U(S_t, a, \theta, L_t) \Big| \mathcal{F}_{t-1}\right]. \tag{40}
$$

Recall that

$$
U(s, a, \theta, L) = \dot{\mu}(\phi(s, a)^\top \theta)\|\phi(s, a)\|_{L^{-1}}.
$$

By Eq. (40), with probability $1 - 2\delta$ (from Lemma 17), we have that

$$
\begin{aligned}
&\mathbb{E}\left[\max_{a,\theta \in \mathcal{V}_{T+1}} U(S_{T+1}, a, \theta, L_{T+1}) \Big| \mathcal{F}_T\right] \\
&\leq \frac{1}{T}\sum_{t=1}^{T} \mathbb{E}\left[\max_{a,\theta \in \mathcal{V}_t} U(S_t, a, \theta, L_t) \Big| \mathcal{F}_{t-1}\right] \\
&= \frac{1}{T}\sum_{t=1}^{T} \mathbb{E}\left[U(S_t, A_t^{\text{MU}}, \theta_t^{\text{MU}}, L_t) \Big| \mathcal{F}_{t-1}\right] && \text{(c.f. Eq. (37))} \\
&\leq \frac{1}{T}\sum_{t=1}^{T} \mathbb{E}\left[U(S_t, A_t^{\text{MU}}, \omega_t^{\text{MU}}, L_t) + \frac{1}{4}\|\phi(S_t, A_t^{\text{MU}})\|_{L_t^{-1}}|\phi(S_t, A_t^{\text{MU}})^\top(\theta_t^{\text{MU}} - \omega_t^{\text{MU}})| \Big| \mathcal{F}_{t-1}\right] \\
&\leq \frac{1}{T}\sum_{t=1}^{T} \mathbb{E}\left[U(S_t, A_t^{\text{MU}}, \omega_t^{\text{MU}}, L_t) + \frac{1}{4}\|\phi(S_t, A_t^{\text{MU}})\|_{L_t^{-1}}^2 \|\theta_t^{\text{MU}} - \omega_t^{\text{MU}}\|_{L_t} \Big| \mathcal{F}_{t-1}\right] \\
&\leq \frac{1}{T}\sum_{t=1}^{T} \mathbb{E}\left[U(S_t, A_t^{\text{MU}}, \omega_t^{\text{MU}}, L_t) + \frac{1}{4}\|\phi(S_t, A_t^{\text{MU}})\|_{L_t^{-1}}^2 \|\theta_t^{\text{MU}} - \omega_t^{\text{MU}}\|_{H_t(\theta_*)} \Big| \mathcal{F}_{t-1}\right]
\end{aligned}
$$

where the highlighted line is the line that started to make a big difference to the proof of Algorithm 2; in the second line we use Eq. (40); in the thrid line we plug in the definition of $A_t^{\text{MU}}$ and $\theta_t^{\text{MU}}$; in the fourth line we use Taylor expansion and upper bound $\ddot{\mu}(\cdot)$ by $1/4$ (Eqs. (24) and (25)); in the fifth line we used Cauchy-Schwarz inequality; in the sixth line we used the fact that $L_t \preceq H_t(\theta_*)$. Now we give a way to bound $\|\theta_t^{\text{MU}} - \omega_t^{\text{MU}}\|_{H_t(\theta_*)}$. By triangle inequality, and $\theta_t^{\text{MU}}, \theta_* \in \mathcal{E}_t(\delta, \bar{\theta}_t)$,

$$
\|\theta_t^{\text{MU}} - \omega_t^{\text{MU}}\|_{H_t(\theta_*)} \leq \|\theta_t^{\text{MU}} - \theta_*\|_{H_t(\theta_*)} + \|\theta_* - \omega_t^{\text{MU}}\|_{H_t(\theta_*)} \leq 4\gamma_{T+1}(\delta). \tag{41}
$$

Plug in the above bound, we have

$$\frac{1}{T} \sum_{t=1}^{T} \mathbb{E}\left[ U(S_t, A_t^{\mathrm{MU}}, \omega_t^{\mathrm{MU}}, L_t) + \frac{1}{4}\|\phi(S_t, A_t^{\mathrm{MU}})\|_{L_t^{-1}}^2 \|\theta_t^{\mathrm{MU}} - \omega_t^{\mathrm{MU}}\|_{H_t(\theta_*)} \Big| \mathcal{F}_{t-1} \right]$$

$$\leq \frac{1}{T} \sum_{t=1}^{T} \mathbb{E}\left[ U(S_t, A_t^{\mathrm{MU}}, \omega_t^{\mathrm{MU}}, L_t) + \frac{1}{4} \cdot 4\gamma_{T+1}(\delta)\|\phi(S_t, A_t^{\mathrm{MU}})\|_{L_t^{-1}}^2 \Big| \mathcal{F}_{t-1} \right]$$

$$\leq \underbrace{\frac{1}{T} \sum_{t=1}^{T} \mathbb{E}\left[ U(S_t, A_t^{\mathrm{MU}}, \omega_t^{\mathrm{MU}}, L_t) \Big| \mathcal{F}_{t-1} \right]}_{(i)} + \underbrace{\gamma_{T+1}(\delta)\frac{1}{T} \sum_{t=1}^{T} \mathbb{E}\left[ \|\phi(S_t, A_t^{\mathrm{MU}})\|_{L_t^{-1}}^2 \Big| \mathcal{F}_{t-1} \right]}_{(ii)},$$

in the second line we used the fact that both $\theta_t^{\mathrm{MU}}$ and $\omega_t^{\mathrm{MU}}$ are in $\mathcal{V}_t$ and Lemma 17; in the last line we used the linearity of conditional expectation.

Now it remains to bound $(i)$ and $(ii)$. For $(i)$, we use our result on THATS Corollary 2 to get

$$(i) \leq \frac{1}{T} \sum_{t=1}^{T} \sqrt{d}\mathbb{E}\left[ U(S_t, A_t, \bar{\theta}_t, L_t) \Big| \mathcal{F}_{t-1} \right].$$

Now everything starts to follow almost exactly the same to the part of analysis in Algorithm 2 again. Specifically, Eq. (35). By Corollary 4, and put back the definition of $U(S_t, A_t, \bar{\theta}_t, L_t)$

$$(i) \leq \frac{1}{2}\gamma_{T+1}(\delta)\sqrt{d}\frac{1}{T}\left( 4\sqrt{\log(\log(2T/\delta))} + \sqrt{\sum_{t=1}^{T} \dot{\mu}\left(\phi(S_t, A_t)^\top \bar{\theta}_t\right)\|\phi(S_t, A_t)\|_{L_t^{-1}}} \right)^2$$

$$\leq \gamma_{T+1}(\delta)\sqrt{d}\frac{1}{T}\left( 16\log(\log(2T/\delta)) + \sum_{t=1}^{T} \dot{\mu}\left(\phi(S_t, A_t)^\top \bar{\theta}_t\right)\|\phi(S_t, A_t)\|_{L_t^{-1}} \right).$$

Applying Taylor expansion at $\theta_t'$ and abbreviating $\phi(S_t, A_t) =: \phi_t$, for $t \geq 1$ and $\zeta_t$ between $\bar{\theta}_t$ and $\theta_*$, we have that

$$\sum_{t=1}^{T} \dot{\mu}(\phi_t^\top \bar{\theta}_t)\|\phi_t\|_{L_t^{-1}} = \sum_{t=1}^{T} \dot{\mu}(\phi_t^\top \theta_t')\|\phi_t\|_{L_t^{-1}} + \sum_{t=1}^{T} \ddot{\mu}(\zeta_t)\|\phi_t\|_{L_t^{-1}} |\phi_t^\top(\bar{\theta}_t - \theta_t')|$$

$$\leq \frac{1}{2}\sum_{t=1}^{T} \sqrt{\dot{\mu}(\phi_t^\top \theta_t')}\|\phi_t\|_{L_t^{-1}} + \sum_{t=1}^{T} \ddot{\mu}(\zeta_t)\|\phi_t\|_{L_t^{-1}}\|\phi_t\|_{H_t^{-1}(\theta_*)} \cdot \underbrace{\|\theta_t' - \bar{\theta}_t\|_{H_t(\theta_*)}}_{\leq \|\theta_* - \bar{\theta}_t\|_{H_t(\theta_*)} + \|\theta_* - \theta_t'\|_{H_t(\theta_*)}}$$

$$\leq \frac{1}{2}\sum_{t=1}^{T} \|\tilde{\phi}_t\|_{\tilde{V}_t^{-1}} + \gamma_{T+1}(\delta)\kappa \sum_{t=1}^{T} \|\phi_t\|_{V_t^{-1}}^2$$

$$\leq \frac{1}{2}\sqrt{T}\sqrt{\sum_{t=1}^{T} \|\tilde{\phi}_t\|_{\tilde{V}_t^{-1}}^2 + \gamma_{T+1}(\delta)\kappa \sum_{t=1}^{T} \|\phi_t\|_{V_t^{-1}}^2}$$

$$\leq \frac{1}{2}\sqrt{T}\sqrt{d\log((d\lambda_T + T)/(d\lambda_T))} + \gamma_{T+1}(\delta)\kappa\left(d\log((d\lambda_T + T)/(d\lambda_T))\right),$$

where in the second line we upper bounded $\dot{\mu}(\cdot)$ by $1/4$ (Eq. (25)) in the first term and used Cauchy-Schwarz in the second term; in the third line we

1. used self-concordance $|\ddot{\mu}(\cdot)| \leq \dot{\mu}(\cdot)$ (Eq. (24)) and upper bounded $\dot{\mu}(\cdot)$ by $1/4$ (Eq. (25));

2. we did the same argument as Eq. (41) to bound $\|\theta_t' - \bar{\theta}_t\|_{H_t(\theta_*)}$;

3. we defined $\tilde{\phi}_t := \sqrt{\dot{\mu}(\phi_t^\top \theta_t)}\phi_t$ and $\tilde{V}_t := \sum_{s=1}^{t-1} \tilde{\phi}_s\tilde{\phi}_s^\top = L_t$;

in the last line we applied elliptical potential lemma (Lemma 19) twice. The final bound on $(i)$ is therefore

$$(i) \leq \frac{d\gamma_{T+1}(\delta)\sqrt{\log((d\lambda_T + T)/(d\lambda_T))}}{2\sqrt{T}} + \frac{\kappa\gamma_{T+1}(\delta)^2 d^{3/2}\lambda_T \log((d\lambda_T + T)/(d\lambda_T))}{T} + \frac{8\sqrt{d}\log(\log(2T/\delta))}{T}$$

For $(ii)$,

$$(ii) = \gamma_{T+1}(\delta)\frac{1}{T}\sum_{t=1}^{T}\mathbb{E}\left[\|\phi(S_t, A_t^{\mathrm{MU}})\|_{L_t^{-1}}^2 \Big| \mathcal{F}_{t-1}\right]$$

$$\leq \gamma_{T+1}(\delta)\frac{\kappa^2}{T}\sum_{t=1}^{T}\mathbb{E}\left[\underbrace{\dot{\mu}(\phi(S_t, A_t^{\mathrm{MU}})^{\top}\omega_t^{\mathrm{MU}})\|\phi(S_t, A_t^{\mathrm{MU}})\|_{L_t^{-1}}^2}_{U(S_t, A_t^{\mathrm{MU}}, \omega_t^{\mathrm{MU}}, L_t)^2} \Big| \mathcal{F}_{t-1}\right]$$

$$\leq \gamma_{T+1}(\delta)\frac{d\kappa^2}{T}\sum_{t=1}^{T}\mathbb{E}\left[\underbrace{\dot{\mu}(\phi(S_t, A_t)^{\top}\bar{\theta}_t)\|\phi(S_t, A_t)\|_{L_t^{-1}}^2}_{U(S_t, A_t, \bar{\theta}_t, L_t)^2} \Big| \mathcal{F}_{t-1}\right]$$

$$\leq \gamma_{T+1}(\delta)\frac{d\kappa^3}{4T}\sum_{t=1}^{T}\mathbb{E}\left[\|\phi(S_t, A_t)\|_{V_t^{-1}}^2 \Big| \mathcal{F}_{t-1}\right],$$

where in the second line we used $\max_{a,\theta \in \mathcal{V}_t} \dot{\mu}(\phi(S_t, a)^{\top}\theta)/\kappa \leq 1$; in the third line we used Corollary 3; in the last line we used $L_t^{-1} \preceq \kappa V_t^{-1}$ and $\dot{\mu}(\cdot) \leq 1/4$ (Eq. (25)).

By Corollary 4, we upper bound the conditional expectations with the realizations

$$(ii) \leq \gamma_{T+1}(\delta)\frac{d\kappa^3}{4T}\left(4\sqrt{\log(\log(2T/\delta))} + \sqrt{\sum_{t=1}^{T}\|\phi(S_t, A_t)\|_{V_t^{-1}}^2}\right)^2$$

$$\leq \gamma_{T+1}(\delta)\frac{d\kappa^3}{2T}\left(16\log(\log(2T/\delta)) + \sum_{t=1}^{T}\|\phi(S_t, A_t)\|_{V_t^{-1}}^2\right)$$

$$\leq \gamma_{T+1}(\delta)\frac{d\kappa^3}{2T}\left(16\log(\log(2T/\delta)) + d\log((d\lambda_T + T)/(d\lambda_T))\right),$$

where in the last line we use the elliptical potential lemma (Lemma 19). Putting the bounds together on $(i)$ and $(ii)$, we have that

$$R_1 \leq \frac{d\gamma_{T+1}(\delta)\sqrt{\log((d\lambda_T + T)/(d\lambda_T))}}{2\sqrt{T}} + \frac{\kappa\gamma_{T+1}(\delta)^2 d^{3/2}\lambda_T \log((d\lambda_T + T)/(d\lambda_T))}{T}$$

$$+ \frac{8\sqrt{d}\log(\log(2T/\delta))}{T} + \gamma_{T+1}(\delta)\frac{d\kappa^3}{2T}\left(16\log(\log(2T/\delta)) + d\log((d\lambda_T + T)/(d\lambda_T))\right)$$

We now deal with $R_2$.

$$\mathbb{E}\left[\max_a \ddot{\mu}(\xi_a)|\phi(S_{T+1}, a)^{\top}(\theta_* - \tilde{\theta}_{T+1}^{\mathrm{Log}})|^2 \Big| \mathcal{F}_T\right]$$

$$\leq \frac{1}{4}\mathbb{E}\left[\max_a \|\phi(S_{T+1}, a)\|_{H_{T+1}^{-1}(\theta_*)}^2 \|\theta_* - \tilde{\theta}_{T+1}^{\mathrm{Log}}\|_{H_{T+1}(\theta_*)}^2 \Big| \mathcal{F}_T\right]$$

$$\leq \frac{1}{4}\gamma_{T+1}^2(\delta)\mathbb{E}\left[\max_a \|\phi(S_{T+1}, a)\|_{H_{T+1}^{-1}(\theta_*)}^2 \Big| \mathcal{F}_T\right]$$

$$\leq \kappa^2\gamma_{T+1}^2(\delta)\mathbb{E}\left[\max_{a,\theta \in \mathcal{V}_{T+1}} \dot{\mu}(\phi(S_{T+1}, a)^{\top}\theta)^2 \|\phi(S_{T+1}, a)\|_{L_{T+1}^{-1}}^2 \Big| \mathcal{F}_T\right]$$

where in the second line we used $|\ddot{\mu}| \leq \dot{\mu} \leq 1/4$ (Eqs. (24) and (25)) and Cauchy-Schwarz; in the third line we used Lemma 17 and that $\tilde{\theta}_{T+1} \in \mathcal{E}_{T+1}(\delta)$; in the fourth line we used the definition of

$\mathcal{V}_{T+1}$. We now use Lemma 16. To be more specific, we let $K = L_t$, $K' = L_{T+1}$ and $\mathcal{Y}' = \mathcal{V}_{T+1}$, $\mathcal{Y} = \mathcal{V}_t$ for $1 \le t \le T$. Since $\mathcal{V}_{T+1} = (\cap_{i=t+1}^{T+1} \mathcal{E}_i) \cap \mathcal{V}_t$ and $L_{T+1} = L_t + \sum_{i=t}^{T} \dot{\mu}(\phi_i^\top \theta_i') \phi_i \phi_i^\top$ where $\phi_i = \phi(S_i, A_i)$, the conditions of Lemma 16 are satisfied. Then running the same argument as Eqs. (33) and (34),

$$\kappa^2 \gamma_{T+1}^2(\delta) \mathbb{E} \left[ \max_{a, \theta \in \mathcal{V}_{T+1}} \dot{\mu}(\phi(S_{T+1}, a)^\top \theta)^2 \| \phi(S_{T+1}, a) \|_{L_{T+1}^{-1}}^2 \Big| \mathcal{F}_T \right]$$

$$\le \kappa^2 \gamma_{T+1}^2(\delta) \frac{1}{T} \sum_{t=1}^{T} \mathbb{E} \left[ \max_{a, \theta \in \mathcal{V}_t} \dot{\mu}(\phi(S_t, a)^\top \theta)^2 \| \phi(S_{T+1}, a) \|_{L_t^{-1}}^2 \Big| \mathcal{F}_{t-1} \right]$$

$$= \kappa^2 \gamma_{T+1}^2(\delta) \frac{1}{T} \sum_{t=1}^{T} \mathbb{E} \left[ \dot{\mu}(\phi(S_t, A_t^{\mathrm{MU}})^\top \theta_t^{\mathrm{MU}})^2 \| \phi(S_t, A_t^{\mathrm{MU}}) \|_{L_t^{-1}}^2 \Big| \mathcal{F}_{t-1} \right]$$

$$\le \underbrace{2\kappa^2 \gamma_{T+1}^2(\delta) \frac{1}{T} \sum_{t=1}^{T} \mathbb{E} \left[ \dot{\mu}(\phi(S_t, A_t^{\mathrm{MU}})^\top \omega_t^{\mathrm{MU}})^2 \| \phi(S_t, A_t^{\mathrm{MU}}) \|_{L_t^{-1}}^2 \Big| \mathcal{F}_{t-1} \right]}_{(iii)}$$

$$+ \underbrace{\kappa^2 \gamma_{T+1}^2(\delta) \frac{1}{T} \sum_{t=1}^{T} \mathbb{E} \left[ \frac{1}{2} \left| \phi(S_t, A_t^{\mathrm{MU}})^\top (\theta_t^{\mathrm{MU}} - \omega_t^{\mathrm{MU}}) \right|^2 \| \phi(S_t, A_t^{\mathrm{MU}}) \|_{L_t^{-1}}^2 \Big| \mathcal{F}_{t-1} \right]}_{(iv)},$$

where in the third line we used definition of $A_t^{\mathrm{MU}}$ and $\theta_t^{\mathrm{MU}}$ (c.f. Eq. (37)). We then do Taylor expansion on $\dot{\mu}(\phi(S_t, A_t^{\mathrm{MU}})^\top \theta_t^{\mathrm{MU}})$ at $\dot{\mu}(\phi(S_t, A_t^{\mathrm{MU}})^\top \omega_t^{\mathrm{MU}})$. For a $\xi_t := \xi(S_t, A_t^{\mathrm{MU}}, \theta_t^{\mathrm{MU}}, \omega^{\mathrm{MU}})$ that is between $\phi(S_t, A_t^{\mathrm{MU}})^\top \omega_t^{\mathrm{MU}}$ and $\phi(S_t, A_t^{\mathrm{MU}})^\top \theta_t^{\mathrm{MU}}$

$$\dot{\mu}(\phi(S_t, A_t^{\mathrm{MU}})^\top \theta_t^{\mathrm{MU}})^2 = \left( \dot{\mu}(\phi(S_t, A_t^{\mathrm{MU}})^\top \omega_t^{\mathrm{MU}}) + \ddot{\mu}(\xi_t) \cdot \left| \phi(S_t, A_t^{\mathrm{MU}})^\top (\theta_t^{\mathrm{MU}} - \omega_t^{\mathrm{MU}}) \right| \right)^2$$

$$\le \left( \dot{\mu}(\phi(S_t, A_t^{\mathrm{MU}})^\top \omega_t^{\mathrm{MU}}) + \frac{1}{4} \cdot \left| \phi(S_t, A_t^{\mathrm{MU}})^\top (\theta_t^{\mathrm{MU}} - \omega_t^{\mathrm{MU}}) \right| \right)^2$$

$$\le 2\dot{\mu}(\phi(S_t, A_t^{\mathrm{MU}})^\top \omega_t^{\mathrm{MU}})^2 + 2 \cdot \frac{1}{16} \left| \phi(S_t, A_t^{\mathrm{MU}})^\top (\theta_t^{\mathrm{MU}} - \omega_t^{\mathrm{MU}}) \right|^2,$$

where in the second line we used that $|\ddot{\mu}| \le \dot{\mu} \le \frac{1}{4}$ (Eqs. (24) and (25)) and the two terms in the first line are non-negative; in the third line we used $(a + b)^2 \le 2a^2 + 2b^2$. Hence,

$$\kappa^2 \gamma_{T+1}^2(\delta) \frac{1}{T} \sum_{t=1}^{T} \mathbb{E} \left[ \dot{\mu}(\phi(S_t, A_t^{\mathrm{MU}})^\top \theta_t^{\mathrm{MU}})^2 \| \phi(S_t, A_t^{\mathrm{MU}}) \|_{L_t^{-1}}^2 \Big| \mathcal{F}_{t-1} \right]$$

$$\le \underbrace{2\kappa^2 \gamma_{T+1}^2(\delta) \frac{1}{T} \sum_{t=1}^{T} \mathbb{E} \left[ \dot{\mu}(\phi(S_t, A_t^{\mathrm{MU}})^\top \omega_t^{\mathrm{MU}})^2 \| \phi(S_t, A_t^{\mathrm{MU}}) \|_{L_t^{-1}}^2 \Big| \mathcal{F}_{t-1} \right]}_{(iii)}$$

$$+ \underbrace{\kappa^2 \gamma_{T+1}^2(\delta) \frac{1}{T} \sum_{t=1}^{T} \mathbb{E} \left[ \frac{1}{2} \left| \phi(S_t, A_t^{\mathrm{MU}})^\top (\theta_t^{\mathrm{MU}} - \omega_t^{\mathrm{MU}}) \right|^2 \| \phi(S_t, A_t^{\mathrm{MU}}) \|_{L_t^{-1}}^2 \Big| \mathcal{F}_{t-1} \right]}_{(iv)},$$

where in the last line we upper bound $1/8$ by $1/2$. For $(iii)$,

$$(iii) \leq 2d\kappa^2\gamma_{T+1}^2(\delta)\frac{1}{T}\sum_{t=1}^{T}\mathbb{E}\left[\dot{\mu}(\phi(S_t,A_t)^\top\bar{\theta}_t)^2\|\phi(S_t,A_t)\|_{L_t^{-1}}^2\Big|\mathcal{F}_{t-1}\right]$$

$$\leq \frac{1}{8}d\kappa^3\gamma_{T+1}^2(\delta)\frac{1}{T}\sum_{t=1}^{T}\mathbb{E}\left[\|\phi(S_t,A_t)\|_{V_t^{-1}}^2\Big|\mathcal{F}_{t-1}\right]$$

$$\leq \frac{1}{8}d\kappa^3\gamma_{T+1}^2(\delta)\frac{1}{T}\left(4\sqrt{\log(\log(2T/\delta))}+\sqrt{\sum_{t=1}^{T}\|\phi(S_t,A_t)\|_{V_t^{-1}}^2}\right)^2 \qquad (42)$$

$$\leq \frac{1}{4}d\kappa^3\gamma_{T+1}^2(\delta)\frac{1}{T}\left(16\log(\log(2T/\delta))+\sum_{t=1}^{T}\|\phi(S_t,A_t)\|_{V_t^{-1}}^2\right) \qquad (43)$$

$$\leq d\kappa^3\gamma_{T+1}^2(\delta)\frac{1}{T}\left(16\log(\log(2T/\delta))+d\log((d\lambda_T+T)/(d\lambda_T))\right), \qquad (44)$$

where in the first line we used that $\min_{(s,a)\in\mathcal{Z},\theta\in\mathbb{B}_d(S)}\dot{\mu}(\phi(s,a)^\top\theta)\cdot\kappa\geq 1$; in the second line we used $L_t^{-1}\preceq\kappa V_t^{-1}$; in the third line we used Corollary 4; in the fourth line we used $(a+b)^2\leq 2a^2+2b^2$; in the fifth line we used elliptical potential lemma (Lemma 19) to bound the second term. For $(iv)$, we apply Cauchy-Schwarz inequality and Lemma 17 to get

$$(iv) \leq \kappa^2\gamma_{T+1}^2(\delta)\frac{1}{T}\sum_{t=1}^{T}\frac{1}{2}\mathbb{E}\left[\|\phi(S_t,A_t^{\mathrm{MU}})\|_{L_t^{-1}}^2\|\theta_t^{\mathrm{MU}}-\omega_t^{\mathrm{MU}}\|_{L_t}^2\cdot\|\phi(S_t,A_t^{\mathrm{MU}})\|_{L_t^{-1}}^2\Big|\mathcal{F}_{t-1}\right]$$

$$\leq \kappa^2\gamma_{T+1}^2(\delta)\frac{1}{T}\sum_{t=1}^{T}\frac{1}{2}\mathbb{E}\left[\|\phi(S_t,A_t^{\mathrm{MU}})\|_{L_t^{-1}}^4\|\theta_t^{\mathrm{MU}}-\omega_t^{\mathrm{MU}}\|_{H_t(\theta_*)}^2\Big|\mathcal{F}_{t-1}\right]$$

$$\leq 8\kappa^2\gamma_{T+1}^4(\delta)\frac{1}{T}\sum_{t=1}^{T}\mathbb{E}\left[\|\phi(S_t,A_t^{\mathrm{MU}})\|_{L_t^{-1}}^4\Big|\mathcal{F}_{t-1}\right].$$

Then we use the fact that $\min_{(s,a)\in\mathcal{Z},\theta\in\mathbb{B}_d(S)}\dot{\mu}(\phi(s,a)^\top\theta)\cdot\kappa\geq 1$,

$$8\kappa^2\gamma_{T+1}^4(\delta)\frac{1}{T}\sum_{t=1}^{T}\mathbb{E}\left[\|\phi(S_t,A_t^{\mathrm{MU}})\|_{L_t^{-1}}^4\Big|\mathcal{F}_{t-1}\right]$$

$$\leq 8\kappa^4\gamma_{T+1}^4(\delta)\frac{1}{T}\sum_{t=1}^{T}\mathbb{E}\left[\dot{\mu}(\phi(S_t,A_t^{\mathrm{MU}})^\top\omega_t^{\mathrm{MU}})^2\|\phi(S_t,A_t^{\mathrm{MU}})\|_{L_t^{-1}}^4\Big|\mathcal{F}_{t-1}\right]$$

$$\leq \frac{8d\kappa^4}{\lambda_1^2}\gamma_{T+1}^4(\delta)\frac{1}{T}\sum_{t=1}^{T}\mathbb{E}\left[\dot{\mu}(\phi(S_t,A_t)^\top\bar{\theta}_t)^2\|\phi(S_t,A_t)\|_{L_t^{-1}}^2\Big|\mathcal{F}_{t-1}\right]$$

$$\leq \frac{8d\kappa^5}{8\lambda_1^2}\gamma_{T+1}^4(\delta)\frac{1}{T}\sum_{t=1}^{T}\mathbb{E}\left[\|\phi(S_t,A_t)\|_{V_t^{-1}}^2\Big|\mathcal{F}_{t-1}\right]$$

$$\leq \frac{d\kappa^5}{\lambda_1^2}\gamma_{T+1}^4(\delta)\frac{1}{T}\left(16\log(\log(2T/\delta))+d\log((d\lambda_T+T)/(d\lambda_T))\right),$$

where in the third line we used Corollary 3;l in the fourth line we used $L_t^{-1}\preceq\kappa V_t^{-1}$ and in the last line we used similar argument as (iii) (Eqs. (42) to (44)).

Putting all the bounds on $R_2$ together, we have that

$$R_2 \leq d\kappa^3\gamma_{T+1}^2(\delta)\frac{1}{T}\left(16\log(\log(2T/\delta))+d\log((d\lambda_T+T)/(d\lambda_T))\right)$$

$$+\frac{d\kappa^5}{\lambda_1^2}\gamma_{T+1}^4(\delta)\frac{1}{T}\left(16\log(\log(2T/\delta))+d\log((d\lambda_T+T)/(d\lambda_T))\right)$$

And the simple regret,

$$\mathfrak{R}(\hat{\pi}_{T+1}) \leq 2D_{\text{Log}}$$

$$\leq 2(R_1 + R_2)$$

$$\leq \frac{d\gamma_{T+1}(\delta)\sqrt{\log((d\lambda_T + T)/(d\lambda_T))}}{\sqrt{T}} + \frac{\kappa\gamma_{T+1}(\delta)^2 d^{3/2}\lambda_T \log((d\lambda_T + T)/(d\lambda_T))}{T}$$

$$+ \frac{16\sqrt{d}\log(\log(2T/\delta))}{T}$$

$$+ \frac{2}{T}\left(d\kappa^3\gamma_{T+1}(\delta) + d\kappa^3\gamma_{T+1}^2(\delta) + \frac{d\kappa^5}{\lambda_1^2}\gamma_{T+1}^4(\delta)\right) \cdot (16\log(\log(2T/\delta)) + d\log((d\lambda_T + T)/(d\lambda_T)))$$

## F   TECHNICAL LEMMAS

In Theorems 9 and 10, we provided a corrected version of Thm. 3 in Zanette et al. (2021). The reason why it's flawed is: the optimization over $\lambda$ in eq 133 and 134 depends on a random variable, therefore one cannot "choose" without the knowledge of the random quantity. We rectify the situation by forming a geometric cover over possible values for $\lambda$, which solved the issue but introduced a second logarithmic term.

**Theorem 9** (Bernstein's inequality for Martingales). *Consider the stochastic process $\{X_t\}$ adapted to the filteration $\{\mathcal{F}_t\}$. Assume $X_t \leq 1$ almost surely, and $\mathbb{E}[X_t|\mathcal{F}_{t-1}] = 0$. Then*

$$\forall \lambda \in (0,1], \qquad P\left(\sum_{t=1}^T X_t \leq \lambda \sum_{t=1}^T \mathbb{E}[X_t^2|\mathcal{F}_{t-1}] + \frac{1}{\lambda}\log\frac{1}{\delta}\right) \geq 1 - \delta, \qquad (45)$$

*which implies*

$$P\left(\sum_{t=1}^T X_t \leq 3\sqrt{\left(\sum_{t=1}^T \mathbb{E}[X_t^2|\mathcal{F}_{t-1}]\right)\log(\frac{\lg(\sqrt{T})}{\delta}) + 2\log(\frac{\lg(\sqrt{T})}{\delta})}\right) \geq 1 - \delta. \qquad (46)$$

*Proof.* Define the random variable $M_t$ as

$$M_t = M_{t-1}\exp(\lambda X_t - \lambda^2\mathbb{E}[X_t^2|\mathcal{F}_{t-1}]), \qquad (47)$$

where in particular $M_0 = 1$, and $\mathbb{E}[\cdot|\mathcal{F}_0] = \mathbb{E}[\cdot]$ so $M_t$ is $\mathcal{F}_t$-measurable. Recall the inequalities $e^x \leq 1 + x + x^2$ for $x \leq 1$ and $1 + x \leq e^x$:

$$\mathbb{E}[M_t|\mathcal{F}_{t-1}] = M_{t-1}\mathbb{E}\left[\exp(\lambda X_t - \lambda^2\mathbb{E}[X_t^2|\mathcal{F}_{t-1}])|\mathcal{F}_{t-1}\right] \qquad (48)$$

$$\leq M_{t-1}\mathbb{E}\left[1 + \lambda X_t + \lambda^2 X_t^2|\mathcal{F}_{t-1}\right]\exp(-\lambda^2\mathbb{E}[X_t^2|\mathcal{F}_{t-1}]) \qquad (49)$$

$$\leq M_{t-1}\left(1 + \lambda\mathbb{E}[X_t|\mathcal{F}_{t-1}] + \lambda^2\mathbb{E}\left[X_t^2|\mathcal{F}_{t-1}\right]\right)\exp(-\lambda^2\mathbb{E}[X_t^2|\mathcal{F}_{t-1}]) \qquad (50)$$

$$\leq M_{t-1}\exp(\lambda^2\mathbb{E}[X_t^2|\mathcal{F}_{t-1}])\exp(-\lambda^2\mathbb{E}[X_t^2|\mathcal{F}_{t-1}]) \qquad (51)$$

$$= M_{t-1}. \qquad (52)$$

Thus, $\{M_t\}$ is a supermartingale adapted to $\{\mathcal{F}_t\}$. In particular $\mathbb{E}[M_t|\mathcal{F}_{t-1}] \leq M_0 = 1$. Then by the Markov inequality:

$$P\left(\underbrace{\lambda\sum_{t=1}^T X_t - \lambda^2\sum_{t=1}^T \mathbb{E}[X_t^2|\mathcal{F}_{t-1}]}_{\log(M_t)} > \log\frac{1}{\delta}\right) = P\left(M_t > \frac{1}{\delta}\right) \leq \frac{\mathbb{E}\left[\mathbb{E}[M_t|\mathcal{F}_{t-1}]\right]}{\frac{1}{\delta}} \leq \delta, \qquad (53)$$

which proves Eq. (45).

Next, to prove Eq. (46) define the sequence $N(l) := \{\lambda_i = l2^i\}_{i=0}^{\lfloor \lg(1/l)\rfloor} \cup \{1\}$ for a value $l \leq 1$ chosen later, and

$$\hat{\lambda} = \sqrt{\frac{\log(\frac{\lfloor \lg(1/l)\rfloor + 1}{\delta})}{\sum_{t=1}^T \mathbb{E}[X_t^2|\mathcal{F}_{t-1}]}}. \qquad (54)$$

Also, by using Eq. (53) and a union bound over the $\lfloor \lg(1/l) \rfloor + 1$ points in $N(l)$ we get:

$$P\left(\forall \lambda \in N(l): \quad \lambda \sum_{t=1}^{T} X_t - \lambda^2 \sum_{t=1}^{T} \mathbb{E}[X_t^2|\mathcal{F}_{t-1}] \leq \log \frac{\lfloor \lg(\frac{1}{l}) \rfloor + 1}{\delta}\right) \geq 1 - \delta. \quad (55)$$

Firstly, if $1 \leq \hat{\lambda}$, which also means $\sum_{t=t}^{T} \mathbb{E}[X_t^2|\mathcal{F}_{t-1}] \leq \log(\frac{\lfloor \lg(1/l) \rfloor + 1}{\delta})$, for value $\lambda = 1 \in N(l)$ in Eq. (55) we get:

$$\sum_{t=1}^{T} X_t \leq \sum_{t=1}^{T} \mathbb{E}[X_t^2|\mathcal{F}_{t-1}] + \log(\frac{\lfloor \lg(1/l) \rfloor + 1}{\delta}) \leq 2\log(\frac{\lfloor \lg(1/l) \rfloor + 1}{\delta}). \quad (56)$$

Secondly, for $\hat{\lambda} < 1$, according $N(l)$'s construction one of the two cases below holds:

$$\hat{\lambda} < l \quad \text{or} \quad \exists \tilde{\lambda} \in N(l) \text{ st. } \tilde{\lambda} \leq \hat{\lambda} \leq 2\tilde{\lambda}. \quad (57)$$

For $\hat{\lambda} \geq l$ by Eq. (55) and $\tilde{\lambda}$ defined in Eq. (57) we have

$$\sum_{t=1}^{T} X_t \leq \tilde{\lambda} \sum_{t=1}^{T} \mathbb{E}[X_t^2|\mathcal{F}_{t-1}] + \frac{1}{\tilde{\lambda}} \log(\frac{\lfloor \lg(1/l) \rfloor + 1}{\delta}) \quad (58)$$

$$\leq \hat{\lambda} \sum_{t=1}^{T} \mathbb{E}[X_t^2|\mathcal{F}_{t-1}] + \frac{2}{\hat{\lambda}} \log(\frac{\lfloor \lg(1/l) \rfloor + 1}{\delta}) \quad (59)$$

$$\leq 3\sqrt{\left(\sum_{t=1}^{T} \mathbb{E}[X_t^2|\mathcal{F}_{t-1}]\right) \log(\frac{\lfloor \lg(1/l) \rfloor + 1}{\delta})} \quad (60)$$

For $\hat{\lambda} < l$, which means $\log(\frac{\lfloor \lg(1/l) \rfloor + 1}{\delta}) < l^2 \sum_{t=1}^{T} \mathbb{E}[X_t^2|\mathcal{F}_{t-1}]$, by Eq. (55) we have

$$\sum_{t=1}^{T} X_t \leq l \sum_{t=1}^{T} \mathbb{E}[X_t^2|\mathcal{F}_{t-1}] + \frac{1}{l} \log(\frac{\lfloor \lg(1/l) \rfloor + 1}{\delta}) \quad (61)$$

$$\sum_{t=1}^{T} X_t \leq 2l \sum_{t=1}^{T} \mathbb{E}[X_t^2|\mathcal{F}_{t-1}] \quad (62)$$

Finally, by setting $l = \frac{1}{\sqrt{T}}$, the fact that $\sum_{t=1}^{T} \mathbb{E}[X_t^2|\mathcal{F}_{t-1}] \leq T$, and summing up with RHS of Eq. (56) to cover both cases we get:

$$P\left(\sum_{t=1}^{T} X_t \leq 3\sqrt{\left(\sum_{t=1}^{T} \mathbb{E}[X_t^2|\mathcal{F}_{t-1}]\right) \log(\frac{\lg(\sqrt{T})}{\delta})} + 2\log(\frac{\lg(\sqrt{T})}{\delta})\right) \geq 1 - \delta, \quad (63)$$

which proves the second part of the thesis. $\square$

**Theorem 10** (Reversed Bernstein's inequality for Martingales). *Let $\{X_t\}$ be a stochastic process adapted to the filteration $\{\mathcal{F}_t\}$. Assuming $0 \leq X_t \leq 1$ almost surely, then it holds that:*

$$P\left(\sum_{t=1}^{T} \mathbb{E}[X_t|\mathcal{F}_{t-1}] \leq \frac{1}{4}\left(c_1 + \sqrt{c_1^2 + 4\left(\sum_{t=1}^{T} X_t + c_2\right)}\right)^2\right) \geq 1 - \delta, \quad (64)$$

$$c_1 = 3\sqrt{\log \frac{\lg(\sqrt{T})}{\delta}}, c_2 = 2\log \frac{\lg(\sqrt{T})}{\delta}. \quad (65)$$

*Proof.* Consider the random noise

$$\xi_t := \mathbb{E}[X_t|\mathcal{F}_{t-1}] - X_t, \quad (66)$$

which allows us to write

$$\sum_{i=1}^{t} \mathbb{E}[X_i|\mathcal{F}_{i-1}] = \sum_{i=1}^{t} (\xi_i + X_i). \tag{67}$$

Then the Theorem 9 ensures the following statement:

$$P\left(\sum_{t=1}^{T} \xi_t \leq c_1 \sqrt{\sum_{t=1}^{T} \mathbb{E}[\xi_t^2|\mathcal{F}_{t-1}]} + c_2\right) \geq 1 - \delta. \tag{68}$$

Notice that since $0 \leq X_t \leq 1$ almost surely, we have

$$\mathbb{E}[\xi_t^2|\mathcal{F}_{t-1}] = \mathbb{E}[(X_t - \mathbb{E}[X_t|\mathcal{F}_{t-1}])^2|\mathcal{F}_{t-1}] \tag{69}$$
$$= \mathbb{E}[X_t^2|\mathcal{F}_{t-1}] - \mathbb{E}[X_t|\mathcal{F}_{t-1}]^2 \tag{70}$$
$$\leq \mathbb{E}[X_t^2|\mathcal{F}_{t-1}] \tag{71}$$
$$\leq \mathbb{E}[X_t|\mathcal{F}_{t-1}]. \tag{72}$$

Plugging back into Eq. (68) and using Eq. (67) gives

$$P\left(\sum_{t=1}^{T} \xi_t = \sum_{t=1}^{T}(\mathbb{E}[X_t|\mathcal{F}_{t-1}] - X_t) \leq c_1 \sqrt{\sum_{t=1}^{T} \mathbb{E}[X_t|\mathcal{F}_{t-1}]} + c_2\right) \geq 1 - \delta \tag{73}$$

or equivalently

$$P\left(\sum_{t=1}^{T} \mathbb{E}[X_t|\mathcal{F}_{t-1}] \leq \sum_{t=1}^{T} X_t + c_1 \sqrt{\sum_{t=1}^{T} \mathbb{E}[X_t|\mathcal{F}_{t-1}]} + c_2\right) \geq 1 - \delta. \tag{74}$$

Solving for $\sum_{t=1}^{T} \mathbb{E}[X_t|\mathcal{F}_{t-1}]$ gives under such event

$$\sum_{t=1}^{T} \mathbb{E}[X_t|\mathcal{F}_{t-1}] \leq \frac{1}{4}\left(c_1 + \sqrt{c_1^2 + 4\left(\sum_{t=1}^{T} X_t + c_2\right)}\right)^2. \tag{75}$$

$\square$

The following is a corollary to the above theorem, which is easier to use.

**Corollary 4.** *Under the same assumptions of Theorem 10, with probability at least $1 - \delta$ we have*

$$\sum_{t=1}^{T} \mathbb{E}[X_t|\mathcal{F}_{t-1}] \leq \left(4\sqrt{\log(\frac{2\log T}{\delta})} + \sqrt{\sum_{t=1}^{T} X_t}\right)^2. \tag{76}$$

*Proof.* Starting from the right hand side of the inequality in Eq. (64), using the fact that $\sqrt{a+b} \leq \sqrt{a} + \sqrt{b}$ for $a, b \geq 0$ we have:

$$\frac{1}{4}\left(c_1 + \sqrt{c_1^2 + 4\left(\sum_{t=1}^{T} X_t + c_2\right)}\right)^2 \leq \frac{1}{4}\left(6\sqrt{\log\left(\frac{\lg\sqrt{T}}{\delta}\right)} + 2\sqrt{\sum_{t=1}^{T} X_t + 2\sqrt{c_2}}\right)^2 \quad (77)$$

$$\leq \frac{1}{4}\left(6\sqrt{\log\left(\frac{\lg\sqrt{T}}{\delta}\right)} + 2\sqrt{\sum_{t=1}^{T} X_t + 2\sqrt{2\log\left(\frac{\lg\sqrt{T}}{\delta}\right)}}\right)^2 \quad (78)$$

$$\leq \left(4\sqrt{\log\left(\frac{\lg\sqrt{T}}{\delta}\right)} + \sqrt{\sum_{t=1}^{T} X_t}\right)^2 \quad (79)$$

$$\leq \left(4\sqrt{\log\left(\frac{\frac{1}{2}\log T}{\log 2 \cdot \delta}\right)} + \sqrt{\sum_{t=1}^{T} X_t}\right)^2 \quad (80)$$

$$\leq \left(4\sqrt{\log\left(\frac{2\log T}{\delta}\right)} + \sqrt{\sum_{t=1}^{T} X_t}\right)^2 \quad (81)$$

$$(82)$$

**Lemma 19** (Elliptical potential lemma). *Fix $\lambda, A > 0$. Let $\{a_t\}_{t=1}^{\infty}$ be a sequence in $AB_2^d$ and let $V_0 = \lambda I$. Define $V_{t+1} = V_t + a_{t+1}a_{t+1}^\top$ for each $t \in \mathbb{N}$. Then, for all $n \in \mathbb{N}^+$,*

$$\sum_{t=1}^{n} ||a_t||_{V_{t-1}^{-1}}^2 \leq 2d \max\left\{1, \frac{A^2}{\lambda}\right\} \log\left(1 + \frac{nA^2}{d\lambda}\right).$$

*Proof.* See, e.g., Lemma 19.4 of Lattimore & Szepesvári (2020). □

□

**Proposition 1.** *Let $X$ be a bounded random variable, $Y$ be an integrable random variable and $\mathcal{G}$ be a $\sigma$-algebra such that $X$ is $\mathcal{G}$-measurable. Then it holds almost surely that*

$$\mathbb{E}[XY|\mathcal{G}] = X\mathbb{E}[Y|\mathcal{G}].$$

*Proof.* See e.g. section XI.3.(h) of Doob (2012). □

**Proposition 2.** *If $X \sim \mathcal{N}(0, I)$, then $\frac{X}{\|X\|_2} \sim \mathrm{Unif}(\mathbb{S}^{d-1})$.*

**Proposition 3.** *If $X \sim \mathrm{Unif}(\mathbb{S}^{d-1})$ and $u$ is any fixed unit vector, then it follows that*

$$\mathbb{E}[|\langle X, u\rangle|] \geq \sqrt{\frac{2}{\pi d}}.$$

**Proposition 4.** *If $X \sim \mathrm{Unif}(\mathbb{S}^{d-1})$ and $u$ is any fixed unit vector, then it follows that*

$$\mathbb{E}\left[(\langle X, u\rangle)^2\right] = \frac{1}{d}.$$

## G DETAILS AND RESULTS OF EXPERIMENTS

In this section, we provide more details on the experimental setup and additional results to complement those presented in Section 4. We follow the structure of Section 4, i.e., introduce the setup and provide the observed results as well as interpretations first for linear case then logistic case.

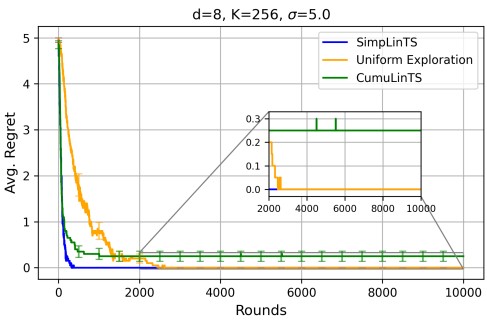 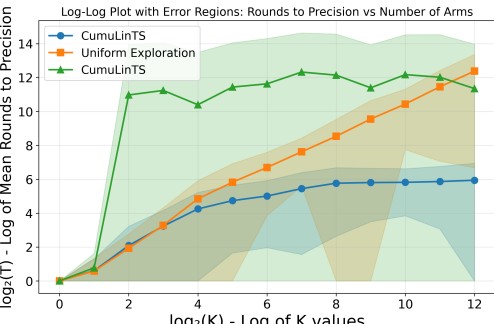

Figure 1: Linear case. The noise is $0$-mean Gaussian with std. dev. $\|\theta_*\|$. Left: Average simple regret vs. number of rounds $T$ for $d = 8, K = 128$. Right: Number of rounds needed to make the simple regret fall below $0.1$ vs. $K$.

### G.1 LINEAR CASE

**Setup** The goal is to show that uniform exploration is expected to perform arbitrarily worse than our SIMPLELINTS and CumuLinTS does not converge as fast as SIMPLELINTS due to its nature of balancing exploration and exploitation, illustrating the necessity of tailoring algorithms to the simple regret objective. We achieve this in the following way: The arm set is the unit sphere in $d = 8$ dimensions. In each round, the learner is shown $K$ arms where $K$ is the parameter that we vary. We design the environment to be adversarial to the uniform sampling algorithm to show that naive approaches would fail in certain scenarios while ours would not. To be more specific, $\theta_* = (5, 0, \dots, 0)$. One of the arms shown is always the optimal arm, $x^\star = (1, 0, \dots, 0)$, while the remaining arms are sampled at random from the uniform distribution restricted to the intersection of the unit sphere and the set of vectors orthogonal to $x^*$. The noise on the reward is Gaussian with standard deviation $\|\theta_*\|$.

**Results and Interpretation** Results are obtained by running all algorithms $100$ times with independent randomness between the runs. We report the average simple regret as a function of the number of rounds $T$ for $K = 128$ and report in Fig. 1 on the number of rounds $T$ needed to make the simple regret fall below $0.1$ for $K = 2, 4, \dots, 4096$. We observe that for $K = 128$, even though CumuLinTS is converging fast at the beginning, it hovers above $0$ for a long time. Intuitively the algorithm needs to balance exploration and exploitation, which causes a slowed-down pace on exploration after it has already found a good arm. This is not the case for SIMPLELINTS which keeps exploring until the end. The uniform exploration baseline is doing poorly as expected. Moreover, we observe that the number of rounds needed for uniform exploration to reach $0.1$ simple regret grows linearly with $K$ while SIMPLELINTS remains constant for $K \geq 64$. Intuitively, this is because uniform exploration with $K \to \infty$ will spend all its time on exploring the suboptimal arms, ignoring the optimal arm, whose estimate is only very slowly refined as a result. We test our algorithm in an environment designed to make naive UE fail. The optimal arm $x_* = e_1$ for $\theta_* = 5 \cdot e_1$ is always available in the action set in each round, where $e_i$ is the $i$-th Euclidean basis, while suboptimal arms are sampled randomly orthogonal to it on the unit sphere. Results from $100$ independent runs, reported in Fig. 1, show that for $K = 128$, CumuLinTS's convergence stalls due to its exploration-exploitation trade-off, unlike our purely exploring SIMPLELINTS. The number of rounds needed for UE's simple regret to fall below $0.1$ scales linearly with $K$ as UE increasingly wastes trials on the growing set of suboptimal arms. In contrast, the requirement of SIMPLELINTS is constant for $K \geq 64$. The number of rounds for CumuLinTS also does not scale with $K$ but it has a much larger magnitude than SIMPLELINTS.

**Implementation details** We set the regularization parameter $\lambda = 1$ for both SIMPLELINTS and CumuLinTS. For the UE baseline, in each round, the learner randomly selects one of the $K$ arms uniformly to pull, and does ridge regression on the history data to estimate $\theta_*$ and the output policy is greedy w.r.t. the estimate after $T$ rounds. For CumuLinTS, we follow the same procedure as SIMPLELINTS except that in each round, the learner samples $\tilde{\theta}_t$ from $\mathcal{N}(\hat{\theta}_t, V_t^{-1})$.

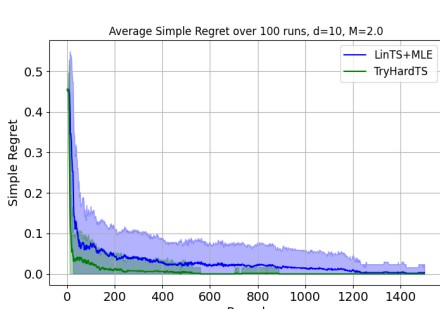 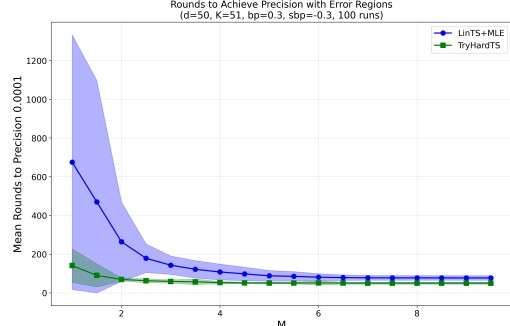

Figure 2: Left: Logistic case where $M = 2$. Average simple regret v.s. number of rounds for $d = 10, T = 1500, M = 2$. Right: Number of rounds needed to make the simple regret fall below $10^{-4}$ v.s. $M$ for $d = 50$.

### G.2 LOGISTIC CASE

**Setup**    The purpose of this experiment is to demonstrate that the trivial extension of SIMPLELINTS to the logistic case leads to poor behavior relative to what is possible with our more complex method, THATS. The extension mentioned simply replaces the least-squares method with MLE. This method will essentially focus on growing the "magnitude" of $V_t$. Hence we design an arm set $\{-e_i\}_{i=1}^{d-1} \cup \{0.3 \cdot e_d, -0.3 \cdot e_d\}$ (the arm set does not change across different rounds) and set $\theta_* = [M, M, \ldots, 1]$. The optimal and second optimal arm are $\pm 0.3 \cdot e_d$ respectively ($\approx 0.57, 0.43$). Similar to the linear case, we first run 100 runs on $M = 2$ (where the suboptimal means $\approx 0.12$) for $T = 1500$ rounds and report the average simple regret and the error of estimating each of the componenta of $\theta_*$. Then we vary $M$ in $\{1 + 0.5 \cdot m\}_{m=1}^{18}$ with $d = 50$ to see how many rounds are needed to make the simple regret fall below $10^{-4}$. The reason why we choose $10^{-4}$ is that in practice, for example, online advertisement clicking, the clicking probability (recall for Bernoulli it is the mean of the distribution) is usually very small: $\sim 10^{-3}$ (Faury et al. (2020)) and a gap of $10^{-4}$ is a relative error of $0.1$ .

**Observations and Interpretations**    Recall that $\kappa \approx \exp(M)$ as $M$ gets large. The optimal arm (after applied an inner product with $\theta_*$) lies in the near-linear, central part of the sigmoid function, i.e., having less curvature, where $\dot{\mu}(\cdot)$ is close to its max, while the suboptimal arms lie in the flat part ($\dot{\mu}$ close to zero). The reward received for $\pm 0.3 \cdot e_d$ will be quite noisy, because the mean (after taking a dot product with $\theta_*$) is close to $1/2$, and not noisy for the other arms. A clever method should thus pull $\pm 0.3 \cdot e_d$ (who also happen to be the best two arms by design) more often than the others to get sufficiently good estimates for separating the arm values. THATS indeed does this, while the extended SIMPLELINTS method will fail to take this into account due to its nature of growing the magnitude of $V_t$ mentioned before.

We also plotted the error of estimating the last component of $\theta_*$ for $M = 2$ and observed that, as expected, the error of THATS vanishes much faster than that of the adapted SIMPLELINTS, while the variance across the runs is also much smaller as demonstrated in Fig. 7. What's more, the estimation of other "unimportant" components of $\theta_*$ (the first $d - 1$ components) by THATS does not converge at all while that of SIMPLELINTS does. This also provides evidence that THATS is focusing its exploration on the important directions only, which is exactly what we want, while SIMPLELINTS wastes its time on these "unimportant" components until it fully rules them out.

As $M$ is varied in $\{1 + 0.1 \cdot m\}_{m=1}^{90}$, for $d = 50$, we observe that for a fixed level of suboptimality, 100 independent runs, the average number of rounds needed for THATS is much less than that of SIMPLELINTS with significantly lower variance as is demonstrateed in Fig. 2.

**Implementation Details**    For THATS, we set $\lambda_{\log} = 1$ and for SIMPLELINTS we set $\lambda_{\lin} = 1$. In order to reduce computational complexity, we set $\bar{\theta}_t$ (the constrained MLE) to be the global MLE $\hat{\theta}_t$ and set $\theta_t'$ to be the projection of $\tilde{\theta}_t$ to the $\ell_2$-ball of radius $S = \|\theta_*\| + 1$.

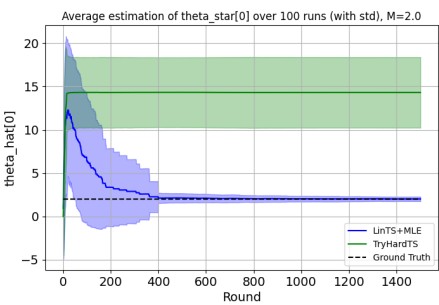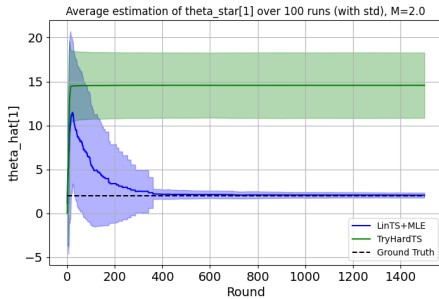

Figure 3: The estimation of "unimportant" components of $\theta_*$ for $M = 2$. Left: $\theta_*[0]$, Right: $\theta_*[1]$.

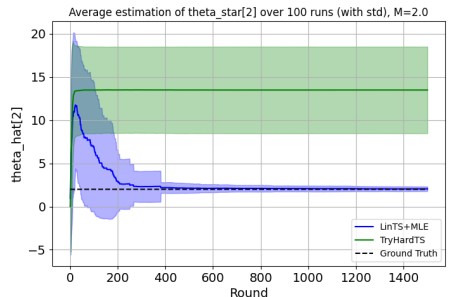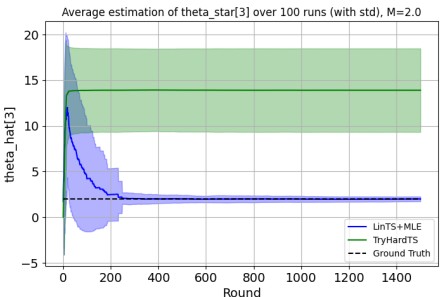

Figure 4: The estimation of "unimportant" components of $\theta_*$ for $M = 2$. Left: $\theta_*[2]$, Right: $\theta_*[3]$.

## H    LLM USAGE DISCLOSURE

**Using an LLM to help with paper writing**    According to ICLR 2026 guidelines on LLM usage disclosure, we describe here our use of LLMs in the preparation of this manuscript. We used Gemini 2.5 Pro to polish on the writing of the numerical experiment section in the main paper. Specifically, we used it to shorten the paragraph so that the main ideas are conveyed more concisely. We also used it to help rephrase some sentences in Appendix D.4 to improve clarity. We also used github copilot to help with writing this section and the README file of the shared code. **We acknowledge that the authors are responsible for the content of the submission.**

**Using an LLM as a research assistant**    We used Claude Opus 4.1 to generate the code for plotting the numerical results in Appendix G. Specifically, we provided it with the implementation of the algorithms and the environment, and asked it to generate the code for plotting the results. We then modified the generated code to suit our needs. **We acknowledge that the authors have validated the generated code and are responsible for the content of the submission.**

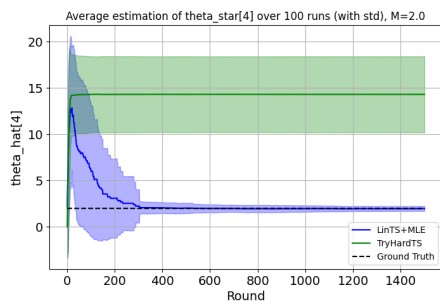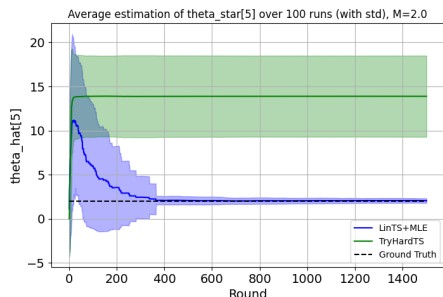

Figure 5: The estimation of "unimportant" components of $\theta_*$ for $M = 2$. Left: $\theta_*[4]$, Right: $\theta_*[5]$.

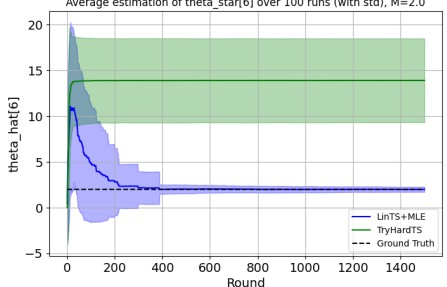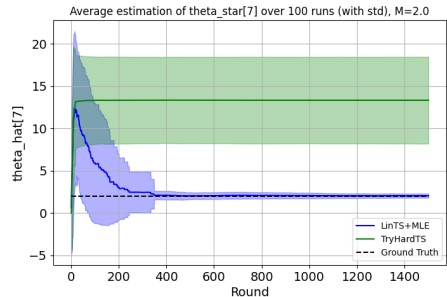

Figure 6: The estimation of "unimportant" components of $\theta_*$ for $M = 2$. Left: $\theta_*[6]$, Right: $\theta_*[7]$.

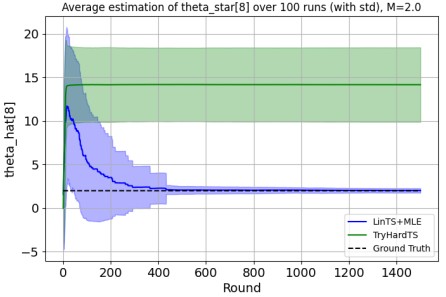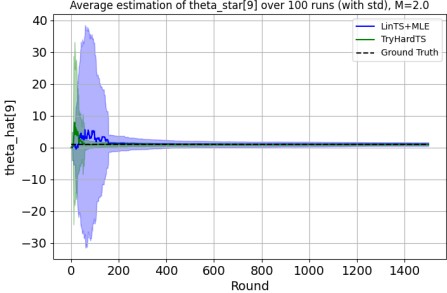

Figure 7: The estimation of components of $\theta_*$ for $M = 2$. Left: $\theta_*[8]$, Right: $\theta_*[9]$ (the "important" component).

