# OpenReview forum: "Efficient Simple Regret Algorithms for Stochastic Contextual Bandits"
_ICLR.cc/2026/Conference — Submitted to ICLR 2026_

### Official Review · Reviewer_oWyi · 2025-10-19

**Soundness:** 2
**Presentation:** 3
**Contribution:** 3
**Rating:** 6
**Confidence:** 3

**Summary:**

This paper considers stochastic contextual bandits under the simple regret objective, studying both the linear and logistic models. The simple regret framework focuses on learning a policy that performs well on future data after a certain exploration horizon, which is different from minimizing cumulative regret during learning.
For the linear model, the paper first presents a deterministic algorithm called MULIN. The algorithm selects actions which maximize predictive uncertainty and achieves a simple regret bound of
$\tilde{O}(d/\sqrt{T})$ where $d$ is the dimension of the feature space and $T$ the number of rounds. They also propose a randomized Thompson Sampling variant which obtains a bound of
$\tilde{O}(d^{3/2} /\sqrt{T})$, matching the dependence on dimension known from cumulative regret settings.

For the logistic model, which introduces nonlinearity through a sigmoid link function, the paper proposes two analogous algorithms. The deterministic version, MULOG, extends the uncertainty-based exploration approach by leveraging self-concordant analysis of the logistic loss. It achieves a simple regret bound of $\tilde{O}(d/\sqrt{T})$ with the leading term free of the exponential constant $\kappa$. The randomized counterpart, THATS, attains a similar bound of
$\tilde{O}(d^{3/2} /\sqrt{T})$ with lower computational cost.

These results are claimed to be the first theoretical guarantees for simple regret minimization in stochastic logistic contextual bandits. The paper develops new analytical techniques to handle randomization and nonlinearity, including a decreasing-uncertainty lemma for the logistic case. Empirical studies on synthetic data corroborate the theoretical findings, showing that the proposed algorithms outperform selected baselines.

**Strengths:**

**Originality**:
This paper introduces the 1st simple-regret analysis for logistic contextual bandits, a nontrivial extension from the linear bandit setting. It develops new analysis tools for randomized algorithms under simple regret, distinct from cumulative regret frameworks. It corrects a known technical mistake in prior work (Zanette et al., 2021) and extends the self-concordant logistic analysis of Faury et al. (2020).

**Quality**:
Theoretical results are mostly solid, with good proofs, detailed derivations, and clear dependence on problem parameters like $T$ and $d$.
The regret bounds seem tight and match known minimax orders in the linear case.
Empirical evaluations, though synthetic, corroborate the theoretical claims and illustrate performance differences between algorithms.

**Clarity**:
The paper has a good clarity. Pseudocode is provided for all four main algorithms.


**Significance**:

This work establishes a foundation for simple regret optimization in nonlinear contextual bandits. This seems an important step toward more challenging generalized linear models.

**Weaknesses:**

There is no major technical flaw detected. But I do have a few concerns.

(1)
The alignment argument used for Thompson sampling assumes isotropy. It is not immediately clear to me why this holds.


(2) The analysis for logistic bandits relies on that the constructed matrices $L_t$ uniformly lower-bound the true curvature. It is not immediately clear to me why this derived property of $L_t$ holds.

Please also see my questions below.

**Questions:**

(1) The paper claims the derived bounds “essentially tight,” but no lower bound for logistic simple regret is provided. Or am I missing something here?

(2) Does the analysis of theorems implicitly rely on independence between the data used to construct $L_t$​ and the estimator $\hat{\theta}_t$?

(3) The regret analysis relies critically on the claim that the constructed matrices $L_t$​ form valid uniform lower bounds on the true Hessians. This is motivated by the minimization of $\mu(\varphi^\top \theta )$ over confidence sets $W_i$​.
I wonder why this inequality holds with high probability for all tt?
In other words, why cannot $\mu(x)$ tend to 0 at a fast rate as $x$ increases? Or is this actually an assumption instead of a derived property?

(4) The regret bound for the Thompson Sampling–based algorithms (SIMPLELINTS in Theorem 3 and THATS in Theorem 4) relies on a dimension-only alignment guarantee inherited from the linear case. However, in the logistic setting, it is unclear why $L_t$ is still isotropic? If it is not, then wouldn’t this break the analysis?

---

> ### Author Response · Authors · 2025-11-21
> **Rebuttal to reviewer oWyi**
>
> We thank reviewer oWyi for careful inspection of our paper and thought-provoking questions.
> Here are the responses to the weakness section:
>
> 1. **Alignment argument assumes isotropy.** We do not assume isotropy. Instead, we conduct a “noise-whitening” step in the proof. For example,  line 734, the proof of the simpler case: contextual linear bandits. To be more specific, our $\tilde\theta_t \sim \mathcal N(0,V_t^{-1})$. Then by properties of Gaussian distributions, $V_t^{1/2}\tilde\theta_t$ follows a standard Gaussian distribution.
>
> 2. **How does L_t uniformly lower bound the true curvature**. $W_i$ is the intersection of confidence sets constructed in all the rounds before $t$. By construction, with high probability the optimal parameter lies in all of the confidence sets (Lemma 2), hence it lies in the intersection of them. Minimizing the curvature over $W_i$ certainly would result in a lower bound of the true curvature.
>
> Here are the responses to the question section:
> 1. **"Claim the bounds are tight while no lower bound is presented."** It is right that there is no lower bound presented. Our claim is an educated guess from the simple regret of linear contextual bandits. To be more specific, the complexity of logistic bandits is close to that of linear bandits except for the curvature parameter $\kappa$. The dependency on $d$ and $T$ are tight (lower bound available in [Wagenmaker et al 2022], theorem 2) while we managed to push the curvature parameter $\kappa$ to the lower order terms. Hence we claimed that our bound is essentially tight.
>
> 2. **"Does the analysis of theorems implicitly rely on independence between the data used to construct ​ and the estimator"**. No, it does not. We completely incorporated the dependency of $\hat\theta_t$ on the data. The confidence set constructed fully respects this fact. See the confidence set of [Abeille et al 2021] which was constructed based on the self-normalized martingale inequality proposed by [Faury et al 2020] for more details.
>
> 3. **"How does L_t uniformly lower bound the true curvature"**. Please see our explanation of bullet No. 2 in the response to the weakness section. Feel free to let us know if the explanation is not satisfactory or there are further questions.
>
> 4. **"Alignment argument assume isotropy."** Similar to the argument given in bullet No. 2 in the response to the weakness section, the reason why we can use the property of an isotropic gaussian is we conduct a “noise-whitening” step on $\tilde\theta_t \sim \mathcal N(0,L_t^{-1})$.

---

> ### Comment · Reviewer_oWyi · 2025-11-26
> **Re: rebuttal**
>
> I thank the author for the rebuttal.
>
> Most responses make sense to me.
>
> Specifically,
> (1) I can buy that the result is probably tight from an educated guess without a proved statistical lower bound.
>
> (2) The explanation regarding L_t uniformly lower bounding the true curvature seems right.
>
> (3) I appreciate the clarification regarding the isotropy.
>
> Overall, I have no further concerns. I will maintain my score at 6 and recommend acceptance.

---

### Official Review · Reviewer_CC7u · 2025-10-31

**Soundness:** 4
**Presentation:** 4
**Contribution:** 3
**Rating:** 8
**Confidence:** 3

**Summary:**

This paper studies the stochastic contextual bandits with simple regret in both linear and logistic models. At the beginning of each round $t$, the learner observes a context $S_t$ sampled from unknown $\nu$. Then, the learner
chooses an action $A_t \in \mathcal{A}(S_t)$ given the past information. After that, they receive the reward $X_t \sim \mathcal{P}(\cdot | S_t, A_t)$.
The goal of the learner is to learn the policy $\pi$ that maps the context space to context-dependent action space. The value of policy $v(\pi)$
 is defined as the expected reward, where the randomness is due to unknown distribution $\nu$. The simple regret is now defined as $R(\pi):=\mathrm{sup}_{\pi}  v(\pi)-v(\pi)$.

Here, the optimal policy is  $\pi (s) = \mathrm{argmax}_{a \in A(s)}   \mu(\phi(s,a))^{\top} \theta^*)$.

The deterministic algorithm is very simple direct uncertainty maximization (MULIN)). It chooses the point that maximizes the uncertainty score $A_t = \mathrm{argmax}_a \| \phi(S_t,a) \|{V^{-1}_t}$ in the rounds $t=1, \ldots, T$. And finally compute $\hat{\pi}: s \rightarrow \mathrm{argmax}_a \phi(s,a)^{\top} \hat{\theta}_T$ where $\hat{\theta}_T $ is the minimizer of the loss function in the regression. The property of decreasing uncertainty is a key lemma together with the reliability of the confidence sets.
The second algorithm is a random one, Thompson Sampling (SIMPLELINTS) algorithm for linear contextual bandits and Try Hard Thompson Sampling (THATS) for logistic models.  All algorithms are efficient and the bounds are essentially tight.

**Strengths:**

1. The methods and analysis are unified in the sense that we can understand the intuitive and important theoretical property in the linear bandits and then the natural extension to the logistic model is described.  The paper is effortless to follow, and the text is flawless.
2. SIMPLELINTS (Theorem 3) achieves $\tilde{O}(d^{3/2}/\sqrt{T})$. Based on it, they also analyze a randomized logistic algorithm based on TS. These randomized methods have computational advantages over the deterministic method.
3. The dominant term in the simple is $\kappa$-free. This highlights the difference in complexity between cumulative vs simple regret.
4. Empirical comparisons of the proposed algorithms are also provided to support theoretical findings.

**Weaknesses:**

The motivation to study the simple regret in practice is not discussed.

**Questions:**

- How do the techniques share with pure exploration/best arm identification?
- What are the motivations/advantages to focus on simple regrets over cumulative regret or pure exploration/best arm identification setting?

---

> ### Author Response · Authors · 2025-11-21
> **Rebuttal to reviewer CC7u**
>
> We thank the reviewer for the careful reading of our paper and the appreciation of our result.  Here are the responses for the questions section:
> 1. Our result implies a best arm identification result. To be more specific,
>
> \begin{align}
> R(\hat \pi)&=\mathbb E[\mu(\phi(S,\hat\pi(S))^\top\theta_\star) -  \mu(\phi(S,\pi^\star(S))^\top\theta_\star)]\newline
> &\ge \mathbb E[\Delta_{\min} \mathbb I(\hat\pi(S) = \pi^\star(S))]\newline
> &=\Delta_{\min} \mathbb P(\hat\pi(S) = \pi^\star(S)),
> \end{align}
>
> where $\Delta_{\min}$ is the minimum reward gap.
> Rearranging the terms gives $\mathbb P(\hat\pi(S) = \pi^\star(S)) \le \frac{R(\hat \pi)}{\Delta_{\min}}$. Then for a target best arm identification probability $\zeta$, one can achieve it by controlling the simple regret to be less than $\zeta \Delta_{\min}$. Note that for logistic bandits this $\Delta_{\min}$ can scale with $1/\kappa$ and we do not think there is a way around it. The intuition is when two arms lie in the far right (left) side of sigmoid function, the difference is going to be extremely small as the derivative is almost $0$. On the other hand, when the optimal arm induces a mean that is extremely close to $1$ (that’s one case when $\kappa$ is large), we do not think it is meaningful to insist on finding the exact optimal arm as the difference is negligible in lots of scenarios.
>
> 2. Thanks for pointing out this issue. Due to the page limit we were unable to discuss the motivation of the simple regret objective but we are more than happy to include it in the camera ready version of this manuscript. As is mentioned above, a bound on the simple regret objective could imply a bound on best arm identification. In addition, for logistic bandits, when the optimal arm induces a mean that is extremely close to $1$, we do not think it is meaningful to insist on finding the exact optimal arm. For the advantage of simple regret over cumulative regret, there are some works in literature that motivate the simple regret objective, or best arm identification, and we kindly refer the reviewer to these works: [Mannor and Tsitsiklis 2004, Bubeck et al. 2008; Soare et al. 2014; Lattimore and Szepesvári 2020]. One of the scenarios where the authors encountered when collaborating with researchers from the application side is: for some of the industry applications, e.g., chip design, simulation might be expensive while the simulation results would have low variance, i.e., for multiple runs of simulation on the same input, the results differ very little. The goal is to use as little data as possible to identify good choices, and that’s when the simple regret objective becomes meaningful.
>
> ****
> References
>
> Zanette et al. 2021: Design of experiments for stochastic contextual linear bandits
>
> Wagenmaker and Jamieson 2022: Instance-Dependent Near-Optimal Policy Identification in Linear MDPs via Online Experiment Design
>
> Mannor and Tsitsiklis 2004 The Sample Complexity of Exploration in the Multi-Armed Bandit Problem
>
> Bubeck et al. 2008: Pure Exploration for Multi-Armed Bandit Problems
>
> Soare et al. 2014: Best-Arm Identification in Linear Bandits

---

### Official Review · Reviewer_gSdk · 2025-11-01

**Soundness:** 3
**Presentation:** 2
**Contribution:** 3
**Rating:** 6
**Confidence:** 3

**Summary:**

This paper studies contextual bandits with stochastic contexts under the simple-regret objective. Featuring a new analysis framework, the authors propose fully online algorithms for both the linear and logistic cases. A central point is that, through the use of a curvature-aware algorithm, the logistic simple-regret bounds have a leading term independent of $\kappa$, which can be as large as $\kappa \approx e^S$. To the best of my knowledge, this $\kappa$-free result for the logistic setting is new. The paper also provides empirical results illustrating why slope-weighted uncertainty is essential for efficient exploration in logistic models.

**Strengths:**

The logistic simple-regret setting is well motivated, and the work fills a clear gap in the literature. To my knowledge, this is the first paper to remove the dependence on the curvature constant $\kappa$ from the leading term of the regret bound. The construction of a monotone surrogate Hessian and the associated decreasing-uncertainty lemma are non-trivial and address the main technical challenge in logistic models, where the uncertainty depends on the unknown slope $\mu'(z)$. These ideas are conceptually elegant and potentially reusable in other generalized linear or reward-free settings.

**Weaknesses:**

Although there are no fatal theoretical flaws, the paper contains numerous typographical and consistency issues that make verification difficult. The most important ones are:

- In both MULIN and SIMPLELINTS, the design matrix $V_{t+1}$ is never updated. The pseudocode should include
  $V_{t+1} \leftarrow V_t + \phi(S_t, A_t)\phi(S_t, A_t)^\top.$

- From Equation (17), we have $\mathcal{V}_{t+1} \subseteq \mathcal{V}_t$, but it is reversed in Rows 1420-1421.

- The term $\varphi(s,a)^\top \theta^{\mathrm{Log}}_{T+1}$ at Rows 1014-1015 is duplicated and should appear only once in the expression for the prediction-error decomposition.

**Questions:**

I have no further questions.

---

> ### Author Response · Authors · 2025-11-21
>
> We appreciate the reviewer's careful reading of our paper and we promise we will try our best to fix the typos in the camera-ready version.

---

### Official Review · Reviewer_2EFJ · 2025-11-01

**Soundness:** 3
**Presentation:** 3
**Contribution:** 3
**Rating:** 4
**Confidence:** 3

**Summary:**

This study investigates simple-regret minimization in stochastic contextual bandits with linear and logistic models. To address this problem, the authors introduce deterministic max-uncertainty algorithms (MULIN/MULOG) and randomized variants designed for simple regret (SIMPLELINTS/THATS).

**Strengths:**

- The authors propose an effective algorithm for simple-regret minimization in stochastic contextual bandits. The regret guarantees are reasonable, and the authors provide sufficient explanations for their derivations.
- In particular, the finite-sample analysis is strong.

**Weaknesses:**

1. In my understanding, several other studies address simple-regret minimization in stochastic contextual bandits. For example, Kato et al. (2024) develop policy-learning algorithms in this setting. Their goal is to train a policy that minimizes simple regret in a best-arm-identification setting, and they characterize regret bounds using the VC dimension, which covers certain linear and logistic models. Theoretically, that analysis may be somewhat coarse, but could those results be applied to the authors’ setting?

- Kato, Okumura, Ishihara, and Kitagawa. "Adaptive Experimental Design for Policy Learning."

2. The result of this study is important, though not particularly surprising. I believe the problem itself is not especially difficult, and with sufficient effort, researchers could reasonably arrive at similar results. That said, this does not diminish the contribution of the study, as it offers meaningful implications for both theoretical and practical research. However, at least for me, the result is not especially exciting.

3. Is there a lower bound?

**Questions:**

1. I would like to understand the relationship between the proposed method and the Top-Two Thompson Sampling(TTTS)  algorithm introduced by Russo (2016). Although Russo’s method was originally developed for the fixed-confidence setting, it is now widely regarded as the standard variant of Thompson Sampling in pure exploration problems.

- Russo (2016). "Simple Bayesian Algorithms for Best Arm Identification"

2. Why do the authors use a variant of Thompson Sampling? In my understanding, for example, TTTS algorithms is used for avoiding complicated computation of optimal allocation ratios. Are there any advantages in the use of a variant of Thompson sampling in this setting?

3. Line 2: "Firstly, We" → "Firstly, we"

---

> ### Author Response · Authors · 2025-11-21
> **Rebuttal to Reviewer 2EFJ: Weakness part**
>
> We thank the reviewer for carefully reading our paper and the constructive comments. Below we respond to the points raised in the Weaknesses.
> 1. **"Comparison with Kato et al. (2024)"**. Thanks for the pointer to Kato et al. (2024) and we will include a discussion of their work in the revised version. While they indeed study policy learning with a simple-regret objective, **we believe that their results do not directly apply to our setting**. Here are the details:
>
>     (a). Their number of arms associated with each context is **finite**, whereas ours can be **infinite**. It is not clear to us how their bound can be transferred to our setting. When there are infinitely many arms and each arm has variance $\sigma^2$, Theorem 5.4 appears to **yield a bound that blows up to infinity**, as far as we can verify. This suggests that their analysis is too coarse to capture the specific structure we exploit in linear and logistic contextual bandits.
>
>     (b). They focus on unstructured bandits, where the reward distributions of different arms are mutually independent, while we consider structured contextual bandits (linear and logistic models) in the sense of Lattimore and Szepesvári (2020), chap 4.
>
> We will revise the related work section to (i) explicitly acknowledge Kato et al. (2024), (ii) clarify these differences in assumptions and goals, and (iii) explain why their bounds do not immediately subsume our results.
>
> 2. **"Problem not difficult; results not surprising"**. We appreciate the reviewer’s honest assessment. We **respectfully disagree** with the implication that the problem being “not especially difficult” or the result being “not surprising” should count against publication. As we discuss in the paper, it was not straightforward how to analyze Thompson Sampling in stochastic contextual linear bandits under a simple-regret objective. Even if, in hindsight, the final guarantees may appear “unsurprising,” the path to a correct and robust analysis is nontrivial and, to the best of our knowledge, was not available in the literature. We would like to emphasize on the contributions that deserves to be published:
>
>     (a). For **randomized algorithms**, even on the cumulative-regret side, the only Thompson Sampling variant on logistic bandits whose first order term is independent from $\kappa$ is TS-ECOLog from Faury et al (2022). However, there are some major issues with this algorithm:
>
>     + Their algorithm is designed for logistic bandit, and as far as the authors can verify, can be nontrivial to extend to **contextual** logistic bandits.
>
>     + Their algorithm relies on a rejection sampling step with no lower bound on the acceptance probability, so the expected running time **cannot be controlled** in general, which weakens its theoretical guarantees.
>
>     + Their algorithm includes a warm-up procedure that makes it remotely practical in real world scenarios, which **limits its practical relevance** under realistic time budgets.
>
>     (b). For the **deterministic algorithms**, if one writes out the form of decreasing uncertainty lemma, it contains a term involving an unknown quantity $H_t(\theta_*)$, which was **not obvious to control** without incurring an exponential dependence. The key idea of replacing $H_t(\theta)$ with $L_t$ is inspired by Faury et al. (2020), but making this replacement **computationally tractable** in the contextual logistic setting is nontrivial. As argued in the paper, one cannot simply replace the admissible parameter set of Faury et al. (2020) with the one from Abeille et al. (2021). We refer the reviewer to the discussion starting at line 1152 of the submitted manuscript for full details.
> In summary, we believe our work fills a few real gaps in the literature. **Given these clarifications, we kindly ask the reviewer to raise their score on our manuscript.**
>
> 3.  **"Lower bound available?"**. As far as we are aware, there is no lower bound available in literature and it is an interesting future direction.
>
> ****
> **References**
> Abeille et al. (2021): Instance-Wise Minimax-Optimal Algorithms for Logistic Bandits
>
> Faury et al. (2020): Improved Optimistic Algorithms for Logistic Bandits
>
> Faury et al. (2022): Jointly Efficient and Optimal Algorithms for Logistic Bandits
>
> Kato et al. (2024): Adaptive Experimental Design for Policy Learning

---

> ### Author Response · Authors · 2025-11-21
> **Rebuttal to Reviewer 2EFJ: Question part**
>
> We thank the reviewer for carefully reading our paper and the constructive comments. Below we respond to the points raised in the Weaknesses section.
> 1. **"Relationship to Top-Two Thompson Sampling (TTTS) (Russo, 2016)"**. Russo (2016) studies **a different problem setup**: finite-armed bandits with a **Bayesian** prior over the arm means, in a fixed-confidence best-arm identification framework. By contrast, our work considers:
>
>     + **Frequentist** stochastic contextual bandits with linear/logistic structure;
>     +  Potentially **infinitely** many arms (contexts);
>
> Our goal here is to remove the dependency of simple regret on the exponentially large parameter $\kappa$ while Russo (2016) focuses on identifying the best arm with a fixed confidence budget. It is **unclear** if there exists a direct way to adapt TTTS to this setting while preserving its theoretical guarantees.
>
> We will clarify in the paper that TTTS is tailored to Bayesian finite-armed best-arm identification, whereas our algorithms are designed for frequentist structured contextual bandits with potentially infinitely many arms.
>
> 2. **"Why a Thompson Sampling–type variant, and advantages of our choice"**. We chose a Thompson Sampling–type algorithm for several reasons:
>     + TS and its variants are already well-understood in contextual linear bandits under cumulative regret. Extending it to simple regret makes it **easier to adopt** in practice and **easier to compare** to existing baselines.
>     + TTTS is tailored to **Bayesian finite-armed best-arm identification** while our focus is on obtaining tight **frequentist** guarantees in **linear and logistic** contextual bandits under **simple regret** objectives. Our TS variant interacts cleanly with the structure of the confidence sets we construct, enabling finite-sample simple-regret bounds.
>     + Directly transplanting TTTS ideas into a structured bandit setting would require nontrivial and potentially costly computations to approximate optimal allocation ratios over **potentially infinitely large** action space. Our variant avoids these complications and remains computationally tractable, while having strong guarantees.
>
> We will expand the discussion in the paper to more clearly explain why we adopt a TS-style algorithm rather than TTTS, and what advantages this brings in our particular setting.
>
>
> 3. **Typographical issue**. We thank the reviewer for catching the typo “Firstly, We” on line 2. We will correct it to “Firstly, we” in the revised version.
>
> We hope these clarifications address the reviewer’s concerns and help convey both the technical substance and the broader relevance of our contributions. We kindly ask the reviewer to raise their score on our manuscript.

---

### Meta-Review · Area_Chair_iMR3 · 2026-01-05

**Summary:**

This work studies sampling-based algorithms for contextual linear bandits and logistic bandits. Reviewers acknowledged the importance of its findings for theory and practice. However, the results were not seen as surprising or originating from a problem of significant difficulty. The major technical contributions, particularly in sampling and design, are strongly based on existing approaches and are therefore incremental. As a result, my recommendation is to reject.

**Reviewer Concerns:**

The major remaining concern, as noted by Reviewer 2EFJ, is the lack of significant technical novelty. In their response, the authors have provided helpful clarifications on issues related to the lower bound and the covariance matrix $L_t$.

**Reviewer Scores:**

I believe the reviewers' scores are unlikely to change following the discussion.

---

### Decision · Program_Chairs · 2026-01-26

Reject